# LncRNA modulates Hippo-YAP signaling to reprogram iron metabolism

Xin-yu He[1,2,3,13], Xiao Fan[1,2,3,13], Lei Qu[1,2,3], Xiang Wang[4], Li Jiang[5], Ling-jie Sang[1], Cheng-yu Shi[1], Siyi Lin[1], Jie-cheng Yang[1], Zuo-zhen Yang[1], Kai Lei[1], Jun-hong Li[1], Huai-qiang Ju[6], Qingfeng Yan[1], Jian Liu[7,8], Fudi Wang[5], Jianzhong Shao[1], Yan Xiong[9], Wenqi Wang[10]✉ & Aifu Lin[1,2,3,11,12]✉

Iron metabolism dysregulation is tightly associated with cancer development. But the underlying mechanisms remain poorly understood. Increasing evidence has shown that long noncoding RNAs (lncRNAs) participate in various metabolic processes via integrating signaling pathway. In this study, we revealed one iron-triggered lncRNA, one target of YAP, *LncRIM* (LncRNA Related to Iron Metabolism, also named *ZBED5-AS1 and Loc729013*), which effectively links the Hippo pathway to iron metabolism and is largely independent on IRP2. Mechanically, *LncRIM* directly binds NF2 to inhibit NF2-LATS1 interaction, which causes YAP activation and increases intracellular iron level via DMT1 and TFR1. Additionally, *LncRIM*-NF2 axis mediates cellular iron metabolism dependent on the Hippo pathway. Clinically, high expression of *LncRIM* correlates with poor patient survival, suggesting its potential use as a biomarker and therapeutic target. Taken together, our study demonstrated a novel mechanism in which *LncRIM*-NF2 axis facilitates iron-mediated feedback loop to hyperactivate YAP and promote breast cancer development.

Metabolic reprogramming is a hallmark of malignancy[1]. Tumor cells require more nutrients, such as glucose, amino acids, and lipids, and energy than normal cells to meet the demands of uncontrolled proliferation[2–4]. Moreover, aberrant metabolic processes in cancer cells lead to inhibited immune cell infiltration and activation, ultimately promoting tumor cell proliferation, immune tolerance, and metastasis[5,6]. Therefore, metabolic differences between normal cells and cancer cells indicate targets for tumor therapies. Recent studies, including our studies, have revealed that the abnormal metabolism of micronutrients such as $Ca^{2+}$ [7] and $K^+$ [8] plays an important role in cancer development and the tumor microenvironment (TME)[9]. However, the specific mechanisms underlying

[1]MOE Laboratory of Biosystem Homeostasis and Protection, College of Life Sciences, Zhejiang University, 310058 Hangzhou, Zhejiang, China. [2]Cancer Center, Zhejiang University, 310058 Hangzhou, Zhejiang, China. [3]Key Laboratory for Cell and Gene Engineering of Zhejiang Province, 310058 Hangzhou, Zhejiang, China. [4]Department of Central Laboratory, the First People's Hospital of Huzhou, 158 Guangchang Back Road, 313000 Huzhou, Zhejiang, P.R. China. [5]Department of Nutrition, Precision Nutrition Innovation Center, School of Public Health, Zhejiang University School of Medicine, 310058 Hangzhou, Zhejiang, China. [6]Sun Yat-sen University Cancer Center, State Key Laboratory of Oncology in South China, Collaborative Innovation Center for Cancer Medicine, 510060 Guangzhou, Guangdong, China. [7]Hangzhou Cancer Institute, Affiliated Hangzhou Cancer Hospital, Zhejiang University School of Medicine, Zhejiang University, 310002 Hangzhou, Zhejiang, China. [8]Zhejiang University-University of Edinburgh Institute (ZJU-UoE Institute), Zhejiang University School of Medicine, International Campus, Zhejiang University, 314400 Haining, Zhejiang, China. [9]Department of Orthopedic Surgery, the Second Affiliated Hospital, School of Medicine, Zhejiang University, 88 Jiefang Road, 310000 Hangzhou, Zhejiang, China. [10]Department of Developmental and Cell Biology, University of California, Irvine, Irvine, CA 92697, USA. [11]International School of Medicine, International Institutes of Medicine, The 4th Affiliated Hospital of Zhejiang University School of Medicine, 322000 Yiwu, Zhejiang, China. [12]Breast Center of the First Affiliated Hospital, School of Medicine, Zhejiang University, 310003 Hangzhou, Zhejiang, China. [13]These authors contributed equally: Xin-yu He, Xiao Fan. ✉e-mail: wenqiw6@uci.edu; linaifu@zju.edu.cn

micronutrient metabolism, including iron metabolism mechanism, are still largely unknown.

As an essential nutrient, cellular iron participates in multiple biological processes and contributes to cell proliferation, TME establishment, and metastasis[10,11]. The pathways of iron acquisition, efflux, storage, and regulation are mostly altered in cancer cells[11]. However, cells with excess iron accumulation sometimes exhibit increased oxidative stress, which damages proteins, lipids, and DNA synthesis[12,13]. Recent studies have indicated that iron also regulates certain cancer-associated signaling pathways (e.g., WNT and JAK-STAT3)[14,15] to promote or inhibit cell proliferation and tumor growth. Divalent metal transporter 1 (DMT1; also called SLC11A2, Nramp2, and DCT1) and transferrin receptor 1 (TFR1) are two vital genes related to iron uptake[13,16]. In our study, we first verified that the oncogenic lncRNA *LncRIM* regulated cellular iron metabolism by altering the expression of downstream DMT1 and TFR1, providing new insights into physiological homeostasis and cancer treatment.

The Hippo pathway has been demonstrated to play vital roles in tumorigenesis, regeneration, and organ size by regulating the expression of various signaling molecules[17,18]. Notably, the Hippo pathway also responds to diverse upstream signaling, such as glucose energy stress, G protein-coupled receptor (GPCR), lipid and mechanoresponse signaling[19–23]. The tumor suppressor Merlin/NF2, encoded by neurofibromatosis type II (*NF2*), is located on the cell plasma membrane and interacts with LATS1 to induce NF2-mediated LATS1 membrane translocation, which ultimately leads to YAP retention in the cytoplasm and degradation[24,25]. Studies have reported that NF2 mutation is sufficient to induce tumor initiation and growth[26,27]. Interestingly, a recent study demonstrated that hyperactivated Merlin-YAP signaling controlled ferroptosis in association with intercellular interactions[28], suggesting a potential role for the Hippo pathway in cellular iron metabolism. Aberrant metabolic processes are hallmarks of cancer and are accompanied by the activation of crucial oncogenesis-related signaling pathways. In addition, our previous study revealed an oncogenic function of YAP via promoting aerobic glycolysis, also known as the Warburg effect[20,29]. However, whether the Hippo signaling pathway and micronutrient metabolism synergistically function to advance cancer progression is still unknown.

Extensive evidence has indicated that lncRNAs are critical regulators of signal transduction and cellular metabolism remodeling[3,7,29–31]. LncRNA-mediated regulation is related to its subcellular location. Recently, many studies, including our previous works, have shown that cytoplasmic lncRNAs participate in multiple metabolic homeostatic pathways and human cancer development; the mediated processes (genes) include glucose metabolism (e.g., *BCAR4* and *GASS*)[3,29], cellular lipid metabolism (e.g., *SNHG9* and *LINK-A*)[30,32], and Ca²⁺-dependent signaling[7]. Thus, further clarification and decoding of subcellular-associated lncRNAs is expected to provide new insights into metabolic reprogramming in human cancer.

In this study, we discovered crosstalk between lncRNA-mediated iron metabolism and the Hippo pathway. We demonstrated that *LncRIM*, a gene downstream of YAP, promoted cell proliferation and tumor growth by regulating cellular iron levels and coordinating with the Hippo pathway signaling. Mechanistically, *LncRIM* directly bound NF2 and inhibited the NF2-LATS1 interaction, which subsequently activated YAP and increased the intracellular iron level by promoting the expression of DMT1 and TFR1. Moreover, we proved that this *LncRIM*-NF2 axis also functions effectively as well as the IRP2/IRE in cellular iron metabolism and largely independent on the IRP2. Besides, the *LncRIM*-NF2 axis further facilitates iron-mediated Hippo-YAP feedback loop regulation. Clinically, the high expression of *LncRIM* and its coordinator DMT1/TFR1 has been shown to be associated with poor clinical outcomes, indicating a novel role for *LncRIM* in breast cancer therapy. Together, our findings revealed that iron-related

*LncRIM* is a crucial regulator of Hippo pathway signaling, cellular iron metabolism, and breast cancer progression.

## Results
### *LncRIM* regulates iron metabolism and breast cancer progression

To identify cellular iron metabolism-associated potential signaling and molecules, we initially performed gene set enrichment analysis (GSEA) by using lipocalin (LCN2)-knockdown cell lines. LCN2-mediated iron transport differs from that in the TF-TFR1 pathway and is involved in diverse cellular physiological processes[33]. As an important iron transporter, LCN2 has been reported to be upregulated in cancer cells and to promote tumor growth by regulating cellular iron accumulation, and knocking down LCN2 has been shown to inhibit cell proliferation and reduce the cellular iron level[34]. Intriguingly, the results of the present study showed that YAP was significantly inactivated when the cellular iron level was reduced ($P = 0.002$, NCBI/GEO/GSE38369, Figs. 1a and S1a). To confirm this observation, calcein-acetoxymethyl (Calcein-AM), a widely used compound to measure cellular iron levels via its fluorescence signal intensity, as the calcein fluorescence level is inversely correlated with the iron level, was used to assess the cellular iron level of MCF-7 cells stably transfected with an empty vector, YAP, an active YAP mutant (YAP-5SA), or an inactive YAP mutant (YAP-S94A)[35]. Our results showed that overexpression of YAP and YAP-5SA significantly increased iron levels, while YAP-S94A did not exert this effect (Fig. 1b, c). In addition, as shown in Fig. 1d, e, YAP activation was positively correlated with the iron level in breast cancer, indicating a potential role for the Hippo pathway in cellular iron metabolism.

Considering the vital role played by lncRNAs in various metabolism-associated signaling pathways, we identified promising lncRNAs involved in iron metabolism by analyzing our previous database[36], in which we had previously identified 40 lncRNAs that were potentially required for YAP1-dependent transcription. We then treated the MCF-7 cells with the iron chelator desferrioxamine (DFO) to decrease the intracellular levels of iron, or ferric ammonium citrate (FAC) to increase the intracellular iron content (Fig. 1f). As shown in Fig. 1g, FAC treatment promoted the expression of several lncRNAs, while DFO treatment significantly decreased the levels of several lncRNAs, which indicated that the lncRNAs that responded to both FAC and DFO stimulation might be involved in iron metabolism homeostasis in cancer cells. We then assessed the regulatory role played by these lncRNAs in iron metabolism by knocking down 6 overlapping lncRNAs (Fig. S1b). We found that knocking down the lncRNA *Loc729013*, hereafter, *LncRIM* (a LncRNA Related to Iron Metabolism), profoundly decreased the cellular iron level (Fig. 1h). A coding potential assessment tool (CPAT) prediction showed that the coding probability of *LncRIM* (NR_034137.1) was 0.0424, much lower than the cutoff value (0.364). In addition, polysome profiling results showed that *LncRIM* was negligibly enriched in polysome components. These findings indicated that *LncRIM* shows little ability to encode proteins (Fig. S1c, d). In addition, given the correlation of *LncRIM* with the tumor-suppressing Hippo signaling pathway, we next examined the functional relationship between *LncRIM* and cancer development. The results showed that *LncRIM* was highly expressed in tumor tissues compared with paired control samples (Figs. 1i and S1e), and similar findings were observed with multiple cell lines of differing breast cancer types (Fig. S1f). In addition, high *LncRIM* expression was correlated with low survival of breast cancer patients in an independent cohort (Fig. 1j). Consistent with this finding, knocking *LncRIM* inhibited the proliferation of MCF-7 cells (Fig. S1g). Together, these data strongly suggested that *LncRIM* plays an important role in breast cancer development.

Many studies have identified lncRNA-mediated cellular metabolic process dysregulation during tumor initiation and tumor

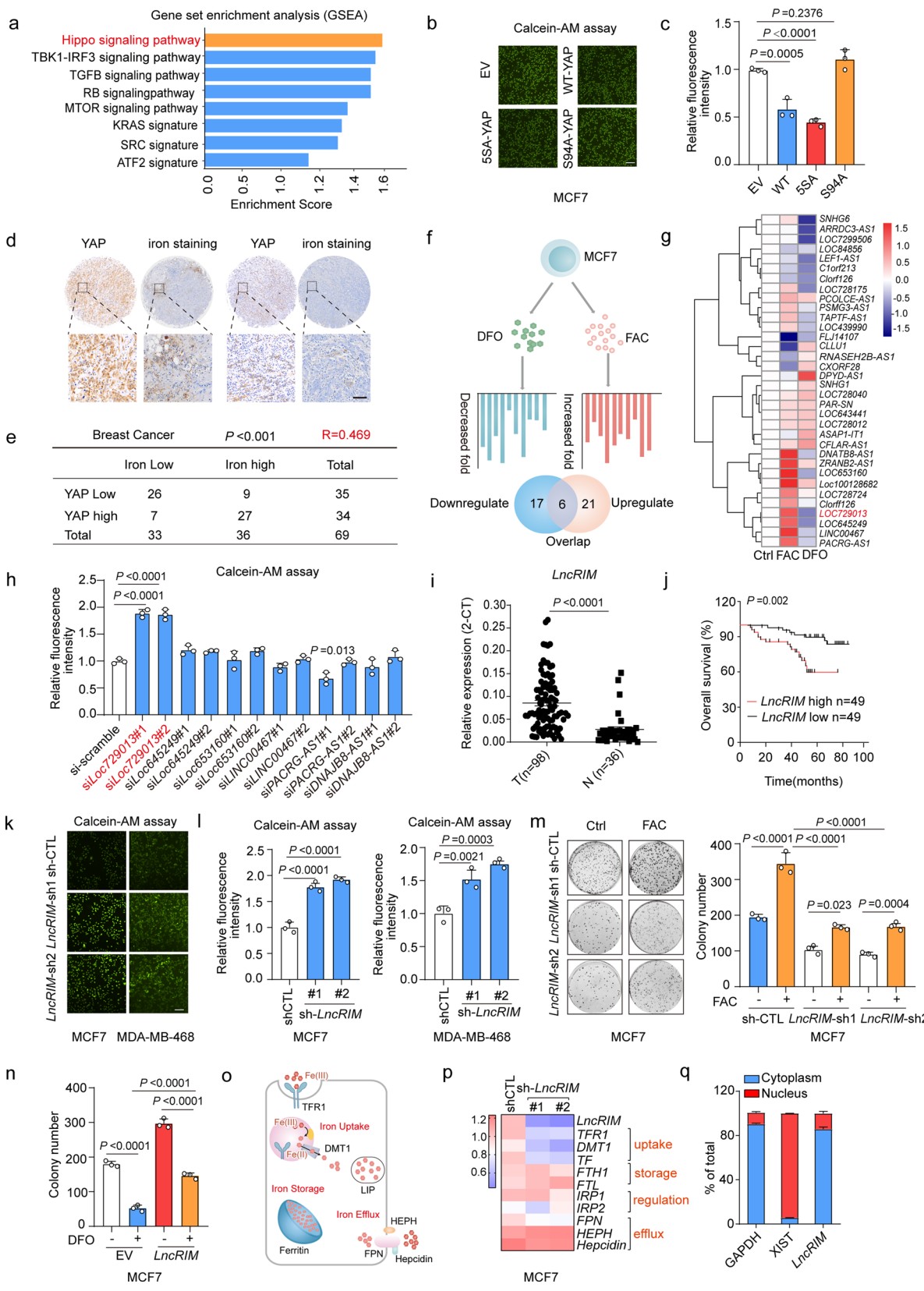

development[3,30]. Therefore, we examined whether *LncRIM* coordinated cellular iron metabolism to induce breast cancer progression. We constructed MCF-7 and MDA-MB-468 cell lines with *LncRIM* over-expressed or knocked down. As shown in Figs. 1k, l and S1h–j, knocking down *LncRIM* significantly decreased the cellular iron level, while

overexpression of *LncRIM* increased the cellular iron level. Notably, as shown in Figs. 1m and S1k–l, FAC stimulation partially reversed the diminished cell proliferation and cell cycle arrest in *LncRIM*-silenced MCF-7 and MDA-MB-468 cells. Moreover, *LncRIM* overexpression partially reduced the inhibition of DFO on cell proliferation (Figs. 1n

**Fig. 1 | LncRIM regulates iron metabolism and breast cancer progression.**
**a** Gene set enrichment analysis using the C6 canonical pathways Broad MsigDB database on gene expression data compared control and LCN2 knockdown MDA-MB-231 cell lines. **b, c** Calcein-AM analysis cellular iron level of MCF-7 cells transfected with indicated YAP mutant (**b**). Scale bar, 100 μm. Values were normalized to control group (**c**). (mean ± SD, n = 3, One-way ANOVA analysis). **d** Enhanced DAB iron staining (n = 69) and immunohistochemistry staining of YAP (n = 69) in breast cancer tissue arrays. Scale bar, 100 μm. **e** Correlations between the YAP and iron levels in human breast tumors were analyzed. two-sided chi-square test; R, correlation coefficient. **f, g** Schematic illustration of the analysis of lncRNA profiles stimulated with FAC (100 μM) or DFO (100 μM) from the human Lincode® siRNA library into MCF-7 cells that were engineered with a TEAD-driven luciferase reporter (**f**). The representative candidates were assayed by RT–qPCR (**g**) (mean ± SD, n = 3, One-way ANOVA analysis). **h** Calcein-AM analysis the cellular iron level of HEK-293T with candidate lncRNAs knockdown. Values were normalized to control group. (mean ± SD, n = 3, two-sided Student's t test). (**i**) RT–qPCR detected the LncRIM

expression in breast cancer tissues (n = 98) and paired adjacent tissues (n = 36). The horizontal black lines represent median values. (mean ± SD, two-sided Student's t test). **j** Kaplan–Meier analysis of overall survival of breast cancer patients with low versus high expression of LncRIM (n = 98, Gehan–Breslow test). **k, l** Calcein-AM detected cellular iron level of control and LncRIM knockdown MCF-7 cells (**k**). Scale bar, 100 μm. Values were normalized to control group (**l**). (mean ± SD, n = 3, One-way ANOVA analysis). **m** Colony formation assay of control and LncRIM knockdown MCF-7 cells with or without 200 μM FAC stimulation (mean ± SD, n = 3, Two-way ANOVA analysis). **n** Colony formation assay of EV and LncRIM overexpressed MCF-7 cells treated with or without DFO (100 μM) (mean ± SD, n = 3, Two-way ANOVA analysis). **o** Graphical illustration of the main proteins involved in the cellular iron homeostasis. **p** RT–qPCR and Heatmap detected the expression of iron metabolism-related genes in control and LncRIM knockdown MCF-7 cells. (mean ± SD, n = 3, One-way ANOVA analysis). **q** RT–qPCR detection of LncRIM expression in cytoplasmic and nuclear fractions. (mean ± SD, n = 3).

and S1m). However, we should acknowledge the limitation of LncRIM in this assay with DFO decreased cell proliferation around 70% in the EV group and around 50% in the LncRIM overexpressed group. Together, these data indicated that iron homeostasis was important for LncRIM-modulated cell proliferation and survival.

Cellular iron metabolism is regulated by many genes related to iron uptake, storage, and efflux[13] (Fig. 1o). We performed RT–qPCRs to assess the expression change of these genes after LncRIM was knocked down or overexpressed. As shown in Figs. 1p, S1n, and S1o, the expression of TFR1 and DMT1 was significantly decreased in LncRIM-knockdown cell lines, while overexpression of LncRIM increased the expression of these genes. The protein levels of DMT1 and TFR1 were changed consistently with this observation (Fig. S1p). Moreover, LncRIM is a YAP-transcription related lncRNA, knocking down LncRIM decreased the expression of genes downstream of YAP (Fig. S1q), suggesting a potential link between LncRIM-mediated cellular iron metabolism and the Hippo pathway. The functions and specific mechanisms of many lncRNAs are related to their subcellular location[37]. Therefore, we assessed the cellular locations of LncRIM. The data showed that LncRIM was mostly localized to the cytoplasm (Fig. 1q). Taken together, our data revealed a newly discovered cytoplasmic lncRNA, LncRIM, which was shown to play a critical role in cellular iron metabolism and breast cancer development.

### LncRIM interacts with NF2 to inactive LATS1 kinase

Cytoplasmic lncRNAs have been thought to form RNA–protein complexes that regulate various cellular physiological processes[38]. Considering the function of LncRIM in both cellular iron metabolism and the Hippo pathway (Figs. 1p and S1q), we performed an RNA pull-down assay using MCF-7 cell lysates to identify potential proteins in the Hippo pathway that might be involved in the LncRIM-related iron metabolic process. Interestingly, sense LncRIM, but not antisense LncRIM, was found to bind to NF2, an important membrane–cytoskeleton scaffold upstream of the Hippo-YAP signaling pathway (Fig. 2a). However, other Hippo pathway components, including YAP, LATS1, MOB1, and MST1, were not associated with sense-strand LncRIM (Fig. S2a). In addition, an RNA–protein binding assay with recombinant NF2 verified a direct interaction between LncRIM and NF2 (Fig. 2b, c). To confirm this finding, an RNA immunoprecipitation (RIP) assay with MCF-7 cells expressing SFB-NF2 was performed. The results also showed that LncRIM directly interacted with NF2 (Fig. 2d). In addition, as shown in Fig. S2b, c, approximately 690 copies of LncRIM were found for every MCF-7 cell and 502 copies of LncRIM were found for every MDA-MB-468 cell, indicating a relatively high abundance of LncRIM compared with that of several other functional lncRNAs: For example, approximately 150 copies of LINK-A have been found for every MDA-MB-231 cell, and

approximately 937 copies of CamK-A have been found per MDA-MB-231 cell[7,32]. Together, these data suggested that LncRIM might regulate cellular iron metabolism through NF2-associated mechanisms.

More importantly, LncRIM and NF2 were colocalized to the cell membrane, as shown by RNA fluorescence in situ hybridization (FISH) (Fig. 2e, f). However, overexpression of LncRIM did not exert an impact on the membrane expression of NF2 (Fig. S2d, e). As a tumor suppressor, NF2 interacts with and recruits the kinase LATS1 to the plasma membrane, which leads to the subsequent phosphorylation of LATS1 and the cytoplasmic retention of YAP[25]. We therefore assessed the effect of LncRIM on the interaction of NF2 and LATS1 and the recruitment of LATS1 to the plasma membrane. As shown in Figs. 2g, h and S2f–h, overexpression of LncRIM significantly reduced the interaction between NF2 and LATS1, while knocking down LncRIM markedly increased the NF2-LATS1 interaction. Consistent with these findings, immunofluorescence (IF) and subcellular fractionation assays confirmed that LncRIM decreased the NF2-induced membrane translocation of LATS1 (Figs. 2i–k and S2i). The N-terminus of LATS1 (LATS1-NT) contributes to the binding of NF2[39]. By using recombinant GST-LATS1-NT and His-NF2, we performed GST pull-down assays with different concentrations of in vitro-transcribed sense LncRIM to assess the binding kinetics of NF2 and LTAS1, and the findings were consistent with aforementioned observations (Fig. S2j, k). Furthermore, we constructed two deletion mutants of NF2 and performed protein purification (Figs. 2l and S2l). The co-IP result showed that the C-terminal domain (CTD)-deletion NF2 mutant bound LncRIM as efficiently as wild-type NF2, while the FERM-domain-deletion NF2 mutant showed no binding with LncRIM (Fig. 2m). A previous study had reported that the FERM domain consists of three lobes (F1, F2, and F3) and that F2 is critical for the interaction of NF2 and LATS1[39]. To further analyze the precise interaction of LncRIM and NF2, we constructed three different NF2 deletion mutants. As shown in Fig. S2m, the F2 (112–212 amino acids) and F3 (220–311 amino acids) lobes were equally crucial for the LncRIM-NF2 interaction. Moreover, considering the predicted secondary structure, we constructed LncRIM mutants with three different loops truncated (the S1 mutant consisted of nt 1–580, the S2 mutant consisted of nt 581-893, the S3 mutant consisted of nt 894–1113) (Fig. S2n). A binding analysis showed that the LncRIM S1 truncation was crucial for the LncRIM-NF2 interaction (Fig. 2n). Additionally, restoration of the S1 domain loop significantly inhibited the binding between LATS1 and NF2 in LncRIM-silenced cells, showing an effect similar to that of full-length LncRIM (Fig. 2o).

We then assessed the function of the LncRIM-NF2 axis in cellular iron metabolism. Interestingly, we found that overexpression of either the full-length LncRIM or the S1 truncation loop sufficiently restored the cellular iron level in LncRIM-silenced cells (Fig. 2p) and reversed the phosphorylation of YAP and LATS1

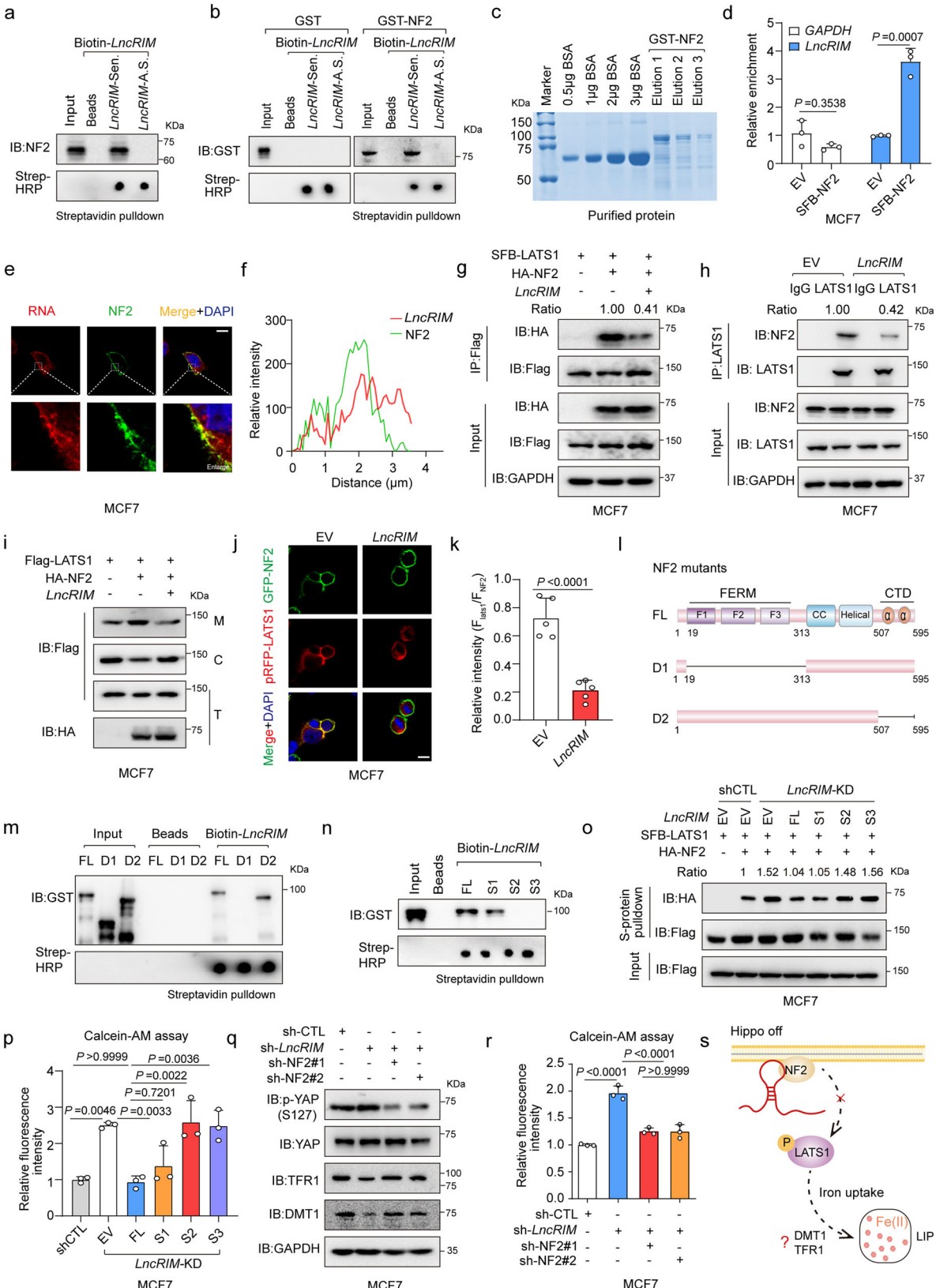

(Fig. S2o). In addition, knocking down NF2 in *LncRIM*-deficient cells partially restored YAP activation, the expression of DMT1 and TFR1 and the cellular iron level (Fig. 2q, r). Together, these data clearly indicated a novel mechanism through which *LncRIM* regulates cellular iron metabolism by directly binding NF2 to inhibit the NF2-LATS1 interaction (Fig. 2s).

## *LncRIM* modulates iron metabolism in a Hippo-YAP pathway-dependent manner

We then investigated the link between *LncRIM*-NF2 axis-mediated cellular iron metabolism and the Hippo-YAP pathway. We found that the phosphorylation of LATS1 at Ser909 and Thr1079 and that of YAP at Ser127 were decreased and that the expression of DMT1 and TFR1

**Fig. 2 | LncRIM interacts with NF2 to inactive LATS1 kinase. a** In vitro-transcribed biotinylated *LncRIM* sense (Sen.) or antisense (A.S.) transcripts were incubated with MCF-7 cell lysates for RNA pull-down assay. The biotin-RNAs was detected by dot blot using streptavidin-HRP. **b** In vitro-transcribed biotinylated *LncRIM* transcripts were incubated with GST-NF2 recombinant proteins for RNA pull-down assay. **c** Coomassie staining gel of the purified GST-NF2 protein. **d** The RIP assay and RT-qPCR were performed to assess the enrichment of NF2 on *LncRIM*. (mean ± SD, *n* = 3, two-sided Student's *t* test). **e, f** RNA FISH assay and immunofluorescent staining to examine the localization of *LncRIM* and NF2 (**e**). A line scan of the relative fluorescence intensity of the signal (dotted line) is plotted to show the peak overlapping (**f**). The *LncRIM* probe was labeled with Cy3 (Red) and NF2 was detected with Alexa Fluor 488(Green). Scale bar, 10 μm. **g, h** Co-IP analysis for the interaction between LATS1 and NF2 in *LncRIM* overexpressed MCF-7 cells. **i** Analysis of LATS1 subcellular localization in the fractions of MCF-7 cells with overexpression of *LncRIM* (M: membrane, C: cytoplasm, T: total). **j, k** Immunofluorescent staining to detect the interaction between LATS1 (Red) and NF2 (Green) in control and *LncRIM* overexpressed MCF-7 cells. Scale bar, 10 μm. (mean ± SD, *n* = 5, two-sided Student's *t* test). **l** Schematic illustration of NF2 structures and truncated mutants. **m** Wild type and NF2 mutant recombinant proteins were incubated with in vitro-transcribed *LncRIM*, and pulled down by streptavidin beads. **n** Immunoblot detection of GST-NF2 protein retrieved by in vitro–transcribed biotinylated *LncRIM* and different *LncRIM* truncations (S1, S2, and S3). **o** Co-IP assay was performed to detect the interaction of LATS1 and NF2 with different truncations of *LncRIM*. **p** Calcein-AM analysis cellular iron level of MCF-7 cells transfected with different truncations of *LncRIM*. The values were normalized to the control group. (mean ± SD, *n* = 3, One-way ANOVA analysis). **q, r** The DMT1, TFR1 expression (**q**) and the cellular iron level (**r**) of control and *LncRIM* knockdown MCF-7 cells with NF2 silence. The values were normalized to the control group (**r**). (mean ± SD, *n* = 3, One-way ANOVA analysis). **s** Graphical illustration of *LncRIM*-NF2 axis in cellular iron metabolism.

was increased when *LncRIM* was overexpressed; in contrast, the opposite results were observed when *LncRIM* was knocked down (Fig. 3a, b). More importantly, overexpression of *LncRIM* in YAP-silenced MCF-7 cells did not promote the expression of DMT1 and TFR1 or increase the iron level, in contrast to the effect on control cells (Fig. 3c–f). Notably, re-expression of YAP significantly restored the expression of DMT1 and TFR1 as well as that of YAP target genes to levels similar to that induced by *LncRIM* overexpression, and the cellular iron level was restored (Figs. 3g, h and S3a, b). These findings suggest important roles for the Hippo pathway in the *LncRIM*-NF2 axis-mediated regulation of cellular iron metabolism.

Interestingly, overexpression of YAP-5SA but not YAP-S94A led to significantly increased DMT1 and TFR1 expression (Fig. S3c–e), indicating that DMT1 and TFR1 were positively regulated by YAP. Moreover, wild-type YAP and the YAP-5SA mutant significantly diminished the inhibitory effect of *LncRIM* silencing on the expression of DMT1 and TFR1 and counteracted the reduction in the cellular iron level (Fig. S3f, g). A previous study reported that TFR1 is a downstream target of YAP[28]. To determine whether YAP directly regulates DMT1 transcription, we analyzed the DMT1 promoter by using GEO database (GSE107013)[40], and identified one YAP/TEAD4-binding site (Figs. 3i and S3h). Moreover, luciferase reporter and chromatin immunoprecipitation (ChIP-PCR) assay results together showed that YAP directly bound to the DMT1 promoter region (Figs. 3j and S3i). Further, deletion of the YAP/TEAD-binding site (CATTCT) in the DMT1 promoter significantly attenuated YAP-5SA-induced DMT1 promoter luciferase activity (Fig. 3k). Together, these results demonstrated that DMT1 was a direct transcriptional target of YAP.

DMT1 mRNA encodes four different isoforms due to variations in the 3′-UTR (an iron-responsive element (IRE)-containing or a non-IRE-containing UTR) and 5′ end mRNA-processing variants (1A and 1B)[41,42]. We examined the expression of these four isoforms in breast cancer cells with a specific set of pre-established targets of these four isoforms. As shown in Fig. S3j, DMT1 isoform 1 (the IRE-containing variant) was highly expressed in MCF-7 and MDA-MB-468 cells, among all isoforms. In addition, we found that knocking down IRP2 decreased DMT1-IRE expression but did not affect the expression of non-IRE-containing DMT1 (Fig. S3k). Previous studies have reported that cellular iron homeostasis is largely controlled by an iron regulatory protein (IRP)/IRE system[10,43,44]. IRP2, but not IRP1, has been previously shown to regulate iron homeostasis in breast cancer[45,46]. Therefore, we compared *LncRIM*-NF2 axis activation in cellular iron metabolism on the basis of a previously reported IRP2 system. As shown in Figs. 3l, m and S3l, m, knocking down IRP2 significantly decreased the expression of both DMT1 and TFR1, however, overexpression of *LncRIM* and YAP still partially rescued the expression of DMT1 TFR1, and the cellular iron level after IRP2 knockdown. However, due to the limitation of IRP2 knockdown, these results indicated that the *LncRIM*-NF2 axis is largely independent of IRP2 system to regulate cellular iron metabolism, but

not completely. Furthermore, the effects of *LncRIM* knockdown were comparable to those of IRP2 knockdown; that is, they both led to decrease in the DMT1 level (54% for IRP2 vs. 52% for *LncRIM*) and TFR1 (51% for IRP2 vs. 59% for *LncRIM*) as well as on the decrease in cellular iron concentration (42% for IRP2 vs. 36% for *LncRIM*) (Fig. S3n–q). In conclusion, above data together suggested that the *LncRIM*-NF2 axis played an important role in cellular iron metabolism perhaps in a manner differs from the classical IRP/IRE system.

## The iron-triggered LncRIM-NF2 feedback loop hyper-activates YAP

The Hippo signaling pathway precisely regulates cellular physiological activities and feedback is a common regulatory mechanism in the Hippo signaling pathway[47,48]. Indeed, in this study, *LncRIM* was shown to both respond to iron stimulation and regulate the Hippo signaling pathway (Figs. 1g and 3a, b). To further explore the potential effects of the cellular iron level on the Hippo pathway, we stimulated MCF-7 cells and MDA-MB-468 cells with FAC or DFO for the indicated times. The phosphorylation of YAP and LATS1 was robustly decreased after FAC stimulation, while DFO stimulation increased the phosphorylation levels of LATS1 and YAP (Figs. 4a and S4a). In addition, DFO stimulation upregulated Hippo pathway activation in a dose-dependent manner, while the addition of FAC decreased YAP phosphorylation in a time-dependent manner (Fig. S4b, c). Notably, as shown in Fig. 4b, a lower concentration of FAC significantly decreased the phosphorylation of YAP and LATS1, while a higher concentration of iron restored the phosphorylation levels of these proteins. Unexpectedly, we also found that the expression of *LncRIM* after FAC or DFO stimulation was consistent with the change in YAP activation (Fig. 4c, d). Similar to other YAP target genes, the YAP-5SA mutant clearly promoted the expression of *LncRIM* (Fig. S4d), which suggested that *LncRIM* may be positively regulated by YAP. We then carried out luciferase reporter and ChIP-PCR assays to verify this possibility. As shown in Fig. 4e, f, YAP/TEAD significantly increased *LncRIM* promoter luciferase activity and directly bound to the *LncRIM* promoter. And these results suggested the potential feedback loop of *LncRIM*-Hippo in cellular iron metabolism. Excessive iron has been increasingly considered to be an important mediator of cell death, such as apoptosis and ferroptosis by producing reactive oxygen species (ROS)[49,50]. In addition, studies also showed YAP is inactivated in response to apoptosis induction[51]. We then measured cell viability after FAC treatment at different concentrations. The results showed that a higher concentration of iron led to slower cell proliferation, while a lower concentration of FAC significantly promoted cell proliferation compared to the effect of either change on the control groups (Fig. S4e), which was consistent with the YAP activation change and Gaussian effect of iron-triggered *LncRIM* expression in Fig. 4b, c.

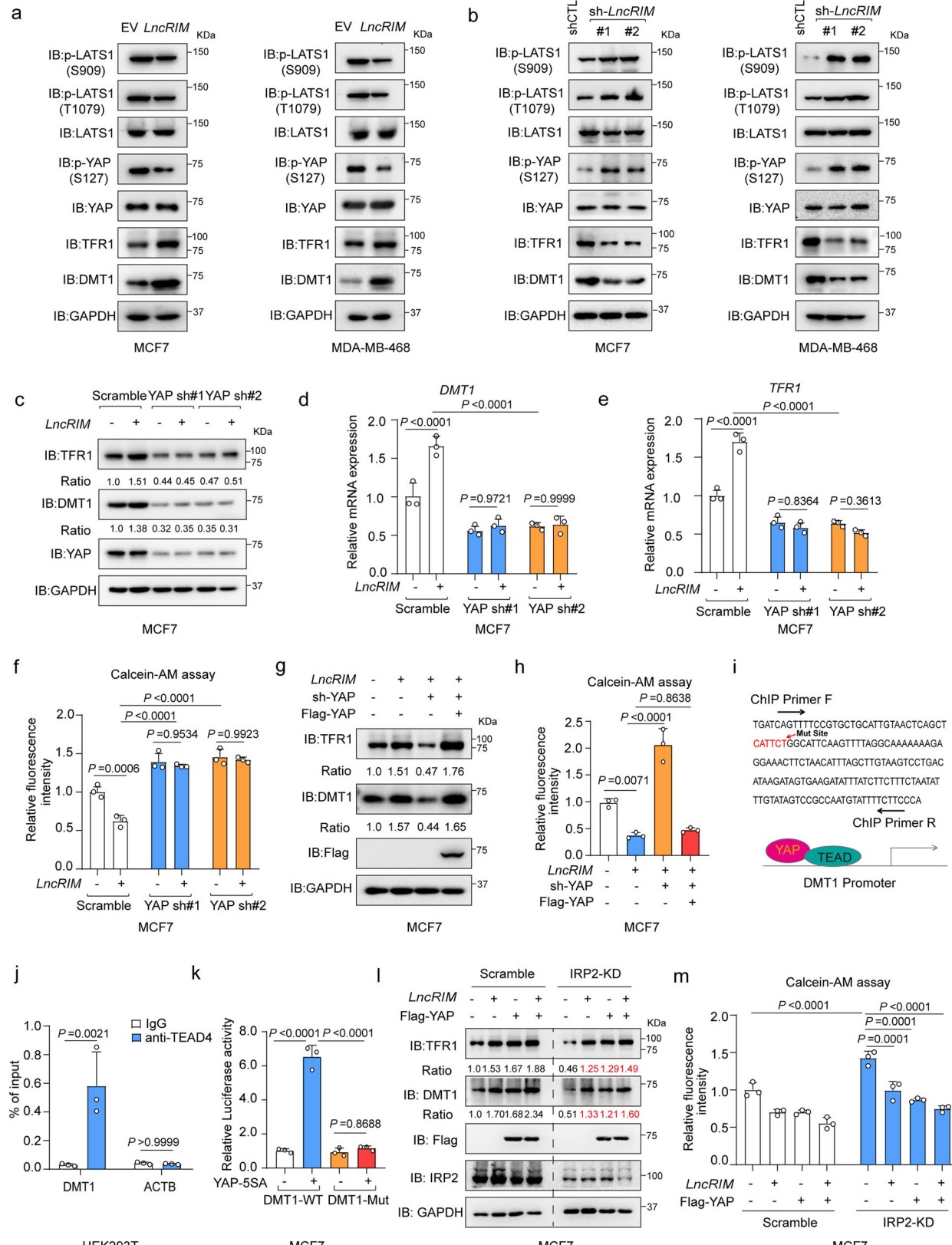

Further, we found that FAC treatment markedly enhanced the interaction of *LncRIM* and NF2 and reduced the binding of NF2 to LATS1 (Fig. 4g, h), which indicated a crucial role for iron-triggered *LncRIM* in the iron-induced modulation of Hippo-YAP signaling pathway activation. Besides, the IF assay results showed that iron overload increased the expression of *LncRIM* on the cell membrane and the

interaction between *LncRIM* and NF2, while further decreasing LATS1 recruitments to the plasma membrane (Fig. S4f, g). However, we cannot ignore the effect of iron itself on protein binding.

Considering the feedback loop of the *LncRIM*-NF2 axis in response to iron stimulation, we treated MCF-7 and MDA-MB-468 breast cancer cells with FAC under control, *LncRIM*-knockdown, and *LncRIM*-

**Fig. 3 | *LncRIM* modulates iron metabolism in a Hippo-YAP pathway-dependent manner. a**, **b** The expression of DMT1 and TFR1, and the levels of p-LATS1 (S909, T1079), and p-YAP (S127) were detected in control, *LncRIM* knockdown or *LncRIM* overexpressed MCF-7 and MDA-MB-468 cells. **c**–**e** *LncRIM* promoted the expression of DMT1 and TFR1 in YAP-dependent manner. Immunoblot (**c**) and RT-qPCR were performed to detect the expression of DMT1 (**d**) and TFR1 (**e**) in control and YAP knockdown MCF-7 cells with or without overexpression of *LncRIM*. (mean ± SD, *n* = 3 biologically independent experiments, two-way ANOVA analysis). **f** The cellular iron level of control and YAP-silenced MCF-7 cells with or without *LncRIM* overexpression was measured by Calcein-AM assay. Values were normalized to the control group (mean ± SD, *n* = 3, Two-way ANOVA analysis). **g**, **h** The expression of DMT1 and TFR1 (**g**) as well as the cellular iron level (**h**) was detected by immunoblot and Calcein-AM assay in YAP-rescued MCF-7 cells. Values were normalized to those in the control group. (mean ± SD, *n* = 3, One-way ANOVA analysis). **i** MEME analysis of YAP/TEAD binding motif in the DMT1 promoter region by using GSE38369. **j** YAP/TEAD directly regulates the transcription of DMT1. Chromatin immunoprecipitation (ChIP) assay combined with RT-qPCR were performed by using the IgG and TEAD4 antibodies. (mean ± SD, *n* = 3, two-sided Student's *t* test). **k** Luciferase reporter assay was performed with overexpression of YAP-5SA and DMT1-promoter or DMT1-promoter deleted mutants (CATTCT) in MCF-7 cells (mean ± SD, *n* = 3, Two-way ANOVA analysis). **l**, **m** Immunoblot (**l**) and Calcein-AM assays (**m**) were performed to assess the expression of DMT1, TFR1 and iron level in control or IRP2 knockdown MCF-7 cells with overexpression of YAP or *LncRIM*. (mean ± SD, *n* = 3, two-way ANOVA analysis).

overexpression conditions. The results showed that FAC treatment partially restored *LncRIM* silencing-induced phosphorylation of YAP and LATS1 and further enhanced the activation of YAP when *LncRIM* was overexpressed (Fig. 4i, j). Interestingly, iron overload increased the expression of genes downstream of YAP, with the increase obvious after exposure to FAC for at least 36 h (Fig. S4h). The *LncRIM*-Hippo feedback loop significantly attenuated the inhibitory effect of *LncRIM* knockdown on the YAP target genes *CTGF* and *CYR61* and increased the *LncRIM* overexpression-mediated promotion of YAP downstream targets after FAC stimulation (Fig. S4i, j).

After shuttled into endosomes via the TF-TFR1 pathway and the action of DMT1, intracellular iron can be immediately used, and excess iron promotes ferritin synthesis[52]. We illustrated that knocking down *LncRIM* downregulated the expression of the iron uptake proteins DMT1 and TFR1. As expected, knocking down *LncRIM* significantly inhibited ferritin synthesis, in contrast to its production in control cells after iron stimulation; in contrast, the overexpression of *LncRIM* led to the opposite results (Fig. 4i, j), further demonstrating the important role of *LncRIM* in regulating cellular iron metabolism. In addition, an IF assay showed that knocking down *LncRIM* resulted in YAP sequestration in the cytoplasm, while FAC stimulation largely ameliorated the translocation of YAP into the nucleus (Fig. 4k, l). Together, these data confirmed that the iron-triggered *LncRIM*-NF2 feedback loop hyperactivated YAP to promote cell proliferation (Fig. S5a, b).

### *LncRIM*-YAP axis-mediated iron metabolism promotes tumor progression

We next examined the role played by the *LncRIM*-YAP iron metabolism axis in tumorigenesis in vivo. Consistent with the cell line experiments (Fig. S1g), knocking down *LncRIM* suppressed both the size and weight of xenograft tumors (Fig. 5a–c) and significantly reduced cell proliferation, as indicated by the decreased expression of Ki67 and YAP (Fig. 5d). Moreover, knocking down *LncRIM* impaired angiogenesis, as indicated by reduced staining intensity of CD31 (an endothelial cell marker) and reduced cellular iron content, as indicated by an enhanced DAB iron-staining assay (Fig. 5d, e). Moreover, immunoblotting and immunohistochemical (IHC) staining showed decreased expression of both DMT1 and TFR1 in *LncRIM*-silenced tumors, with increased phosphorylation levels of LATS1 and YAP and decreased expression of YAP downstream targets (Figs. 5f, g and S5c, d). These results showed a positive correlation between *LncRIM*-mediated tumor progression and cellular iron metabolism.

Previous studies have shown that DFO combination therapy can inhibit cancer cell proliferation by reducing iron levels[53–55]. We therefore treated nude mice via an intraperitoneal injection of DFO at the indicated concentrations. As shown in Fig. S5e–h, DFO treatment significantly suppressed tumor growth and the cellular iron level compared to those in the control group, while overexpression of *LncRIM* partially restored tumor growth rate and the cellular iron level, as indicated by Ki67 and YAP levels and enhanced DAB iron-staining assay. Additionally, DFO treatment significantly decreased the *LncRIM*-mediated upregulation of YAP target genes (Fig. S5i).

To further validate the association between *LncRIM*-mediated cellular iron metabolism and tumor growth, we constructed *LncRIM* overexpressing, DMT1 and TFR1 double-knockdown under *LncRIM* overexpressed, cell lines, and each line was subsequently injected orthotopically into nude mice. As shown in Figs. 5h–k and S5j, knocking down both DMT1 and TFR1 led to a significant decrease of *LncRIM*-mediated xenograft tumor growth, and this decrease was accompanied by a decreased cellular iron level and reduced Ki67 and YAP expression (Fig. 5j, k). Moreover, a colony formation assay showed the same results (Fig. S5k, l). Collectively, these results demonstrated to some extent the idea that the *LncRIM*-NF2 axis promotes cell proliferation and breast cancer growth by upregulating cellular iron metabolism.

### High *LncRIM* expression correlates with poor clinical outcomes for breast cancer patients

Since *LncRIM* functions closely with YAP to promote iron metabolism reprogramming and tumor growth, they may be pathologically involved in breast cancer development. To test this hypothesis, we examined the expression of *LncRIM* in a cohort of breast cancer tissues by using RT–qPCR and subsequently categorized these data into *LncRIM*-low and *LncRIM*-high groups by comparing the *LncRIM* expression level to the respective median. We further detected its correlation with proliferation, angiogenesis, and iron metabolism by performing IHC assays. As shown in Fig. 6a, b, high expression of *LncRIM* was positively correlated with Ki67 and CD31, which are markers of proliferation and angiogenesis, respectively. In addition, high *LncRIM* expression was positively correlated with YAP, DMT1, and TFR1 expression as well as an increased cellular iron level in breast cancer (Fig. 6a). More importantly, double staining for iron with Perl's blue and for CD45, an immune cell marker, revealed that *LncRIM*-mediated changes in iron levels were mostly located in breast cancer cells, not in other cells (Fig. 6a).

In addition, the expression of *LncRIM* was positively correlated with YAP target genes, including *CTGF*, *CYR61*, *DMT1*, and *TFR1*, in breast cancer patient samples (Fig. 6c, d). Notably, the expression of DMT1 and TFR1 in breast cancer tissues was significantly higher than that in paired control samples (Fig. 6e, f), and high expression of DMT1 and TFR1 was correlated with poor survival in breast cancer patients in an independent cohort (Fig. 6g, h). Together, these data suggested the possibility of the *LncRIM*-NF2-DMT1/TFR1 axis being a therapeutic target in the clinical treatment of breast cancer (Fig. 6i).

## Discussion

Recent studies have clarified that multiple lncRNAs are involved in various physiological processes and human diseases, including cancer[3,7,29,30,56]. In addition, considering their subcellular location, lncRNAs are involved in different molecular mechanisms[57]. Previous studies, including our previous work, have suggested that some lncRNAs cause disease by disrupting a metabolic process or signaling transduction[3,29,30,36], showing evidence contributing to an in-depth understanding of cancer metabolism. In this study, we discovered a

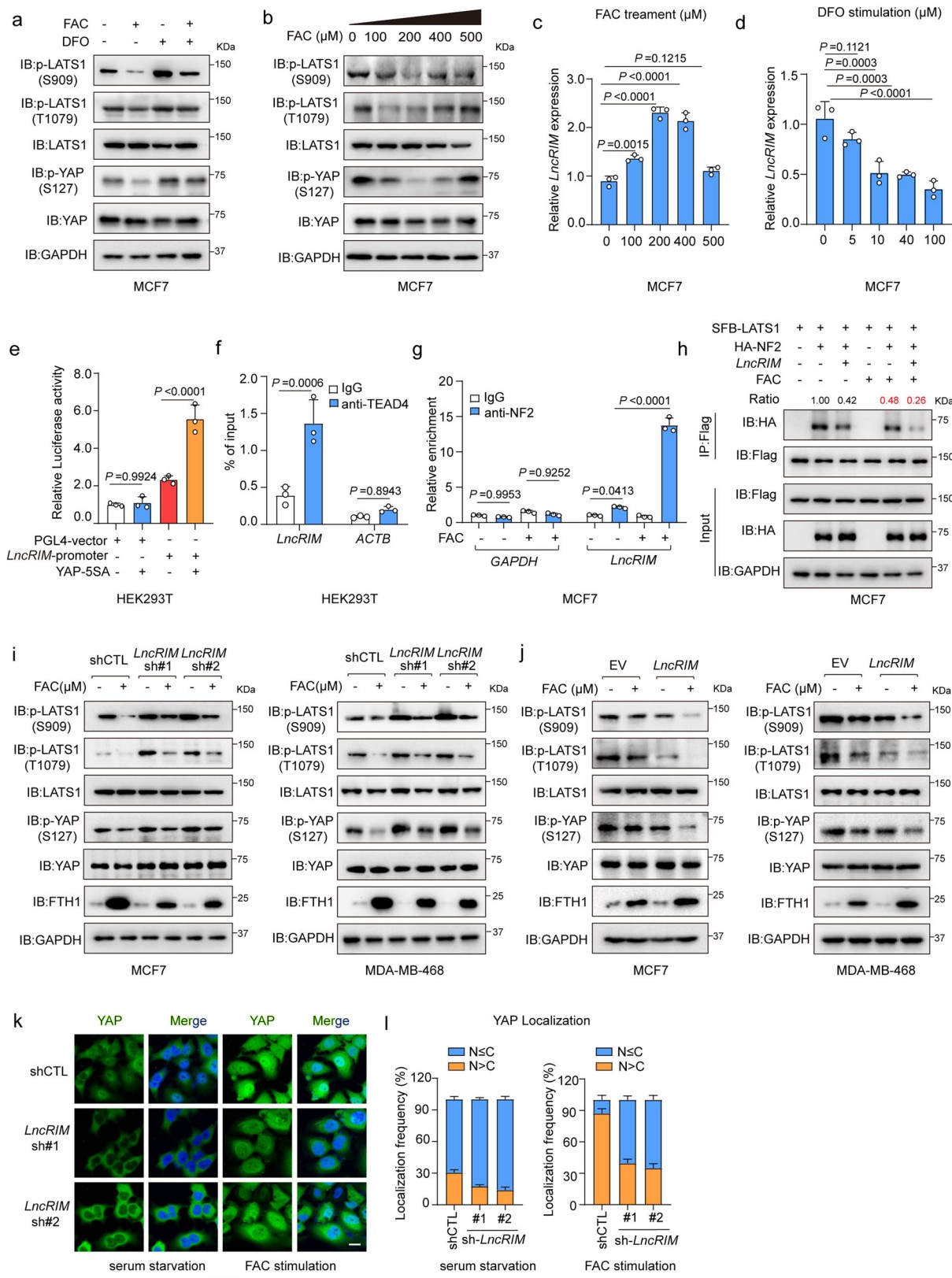

novel mechanism of lncRNA-mediated iron metabolism reprogramming in breast cancer initiation and progression, which further increased our knowledge of the function of lncRNAs in cancer metabolic processes. The dysregulation of the Hippo pathway is closely associated with cancer development[17], suggesting that this pathway shows potential to be a target in cancer therapy. Extensive studies have

demonstrated that the Hippo pathway responds to many upstream metabolic signals[58]. Notably, Han et al.[59] reported that the Hippo pathway plays an important role in heavy metal homeostasis by phosphorylating MTF1. Herein, we favored one model in which iron-triggered *LncRIM* wired up the Hippo pathway, which, at least in part, exerted an oncogenic effect via the regulation of cellular iron levels.

**Fig. 4 | The iron-triggered *LncRIM*-NF2 feedback loop hyperactivates YAP.**
**a** Immunoblot was performed to examine the level of p-YAP (S127) and p-LATS1(S909, T1079). Serum-starved MCF-7 cells were treated with FAC (200 μM) or DFO (100 μM) for 24 h. **b** Serum-starved MFC7 cells were stimulated with different concentrations of FAC for 24 h. Immunoblot was performed to detect the level of p-YAP (S127) and p-LATS1(S909, T1079). **c** RT−qPCR detection of the *LncRIM* expression of MCF-7 cells stimulated with different concentration of FAC for 24 h. (mean ± SD, *n* = 3, one-way ANOVA analysis). **d** The expression of *LncRIM* in MCF-7 cells stimulated with different concentration of DFO for 24 h was analyzed with RT-qPCR. (mean ± SD, *n* = 3, one-way ANOVA analysis). **e** Luciferase reporter assay was performed of HEK-293T cells with overexpression of YAP-5SA and *LncRIM* promoter. (mean ± SD, *n* = 3, one-way ANOVA analysis). **f** YAP/TEAD4 directly regulates the transcription of *LncRIM*. CHIP-qPCR assay was performed by using IgG and TEAD4 antibodies. (mean ± SD, *n* = 3, two-sided Student's *t* test). **g** RIP and RT−qPCR assays were performed to assess the interaction between *LncRIM* and NF2 of MCF-7 cells with or without FAC (200 μM) stimulation for 24 h. NF2 was immunoprecipitated by NF2 antibody and Protein A/G beads. IgG was used as the negative control. (mean ± SD, *n* = 3, One-way ANOVA analysis).(**h**) SFB-LATS1, HA-NF2 and *LncRIM* were co-transfected into MCF-7 cells with or without FAC (200 μM) stimulation for 24 h. SFB-LATS1 was immunoprecipitated by Flag beads. **i, j** Serum-starved control, *LncRIM* knockdown (**i**) or *LncRIM* overexpressed (**j**) MCF-7 and MDA-MB-468 cells were treated with FAC (200 μM) for 24 h, and the levels of p-LATS1(S909, T1079), p-YAP (S127) and FTH1 were detected by immunoblot. **k, l** Immunofluorescence staining of YAP in control and *LncRIM* knocked down serum-starved MCF-7 cells with or without FAC (200 μM) stimulation for 24 h (**k**). Scale bar, 10 μm. YAP localization in cells from three randomly selected fields of view was quantified (**l**). (mean ± SD, *n* = 3).

Importantly, we validated that *LncRIM*−NF2 axis mediated iron status in turn lead to YAP hyperactivity. This study provides functional evidence linking lncRNAs and the Hippo pathway to cellular iron metabolism and tumorigenesis. Interestingly, a previous study by Yuan et al. found that depression of lncRNA *MAYA* reduced iron levels by activating YAP in non-alcoholic fatty liver disease (NAFLD)[60], contradicting our finding indicating that YAP promoted iron overload in breast cancer cells. These results indicated that YAP and iron may be interconnected in different ways depending on the disease model and upstream molecules.

Iron has been reported to play a vital role in tumor initiation and progression[53,61]. Specifically, in breast cancer, the expression of iron-related genes, including TFR1, HEPH, and FPN, is altered, and this aberrant expression is associated with patient clinical outcomes[11]. However, the specific mechanism by which iron overload promotes breast cancer growth is largely unknown. Wang et al[53] found that the H3K9 methyltransferase G9a stimulated breast cancer development by repressing HEPH expression and increasing the cellular iron content. In the present study, we showed that the *LncRIM*-NF2 axis affected cellular iron metabolism via the downstream molecules DMT1 and TFR1, which are vital iron uptake-related proteins. Notably, for the first time, we also showed that DMT1 and TFR1 are both targets of YAP, clarifying a novel mechanism through which the *LncRIM*-NF2-DMT1/TFR1 axis ultimately promotes tumor progression by inducing changes in cellular iron levels.

In recent years, many studies have demonstrated a connection between cancer cell iron metabolism and the TME, showing that the relationship shapes the immune landscapes around cancer cells, especially macrophages[62]. On the one hand, M2 macrophages residing in the TME manifest "iron-donor" phenotype with high expression of ferroportin, and can act as one important iron source to promote cancer cell proliferation in iron-dependent manner[62].Moreover, LCN2 derived from the TAM enhances cancer cells uptake of iron in TF-TFR1-independent manner and stimulate cancer cell growth[34]. On the other hand, studies have shown that tumor cells can also in turn directly contribute to macrophages polarization[63]. And iron overload in multiple cancer cells favors M2 macrophage polarization in the TME[64]. Besides, breast cancer cells in turn induce pro-tumor associated M2 macrophages to acquire an iron-release phenotype[65]. Interestingly, in this study we found that overexpression of *LncRIM* slightly promoted M2 macrophage polarization, while knocking down both TFR1 and DMT1 significantly decreased the number of M2 macrophages and increased the number of M1 macrophages (Fig. S5m), providing evidence for a potential link to lncRNA-mediated iron metabolism of tumor cells and the macrophage polarization and functional abnormalities in the TME, but the underling mechanism is still unknown.

Previous studies have shown that cellular iron homeostasis is coordinately regulated by the IRP/IRE system, including IRP1 (known as ACO1) and IRP2 (also known as IREB2)[43,44,66]. In addition, IRP2 but not

IRP1 has been shown to regulate iron uptake in breast cancer[45]. In this study, we examined the relationship between *LncRIM*-NF2 and the IRP2/IRE from different perspectives. First, *LncRIM*-YAP still partially enhanced the expression of DMT1 and TFR1 and increased cellular iron level after IRP2 knockdown. Second, the effects of *LncRIM* knockdown were comparable to those of IRP2 knockdown: the cellular iron level was decreased (42% for IRP2 vs. 36% for *LncRIM*), the expression of DMT1 was decreased (54% for IRP2 vs. 52% for *LncRIM*), and the expression of TFR1 was decreased (51% for IRP2 vs. 59% for *LncRIM*). Moreover, the IRP2/IRE system established a feedback loop to regulate the stabilization of mRNA of both TFR1 and DMT1[10], while the *LncRIM*-NF2 axis created a feedforward mechanism to modulate the transcription of TFR1 and DMT1. Additionally, both nude mouse model and human tissue samples were analyzed, and the results verified the important physiological relevance between the *LncRIM*−Hippo axis and iron metabolism in vivo. All of these data illustrated that the *LncRIM*−NF2 axis exerts a mediating biological effect by regulating cellular iron metabolism. Considering that the IRP2 was typically expressed excess, and the limitation of IRP2 knockdown in this study, the effect of *LncRIM*−YAP axis in promoting the expression of DMT1 and TFR1, as well as that on the cellular iron level, was weaker than that in control cells. Thus, these results also suggested that the effect of *LncRIM*−YAP is perhaps to a large extent independent of IRP2, but not totally, and more research is required to understand the relationship between *LncRIM*−Hippo and IRP/IRE-mediated regulation in iron metabolism.

In conclusion, our study reveals a novel iron metabolism-related mechanism in which *LncRIM* directly binds NF2 to trigger the activation of YAP and then promotes the expression of DMT1 and TFR1, which ultimately increases the cellular iron level and promotes cancer cell proliferation. In addition, we demonstrated that the *LncRIM*−Hippo axis acts in an IRP2-independent manner and causes a biological effect similar to that of IRP2, including increased expression of DMT1 and TFR1 and the cellular iron level. Interestingly, we verified an iron-triggered *LncRIM*−NF2 feedback loop, which in turn hyperactivates YAP. Therefore, compounds that selectively target the iron-dependent *LncRIM*−Hippo axis may show potential for use in breast cancer therapy.

## Methods
### Cell lines
The human breast cancer cell lines MDA-MB-468 (HTB-132; RRID: CVCL_0419), MCF7 (HTB-22; RRID: CVCL_0031), MDA-MB-453 (HTB-131; RRID:CVCL_0418), MDA-MB-231(CRM-HTB-26; RRID: CVCL_0062), BT549(HTB-122, RRID: CVCL1092), T47D (CRL-2865, RRID: CVCL_0553), the human epithelial cell MCF10A (CRL-10317, RRID: CVCL_0598), and the human embryonic kidney cell line HEK293T (CRL-3216; RRID: CVCL_0063) were purchased from National Collection of Authenticated Cell Cultures (China) and characterized by the Cell Line Core Facility (MD Anderson Cancer

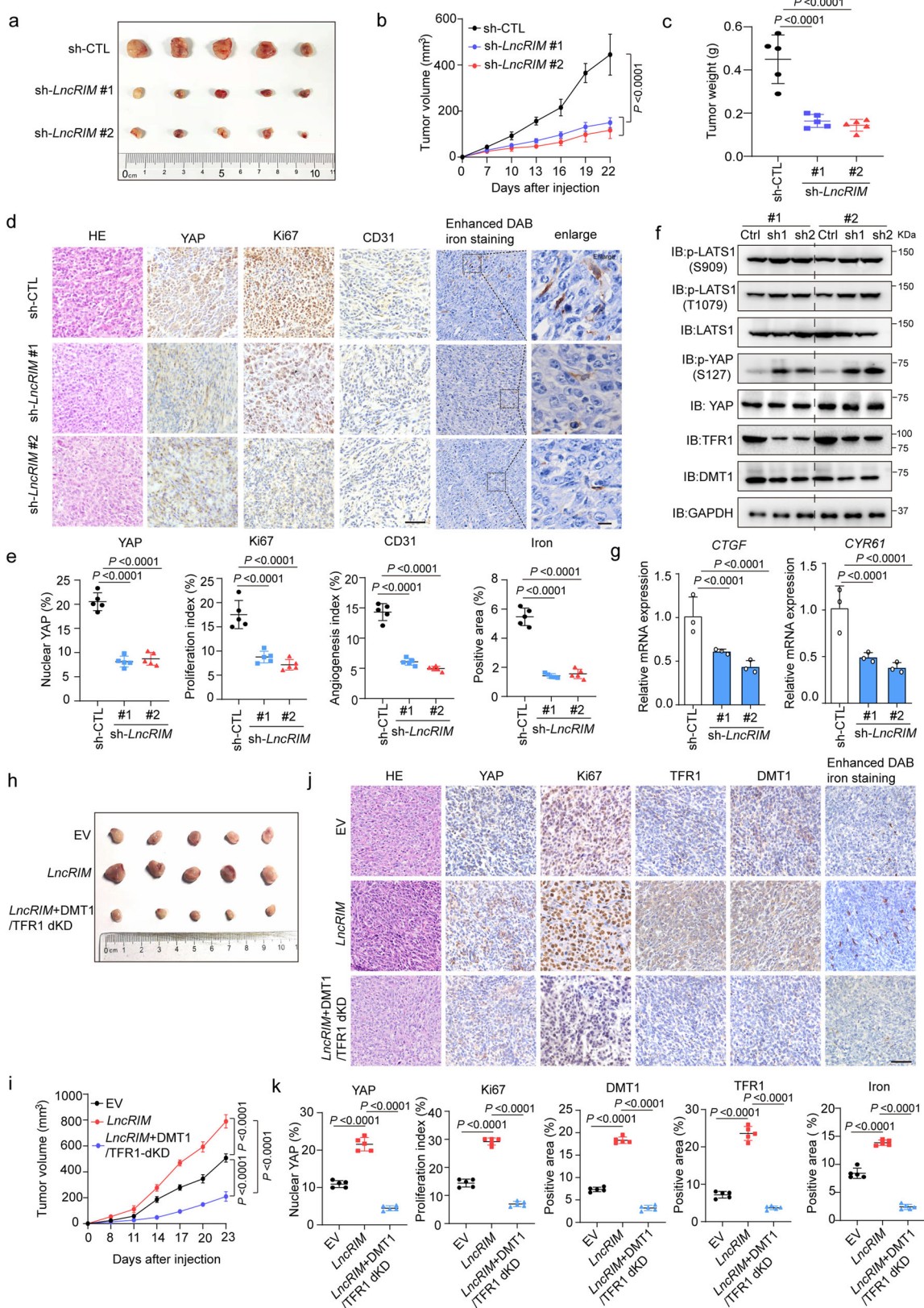

Center). These cell lines were maintained in Dulbecco's modified essential medium (DMEM) supplemented with 10% fetal bovine serum at 37 °C in 5% $CO_2$ (v/v). All cells were negatively tested for mycoplasma contamination and authenticated based on short tandem repeat fingerprinting before use.

**Tissue samples**

A total of 98 breast cancer tissues and matched paired carcinoma tissues (>5 cm away from the tumor) were obtained from patients who underwent surgery at the First People's Hospital of Huzhou (Huzhou, China), and were histologically diagnosed with breast cancer. The

**Fig. 5 | *LncRIM*-YAP axis-mediated iron metabolism promotes tumor progression. a** Xenograft mouse model using control or *LncRIM* knockdown MDA-MB-468 cells. In vivo generated tumors are shown. **b**, **c** Analysis of tumor volume (**b**) and weight (**c**) in xenograft mouse model are shown. The data are presented as the mean ± SD of *n* = 5 mice per group, Two-way/One-way ANOVA analysis. **d**, **e** Representative IHC staining and enhanced DAB iron staining of randomly selected tumors from mice subcutaneously injected with the indicated stably transduced MDA-MB-468 cells (**d**). Scale bar, 100 µm. The relative intensities were quantified by ImageJ (**e**). The data are presented as the mean ± SD of *n* = 5 mice per group, One-way ANOVA analysis. **f** Immunoblot detection of YAP, LATS1, DMT1 and TFR1 expression in randomly selected xenograft tumors. **g** The YAP target genes expression in the indicated subcutaneous xenograft tumors was examined by RT-qPCR. (mean ± SD, *n* = 3, one-way ANOVA analysis). **h**, **i** Nude mice were injected with control, *LncRIM* overexpressed, or double knockdown of DMT1/TFR1 with overexpression of *LncRIM* MDA-MB-468 cell lines. In vivo generated tumors are shown (**h**). The tumor volumes were assessed (**i**). The data are presented as the mean ± SD of *n* = 5 mice per group, two-way ANOVA analysis. **j**, **k** Representative IHC staining and enhanced DAB iron staining of randomly selected tumors from mice subcutaneously injected with the indicated stably transduced MDA-MB-468 cells (**j**). Scale bar, 100 µm. The relative intensities were quantified by ImageJ (**k**). The data are presented as the mean ± SD of *n* = 5 mice per group, one-way ANOVA analysis.

study protocol was approved by the Institutional Review Board of The First People's Hospital of Huzhou. All procedures were performed with the approval of the internal review and ethics boards of The First People's Hospital of Huzhou. Participants were recruited from The First People's Hospital of Huzhou with no perceived bias, and all eligible participants were offered enrollment. All enrolled patients provided written informed consent prior to sample collection. Sixty-nine tissue samples with complete tissue form were used for IHC staining and iron staining. None of the patients were treated with adjuvant radiotherapy or chemotherapy before surgery. Detailed clinical information is listed in Supplementary Tables 1–4.

### Mice
All animal experiments were performed by a protocol approved by the Institutional Animal Care and Use Committee (IACUC), and the mice had a maximum tumor size/burden of less than 15 mm. The care of experimental animals was by appropriate guidelines and approved by the Laboratory Animal Committee of Zhejiang University (ZJU20210028). Female nude mice (BALB/c strain; 4–5 weeks old) were purchased from the Shanghai Laboratory Animals Center and used in the xenograft mouse model assay. Animals were housed in a pathogen-free barrier environment (approximately 20 °C with 40% humidity and a 12-h dark/light cycle) throughout the study. Mice were fed a normal chow diet and water with ad libitum feeding. Control and experimental animals were bred separately.

### Antibodies
Specific antibodies were purchased from the following commercial sources for immunoprecipitation and immunoblotting experiments: anti-YAP (#14074S, 1:1000 for IB), anti-p-YAP (Ser127) (#13008S, 1:1000 for IB), anti-p-LATS1 (Thr1079) (#8654S, 1:1000 for IB), anti-p-LATS1 (Ser909) (#9157S, 1:1000 for IB), anti-LATS1 (#3477S, 1:1000 for IB and 1:100 for IP), and anti-NF2 (#12888S, 1:1000 for IB), anti-Mob1 (#13730S, 1:1000 for IB), anti-Mst1 (#14946S, 1:1000 for IB) from Cell Signaling Technology; anti-YAP (13584-1-AP, 1:1000 for IB), anti-LATS1 (17049-1-AP, 1:1000 for IB), anti-TFR1 (10084-2-AP, 1:1000 for IB), and anti-DMT1 (20507-1-AP, 1:1000 for IB) from Proteintech; anti-IRP2 (sc-33682, 1:1000 for IB) from Santa Cruz Biotechnology; anti-TFR1 (A5865, 1:1000 for IB), anti-FTH1 (A19544, 1:1000 for IB) from ABclonal; and anti-GAPDH (M20050, 1:5000 for IB), anti-DYKDDDDK-tag (M20008, 1:5000 for IB), anti-GST-tag (M20007, 1:2000 for IB), anti-His-tag (M20001, 1:5000 for IB), and anti-HA-tag (M20003, 1:5000 for IB) from Abmart; anti-TEAD4 (ab197589,1:150 for ChIP) was purchased from abcam.

For immunofluorescence, an anti-YAP monoclonal antibody (13584-1-AP, 1:100) and anti-LATS1 (17049-1-AP, 1:100) were purchased from Proteintech; an anti-NF2 monoclonal antibody (#12888, 1:100) was purchased from Cell Signaling Technology; goat anti-rabbit IgG H&L (Alexa Fluor 488, ab150077, 1:400 for IF), and goat anti-rabbit IgG H&L (Alexa Fluor 647, ab150083,1:400 for IF) were purchased from abcam.

For IHC, anti-YAP (13584-1-AP, 1:200), anti-TFR1 (10084-2-AP, 1:200) and anti-DMT1 (20507-1-AP, 1:200) were purchased from Proteintech; anti-Ki67 (#9449, 1:400) was purchased from Cell Signaling Technology, anti-CD31 (A0378, 1:200) was purchased from ABclonal.

For flow cytometry, anti-CD11c-PE (#12-0114, 1:500), anti-CD206-APC (#17-2061, 1:500) were purchased from eBioscience; anti-F4/80-FITC (#123116, 1:500) was purchased from Biolegend.

### Cloning procedures
Full-length *LncRIM*, IRP2, DMT1and TFR1 were cloned from HEK293T cDNA by PCR. The NF2 template was gifted by the laboratory of J.-H. Han. All eukaryotic over-expressed genes (WT and mutants) were cloned into an SFB lentiviral (S-protein, Flag-tag, and SBP-tag fused) vector or pcDNA3.1-Flag/HA empty vectors using the ClonExpress II One Step Cloning Kit (Vazyme). All of the shRNA in this study were cloned into the pLKO.1-Puro vector by using T4 ligase (Promega). DMT1 and *LncRIM* promoter were cloned into pGL4 Luciferase Reporter vector (Promega). *LncRIM* and its deletion mutants were cloned into pGEM-T easy (Promega) for in vitro-transcription. All deletion or truncated mutations were generated by PCR overlapping. Bacterial expression vectors for MBP–His-tagged NF2, GST-tagged NF2 (WT and mutants) and LATS1-NT were constructed by cloning into pET-28a vector or pGEX-4T1 vector.

### Short interfering RNA, short hairpin RNA, and RNA interference
Commercially available Lincode SMARTpool siRNAs targeting *LncRIM*, *Loc645249*, *Loc653160*, *DNAJB8-AS1*, *PACRG-AS1*, and *LINC00467* were purchased from Dharmacon. All short hairpin RNA (shRNA) sequences were designed according to https://portals.broadinstitute.org/gpp/public/. All shRNA sequences were cloned into the pLKO.1-Puro vector, and two shRNAs with the efficient knockdown capability were used in subsequent studies. All of the sequences are listed in Supplementary Table 5.

### Data analysis
Gene set enrichment analysis (GSEA) using the C6 canonical pathways Broad MsigDB database was downloaded from (GSE38369 - GEO DataSets - NCBI (nih.gov)) on gene expression data. ClusterProfiler (R package (4.1.4)) was utilized to perform.

The ChIP-seq analysis of YAP/TEAD was downloaded from the NCBI GEO database (GSE107013 - GEO DataSets - NCBI (nih.gov). Bowtie2 (v2.3.5) was used to map ChIP-seq raw reads to the GRCh38 human reference genome. Then SAMtools v1.9 and bamCoverage program in deeptools were used to remove duplicate reads and generate normalized signals. Subsequently, MACS2 program (v2.2.4) was used to call peaks of ChIP-seq data with the corresponding input data as control. RPKM was calculated to quantify each peak. Only peaks with log10*P* value >6 and fold-enrichment >0.585 were considered in the downstream analyses. Integrative Genomics Viewer (IGV) was used to visualize the peaks.

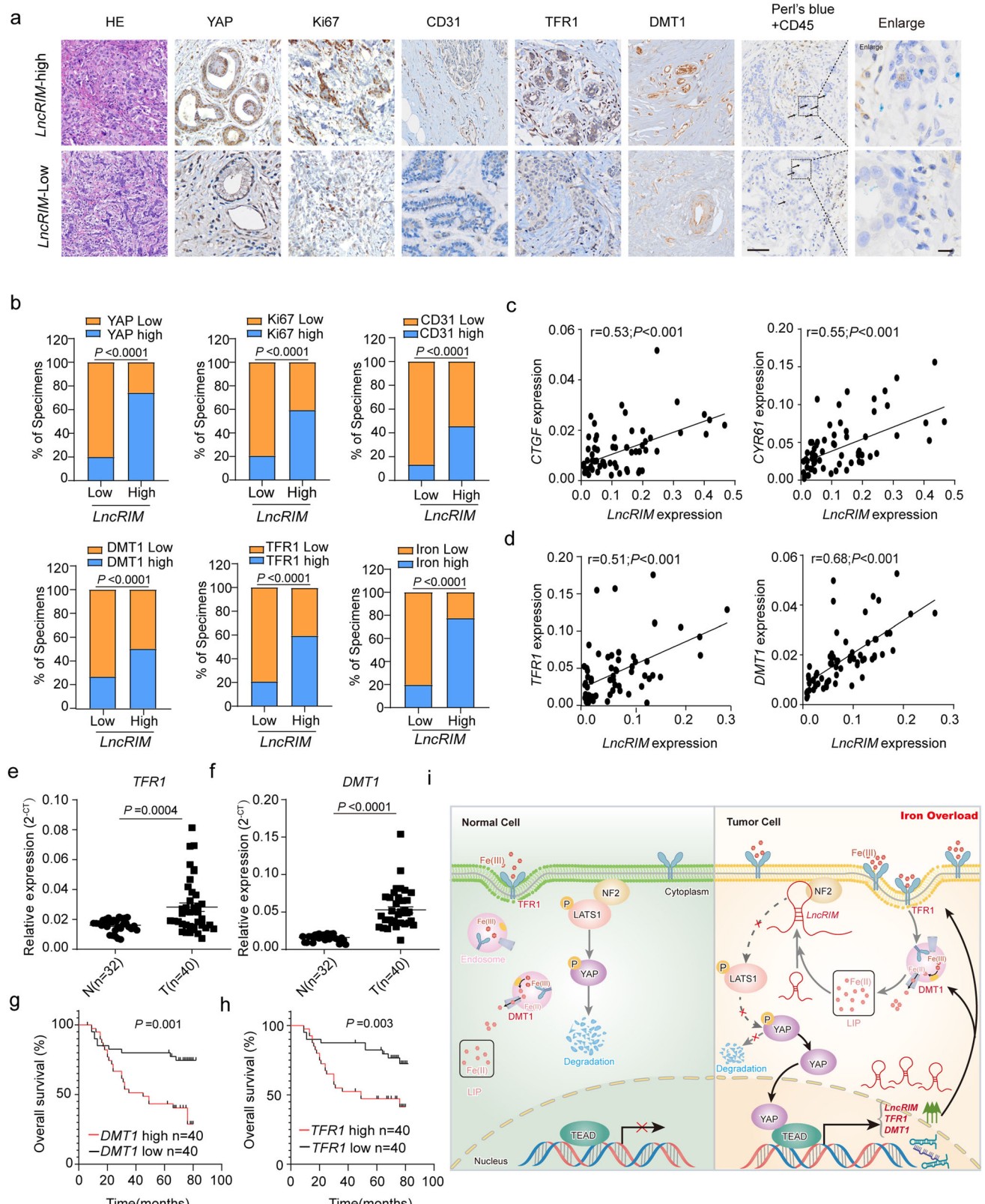

YAP-related sequencing data were download from NCBI SRA database (https://www.ncbi.nlm.nih.gov/sra?term=SRP125142). After trimming by trim_galore (v.0.6.6), clean data were mapped by STAR (v.2.7.10b). Then the peaks were called via macs2 (v.2.2.7.1), and annotated by ChIPseeker (v.1.34.1). The motif was analyzed and visualized by MEME webtools (https://meme-suite.org/meme/tools/meme).

## Protein recombination and purification

The recombinant proteins GST-NF2 (WT and mutants), His-MBP-NF2, and GST-LATS1-NT were expressed in *E. coli* strain BL21-CodonPlusÒ (DE3) RIPL (Agilent Technologies), and IPTG was used at a concentration of 0.1 mM. After sonic disruption, the samples were purified using GST magnetic beads (Sangon Biotech), or His-tagged beads (BBl). The concentration and purity of recombinant proteins were

**Fig. 6 | High *LncRIM* expression correlates with poor clinical outcomes for breast cancer patients. a** The expression of *LncRIM* was positively correlated with the expression of YAP, Ki67, CD31, DMT1, and TFR1 in human breast cancer tissue (*n* = 98 including *LncRIM*-high and *LncRIM*-low subsets). Double staining of iron and CD45 in breast cancer cells (arrows). Scale bar, 100 μm. **b** Percentages of specimens with low and high *LncRIM* expression relative to the levels of YAP, Ki67, CD31, DMT1, TFR1, and iron (two-sided $\chi^2$ test). **c** The expression of *LncRIM* was positively correlated with *CTGF* and *CYR61* as determined by two-sided chi-square test; *R*, correlation coefficient (*n* = 80 tumor patient samples). **d** The expression of *LncRIM* was positively correlated with *DMT1* and *TFR1* as determined by two-sided chi-square test; *R*, correlation coefficient (*n* = 80 tumor patient samples). **e, f** RT–qPCR detection of the *DMT1* and *TFR1* expression in tumor tissues (*n* = 40) and paired control tissues (*n* = 32). The horizontal black lines represent the median values. two-sided Student's *t* test. **g, h** Recurrence-free survival analysis of the *DMT1* status (**g**) and *TFR1* status (**h**) of breast cancer patients (*n* = 80, Kaplan–Meier analysis with the Gehan–Breslow test). **i** Graphic illustration of the *LncRIM*-NF2-DMT1/TFR1 axis in cellular iron metabolism.

measured by SDS–PAGE and Coomassie staining with the standard BSA control.

## Real-time intracellular iron

The amount of calcein-chelated iron within various stably transfected cells was assayed according to the protocol (YEASEN, 40719ES50). MCF7 and MDA-MB-468 treated cells were incubated with 0.15 μM Calcein-AM or 30 min at 37 °C in PBS. After calcein loading, the cells were washed with PBS three times, resuspended in PBS, and then plated in 96-cell plates. The fluorescence was monitored ($\lambda_{ex}$ 488 nm; $\lambda_{em}$ 518 nm) by fluorescence microscopy or the fluorescence value was acquired by using Nanodrop (Bio-Rad).

## Cell lysis, immunoprecipitation, and immunoblotting

Cells were harvested in PBS and homogenized in NETN buffer (25 mM Tris-HCl (pH 8.0), 100 mM NaCl, 1 mM EDTA, and 0.5 mM dithiothreitol (DTT) containing a protease inhibitor cocktail, phosphatase inhibitor cocktail, and Panobinostat. Lysates were cleared by centrifugation at 13,000 × *g* for 15 min at 4 °C. Supernatants were used for IB or IP with the indicated antibodies. For IP, the required primary antibody and the control IgG were added separately to the prepared lysates. After incubation at 4 °C for 5 h with gentle rotation, 10 μl of protein A/G magnetic beads (Pierce, 88803) was added to each lysate, followed by incubation for another 2 h at 4 °C with rotation. The protein-captured beads were washed with NETN buffer 3× for 5 min each at 4 °C with rotation. Then, the beads were eluted with 30 μl of 1× SDS loading buffer, and the eluted proteins or protein complexes were detected by IB. The blotting signals were detected using Clarity Western ECL substrate (Bio-Rad). For tagged-protein IP, the primary antibody and the protein A/G beads were replaced with FLAG-M2 magnetic beads (Sigma, M8823), S-protein agarose beads (Millipore, 69704), or HA magnetic beads (Pierce, 88837). Blot images were obtained using Image Lab v4.1 software (Bio-Rad).

## RNA immunoprecipitation, RNA extraction, and RT–qPCR

The enrichment of the interested protein process was mostly similar to the protein IP indicated in "Cell lysis, immunoprecipitation and immunoblotting" with the following modifications: all processes were RNase-free; additional Ribolock RNase Inhibitor (Invitrogen) was required; the lysis buffer was transferred to polysome buffer; and the wash buffer was transferred to NT2 buffer. Then, TRIzol reagent (Invitrogen) was used to extract the associated RNAs according to the manufacturer's instructions. Reverse transcription was performed using the iScript cDNA synthesis kit (Bio-Rad), and the abundance of target RNAs was detected by iTaqTM Universal SYBR Green Supermix qPCR kit (Bio-Rad) according to the manufacturer's instructions. GAPDH expression was assessed concurrently on the same plate as the mRNAs for normalization. All data were analyzed with GraphPad Prism 8 or GraphPad Prism 7.

## Immunofluorescence

Cells cultured in chamber slides were fixed with 3.7% formaldehyde in PBS for 10 min at RT, followed by permeabilization with 0.5% Triton X-100 in PBS for 10 min. The cells were then blocked with 5% BSA for 30 min at RT, incubated with the indicated primary antibodies at 4 °C

overnight, and incubated with anti-rabbit IgG (H + L) and F (ab') 2 fragments (Alexa Fluor 647 or 488 conjugate) from abcam for 1 h at RT. Nuclei were counterstained with DAPI for 10 min (Sigma–Aldrich, D9542) for observation. IF images were acquired on an FV3000 confocal microscope (Olympus) or Super resolution Confocal Laser scanning microscope TCS SP8 STED (Leica). For each channel, all images were acquired with the same settings. Fluorescence images were obtained using FV31S-SW Viewer (v2.3.1), FV31S-DT (v2.3.1) software (Olympus) or Leica Application Suite X (v3.3.0.16799) (Leica).

## Immunohistochemistry and iron staining

For IHC staining, the paraffin-embedded tissues were deparaffinized, rehydrated, and subjected to antigen retrieval. After incubation with the primary and secondary antibodies (listed in the Antibodies section), the slides were dehydrated and stabilized with a mounting medium, and the images were acquired with an Olympus DP72 microscope. The quantification of IHC staining density was performed by ImageJ (Fiji version 2.3.0) software and calculated based on the average staining intensity and the percentage of positively stained cells. The total protein expression scores were calculated from both the percentage of positive cells and the staining intensity. High and low protein expression was defined using the mean score of all samples as a cutoff point. Spearman rank correlation was used for statistical analyses of the correlation between each marker and clinical stage.

Non-heme iron staining was performed using a standard Perl's Prussian Blue staining (38016SS7, Leica). Briefly, the paraffin-embedded tissues were deparaffinized, rehydrated, and then incubated with Perlsstain for 15–20 min at room temperature. Following this, the slides were washed totally with distilled water for 5 min, and stained the nuclear with eosin reagent for 1 min. After that, the slides were washed, dehydrated and stabilized for observation. Photomicrographs were taken with a Nikon Eclipse E400 microscope (Nikon, Melville, NY). The tissue iron staining can also carry out according Enhanced DAB iron staining kit (Solarbio, #G1428).

## Membrane fractionation purification

All steps of the cytosolic and solubilized particulate membrane fractions were performed at 4 °C[53]. Cells were homogenized with a tissue grinder in buffer A (0.025 M Tris-HCl, pH 7.4, 0.025 M NaCl, and a protease inhibitor cocktail) and centrifuged at 16,000 × *g* for 15 min. The cytosolic fractions were obtained by re-centrifuging the supernatants at 10,000 × *g* for 1 h. The pellets were resuspended in buffer B (buffer A with 0.25% [v/v] Tween-20), sonicated at 25 Watts for 1 min in an ice water slurry with 15 s of chilling in between and recentrifuged at 16,000 × *g* for 30 min. These supernatants were termed the solubilized membrane fraction.

## RNA pull-down assay and dot blot assay

All steps were performed under RNase-free conditions. In vitro, biotin-labeled *LncRIM* was transcribed with SP6/T7 RNA polymerase by using a TranscriptAid SP6/T7 High Yield Transcription kit (Thermo Fisher Scientific, AMB13345) and Biotin RNA Labeling Mix (Roche). Cell lysate was prepared using polysome buffer (25 mM Tris-HCl (pH 7.5), 150 mM KCl, 0.5 mM DTT and 0.5% NP-40) with complete protease inhibitor cocktail (Roche) and Ribolock RNase Inhibitor (Invitrogen). M-280

Streptavidin Dynabeads (Invitrogen) were prepared according to the manufacturer's instructions. Then cell lysates, and 3-7 µg Biotin-labeled RNA in RNA capture buffer (20 mM Tris-HCl (pH 7.5), 1 M NaCl, and 1 mM EDTA) were separately mixed with Streptavidin Dynabeads (Invitrogen) at room temperature (RT) for 1 h. Then, the cell lysates were added to RNA-captured beads, and the mixtures were rotated for 2–4 h at 4 °C. Then, beads were washed with NT2 buffer three times, NT2 high-salt buffer (NT2 buffer with 500 mM NaCl) twice, and PBS once for 5 min at 4 °C. Finally, 50 µl of 2× SDS loading buffer was added at 100 °C for 10 min. For western blot detection, 0.5–1 mg of cell lysate and 1-3 µg of biotin RNA were sufficient. For the purified protein RNA pull-down assay, 1–2 µg of purified protein and 1–3 µg of biotin RNA were sufficient.

For the dot blot assay, the recombinant proteins were incubated with transcribed biotinylated *LncRIM* on a PVDF membrane (GVS) for 30 min at RT, followed by ultraviolet (UV) crosslinking. Then, the mixtures were blocked at RT for 2–3 h, after which the hybridized membranes were incubated with various primary antibodies at room temperature for 3 h and then with secondary antibodies for another 1 h. The protein-bound RNA sequences were visualized by the detection of streptavidin-HRP signals.

### Cell viability and colony formation assay

For cell viability, equal numbers of MCF7 and MDA-MB-468 stable cells were plated onto 96-well plates. After 12 h, each cell line was treated with 100 µM FAC or PBS as control. Cell proliferation assays were carried out with an MTS reagent (Promega, G1111). The absorbance was measured at a wavelength of 562 nm.

For colony formation assays, MCF7 and MDA-MB-468 stable cells were seeded in 6-well plates at densities of 800 cells/well and 1000 cells/well, respectively. The cells were cultured for 12–14 days until the colonies became visible. The cells were treated with 100 µM FAC every four days. Then, the colonies were fixed with 4% formaldehyde for 20 min at room temperature, followed by staining with 1% crystal violet.

### Flow cytometry

Cells were stained with propidium iodide (PI) and then evaluated for cell cycle by flow cytometry according to the manufacturer's protocol (YEASEN, 40301ES50,). Briefly, cells seeded at 6-wells were firstly collected and wash with cold PBS, then fixed with cold 75% ethanol at 4 °C overnight. After centrifugal and washing with PBS, pellets were suspended in 500 µl of binding buffer and incubated with 10 µl of PI solution and 5 µl RNaseA at 37 °C for 30 min.

For mouse macrophages, MDA-MB-468 cells ($3 \times 10^6$) were subsequently injected orthotopically into nude mice. After 3 weeks, mice were sacrificed, and the tumors were dissected. Tumor tissues were minced and excised into small pieces followed by incubation in DMEM containing 1 mg/ml collagenase IV (Sigma, C4-28-100MG) and $10^{-3}$ U/L DNase I (Invitrogen, EN0521) for 0.5-1 h. After lysed, single-cell suspensions were stained with fluorochrome-conjugated antibodies: anti-CD11c-PE (eBioscience, #12-0114), anti-F4/80-FITC (Biolegend, #123116), anti-CD206-APC (eBioscience, #17-2061) for 20 min at room temperature. After washing three times with PBS, the cells were analyzed using flow cytometry (Beckman Coulter Cytoflex) and data were analyzed by using CytExpert V2.3 and FlowJo X software version 7.6.4.

### Luciferase reporter assay and ChIP assay

The human *DMT1* promoter-reporter and *LncRIM* promoter-reporter were amplified from human genomic DNA using primers (sequences are listed in Supplementary Table 5). HEK293T cells or MCF7 cells seeded into 12-wells were transfected with luciferase reporter constructs and indicated plasmids. The Renilla plasmid was used as the transfection efficiency indicator to normalize firefly luciferase. After 48 h, whole-cell lysates were extracted, and luciferase activity was determined using the Dual-Luciferase Reporter Assay kit (Promega, E1910) and an illuminometer instrument. The fluorescence was determined by Nanodrop (Bio-Rad).

ChIP-qPCR assays were carried out according to the manufacturer's protocol (Merck Millipore, #17-371). Briefly, the cell lysates were crosslinked with 1% formaldehyde for 10 min, and glycine was added to a final concentration of 125 mM for 5 min. After washing with cold PBS, the cells were resuspended in SDS lysis buffer and sonicated for 10 min to shear DNA to an average fragment size of 300–500 bp. Then, the chromatin solution was precleared with 20 µl of ChIP-Grade protein G agarose beads (Santa Cruz Biotechnology, #LO219). The soluble fraction was collected, and the chromatin was incubated with ChIP grade TEAD4 antibody (abcam, ab58310) for 4 h, followed by the addition of protein A/G beads to the tube for another 1 h. The ChIP-enriched DNA was analyzed by quantitative PCR using the specific primers described in Supplementary Table 2. The enrichment of specific genomic regions was assessed relative to the input DNA followed by normalization to the respective control IgG values.

### Xenograft mouse model

All animal experiments were performed by the protocol approved by the Institutional Animal Care and Use Committee. Mice were housed in a barrier facility proactive for environmental enrichment and fed a normal chow diet and water ad libitum. Prepared tumor cells in 30 µl of sterile PBS were injected separately into the flanks of 4-5week-old female BALB/c nude mice. Tumor size was measured once every two days using a caliper, and tumor volume was calculated using the following standard formula: $0.54 \times L \times W^2$, where $L$ is the longest diameter and $W$ is the shortest diameter. Mice euthanized by cervical dislocation when they met the institutional euthanasia criteria for tumor size and overall health condition. The solid tumors were removed, photographed, and weighed.

### Statistics and reproducibility

All statistical results are reported as the mean ± SD of three or more independent biological replicates. Representative images for fluorescence staining, IHC staining, and immunoblot are shown. Each of these experiments was independently repeated three times. Relative quantities of gene expression levels were normalized to GAPDH, or U6. $P$ values <0.05 were considered to be statistically significant (variance is similar between the groups). Correlations were performed using the Pearson correlation test. The overall survival curves of patients were drawn by the Kaplan–Meier method. For every figure, statistical tests are justified as appropriate. Analyses and graphical presentation were performed using the GraphPad Prism 8.0 and 7.0 software. The experiments were not randomized. The investigators were not blinded to allocation during experiments and outcome assessment.

### Reporting summary

Further information on research design is available in the Nature Portfolio Reporting Summary linked to this article.

## Data availability

All data generated or analyzed during this study are included in the Supplementary Information files or with the hyperlink. The Gene set enrichment analysis (GSEA) data in this study was available in the Broad MsigDB database under accession code https://www.ncbi.nlm.nih.gov/gds/?term=GSE38369.The ChIP-seq analysis of YAP/TEAD in this study was available in the NCBI GEO database under accession code https://www.ncbi.nlm.nih.gov/gds/?term=GSE107013. The analysis of *LncRIM* expression in tumor in this study was acquired using TCGA public database from online web server (https://xenabrowser.net/datapages/?dataset=tcga_RSEM_gene_tpm&host=https%3A%2F%2Ftoil.xenahubs.net&removeHub=https%3A%2F%2Fxena.treehouse.gi.ucsc.edu%3A443). Source data are provided with this paper.

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

## Acknowledgements

We thank Professor F.-D. Wang (Zhejiang University) for the support and suggestions provided during this study. We thank J.-H. Han (Xiamen University) for gifting the NF2 template vectors. We thank Professor X.W. for gifting the breast cancer clinical samples. This work was supported in part by the National Science Fund for Distinguished Young Scholars (No. 32225014), "Lingyan" R&D Research and Development Project (No. 2023C03023), National Key R&D Program of China (No. 2021YFC2700903), National Natural Science Foundation of China (No. 81672791, No. 81872300, No. 82071567), and Zhejiang Provincial Natural Science Fund for Distinguished Young Scholars of China (No. LR18C060002).

## Author contributions

A.L. and W.W. conceived and designed the research; X.H. and X.F. performed most of the biochemical, molecular, and bioinformatics experiments with assistance from L.Q., X.W., L.J., L.-j.S., C.-y.S., S.L., J.-c.Y., Z.-z.Y., K.L., and J.-h.L.; X.W., X.H., and H.J. ascertained and processed the clinical specimens; L.Q. and Z.-z.Y. conducted the bioinformatic analysis; X.H., X.F., K.L., J.-h.L., and J.-c.Y. per-formed the xenograft experiments and IHC analyses; A.L., W.W., Y.X., J.S., F.W., J.L., Q.Y., and X.W. contributed to the discussion and data interpretation; A.L., W.W. and J.L. edited the manuscript; A.L. and W.W. initiated and supervised the project; A.L. and X.H. wrote the manuscript.

## Competing interests

The authors declare no competing interests.
