## [Peer Review File · Nature Communications]

LncRNA Modulates Hippo-YAP Signaling to Reprogram Iron MetabolismREVIEWER COMMENTS

Reviewer #1 (Remarks to the Author):

In their manuscript He et al. aim to investigate the possible correlation among long noncoding RNA, the Hippo-YAP pathway and iron metabolism, in a breast cancer context. Overall the work presented is extremely well thought and executed. It elegantly combines the molecular mechanism and the functional aspects of the role of lncRIM in promoting breast cancer progression through modulation of iron metabolism. Worth noting are the experiments that uncover the molecular interaction between lncRIM and NF2. Despite it, this reviewer strongly believes that such a nice work would certainly benefit from more robust *in vivo* experiment. Even that xenograft transplantation is universally accepted as a model, orthotopic injections result in a more physiological tumour environment. Thus, performing orthotopic transplantation, using cells depleted for lncRIM and/or overexpressing it, will allow taking into account also the role of tumour microenvironment. This is required for further evaluation in *Nature Communication*.

Specific comments listed below need to be fully addressed:

- 1) In Figure 1m the effect of FAC in rescuing cell growth is really minimal. Showing the statistical significance for all the columns would certainly make the data more robust.
- 2) In the section titled "lncRIM interacts with NF2 to inactivate LATS1 kinase" you claim that overexpression of lncRIM significantly reduced the association of NF2 and LATS1 both *in vivo* and *in vitro*, as shown in Fig 2e-f and Fig S2b. However no *in vivo* data are shown.
- 3) In Fig 3c the silencing of YAP performed with the construct #2, does not seem to work as nice as the #1. In fact, overexpression on lncRIM in those conditions brings back TFR1 protein levels. That should be taken into account in the text.
- 4) In Fig 3i-k data the authors only show data on DMT1. Data on TFR1 levels in the presence of active or inactive YAP mutant are never shown. Panels should be added to the figure.
- 5) Fig 3n would certainly benefit from an additional control, without the IRP2 KO, in order to assess the level of the rescue. The same for Fig S3g-h
- 6) In Fig 4b-c you state that the effect of FAC on LATS1 and YAP phosphorylation is dose dependent. However, from the images it is clear that higher concentrations of FAC bring the phosphorylation back. The text should be modified accordingly and an explanation should be provided.

Reviewer #2 (Remarks to the Author):

He, Qu et al. investigate the interconnection between lncRNAs, YAP signaling, and iron metabolism in breast cancer. They identify lncRIM, a lncRNA whose expression is stimulated by iron and is associated with poor survival in breast cancer patients. They also show that YAP signalling increases cellular iron levels, likely due to direct stimulation of TFRC and DMT1 transcription by YAP/TEAD. The authors then connect the two findings and describe a positive feedback loop whereby lncRIM itself enhances YAP signaling to further increase iron levels in the cell. From a mechanistic point of view, the authors identify and map an interaction surface between lncRIM and the tumor suppressor NF2. This interaction impedes the binding of NF2 to LATS1, resulting in downstream activation of YAP. Tumor transplantation experiments in mice and clinical data from cancer patients suggest that this YAP-iron-lncRIM feedback loop may contribute to promoting tumor growth by increasing cellular iron availability in cancer cells.

Understanding how exactly cancer signalling pathways and iron metabolism crosstalk is a very interesting topic, yet it remains poorly understood. He, Qu et al. make an important contribution to the field and their work will be of broad interest to scientists interested in cancer signalling and cancer metabolism. The authors present a tremendous amount of data, and their detailed characterisation of the mechanism through which lncRIM affects YAP signalling and iron is rather advanced. As in every study, some aspects require improvement.

The association between YAP/iron metabolism and tumor characteristics is largely based on correlations, without formal evidence of causality. This is normal when looking at clinical data. From

an experimental viewpoint, the utilization of an iron chelator/donor to show that lncRIM promotes cell/tumor growth by increasing iron levels via YAP is not convincing enough. As discussed below, DFO can inhibit tumor growth or colony formation regardless of lncRIM status. In vivo, DFO might also influence tumor development indirectly. Furthermore, adding an iron donor does not improve colony formation in lncRIM-KD cells. This shows that lncRIM can influence tumor growth independently of iron. More specific approaches would be better. For example, if the stimulation of iron via YAP would be due to upregulation of TFRC/DMT1, preventing TFRC/DMT1 upregulation would be predicted to antagonize the effect of lncRIM on cell/tumor growth. In the absence of a clear demonstration, the authors shall at least temper their statements throughout the manuscript.

The data based on Perl's staining without DAB enhancement is hard to judge given how weak the signal is (furthermore, localization of the signal is of importance but is not discussed). Similarly, it is sometimes hard to appreciate the immunoblot results from a quantitative perspective as the effects are in some cases rather subtle. The authors may also want to consult a statistician to inquire whether all their statistical analyses are optimal. Using a student t-test for small samples is generally not appropriate, so is performing multiple t-tests to compare more than two groups.

It would be interesting to discuss the data in relation to the recent work by Yuan et al. on the interconnection between the lncRNA MAYA, YAP signaling and iron (PMID: 34190396). In that study, YAP was found to prevent iron loading in a model of NAFLD.

For curiosity, it could be an option to say a few words on the lncRNA PACRG-AS1, which is also iron responsive and exerts a negative effect on iron.

I hope the specific comments below could help the authors improving their manuscript:

- Fig. 1A, S1A: The GSEA analysis shall be better explained: type of cells the data set was obtained from, why choosing LCN2 KD (given the minor role LCN2 plays in iron import compared to e.g. TFRC, etc.)
- Fig. 1B,C: What is the justification for using HEK cells that are not breast cancer cells?
- Calcein-AM is not so specific and may react with other divalent metals than iron, what is the evidence that what is measured is iron. Did the authors unquench calcein-AM in YAP-cells with a strong iron chelator such as SIH or equivalent?
- Fig. 1D,E: it is very hard to visualise the iron in the Perl's images. Furthermore, the iron signal in the first sample on the left seems localized in non-tumor cells. Were changes in iron levels in cells of the tumor micro-environment taken into consideration when scoring the samples?
- lanes 131-134: phrasing is a bit convoluted and could be improved.
- Fig. 1F: it is not clear how these data have been generated. Does it refer to previous work, in that case where is the data source? Or was it done specifically for this study, in which case the exact cell culture conditions must be specified. The main text indicates that HEK cells were treated with FAC/DFO (lane 134), the figure indicates MCF7 cells. Which one is it?
- Fig. 1M,N: the interpretation of these data may not be correct. Adding iron has no significant effect on colony formation when lncRIM is KD (1M) and DFO decreases colony formation regardless of the lncRIM status (the level is different, but not the response). These experiments show that the effect of lncRIM on this parameter (colony formation) involves iron-unrelated functions of this lncRNA.
- Fig S1 O,P: the effect at the protein level looks rather weak. A quantification of biological replicates

would be helpful, if possible/available.

- Fig.2: studies made with purified components, lysates, and cells overexpressing a protein of interest can sometimes yield false positive interactions. The main text could be more precise about how the experiments have been done. For example, make clear that 2A is done with in vitro transcripts incubated with cell extracts (lane 183: "determine whether CYTOSOLIC LncRIM interacts" sounds like endogenous LncRIM was pulled down), and that 2B is done with cells that over-express NF2.

- lane 195: "both in vivo and in vitro". Which in vitro experiment does this sentence refer to? Is "in vivo" used here to indicate that the experiment was done with endogenous LATS1/NF2 (Fig. 2F)?

- Fig. 2G: the data is hard to read since LncRIM overexpression seems to suppress NF2.

- Fig 2G,H: why is the experiment done with HEK and not with MCF7 cells?

- Fig 2 O: no difference between EV and FL (but there is one between FL and S3 although data spread is bigger). The student t-test is not suitable when comparing several groups (1 way ANOVA would be more adequate).

- Fig. 3A: the changes in Fig.3A are described as robust (lanes 234, 235: "robustly decreased or increased"), but they look rather mild.

- Fig. S3A is unnecessary since 3C also shows the effect of sh-YAP alone.

- Fig. 3F: the authors state that overexpression of LncRIM in YAP-silenced cells does not alter stimulate the expression of DMT1 and TFRC up to the level of what is seen in YAP-normal cells . Same for iron levels (lane 241), however the figure shows a statistically significant effect. Please explain.

- Fig. 3C to F: are YAP-control cells treated with a negative control shRNA?

- Fig. S3C and lanes 247,248: CHIP-Seq reveals the presence of a YAP/TEAD4 binding site in the DMT1 promoter (please specify source data and cell type in figure legend), however it does not formally show that YAP "mediated the transcription of the DMT1 promoter".

- Fig. 3I: Ideally, a mutant reporter lacking specifically the identified TEAD4-binding site would be included in such experiment. However, the reporter assay is not convincing to start with because YAP overexpression stimulates luciferase expression regardless of the presence or not of the DMT1 promoter (same fold change).

- Analysis of DMT1: alternative utilization of promoters and polyadenylation sites generates four DMT1 mRNA isoforms. Please specify if the qRT-PCR used discriminates or not between these isoforms. Similarly, is the anti-DMT1 antibody against a specific isoform or not? Also, indicate where the TEAD4 binding site is located relative to the two DMT1 promoters.

- Fig. 3N,O: data from control cells that have not been treated with IRP2-sh1 and/or LncRIM would be needed to appreciate the effect (or absence of effect) of these treatments.

- Fig. 3F: it would be good to have an uncropped image of this immunoblot. What is the evidence that this is IRP2? The antibody used is not specified. Also, IRPs are known to be present in excess and the KD of IRP2 seems very weak. A CRISPR-Cas9 approach would certainly be more efficient (if feasible with this cell line).

- Fig. S3N: does PLKO express a negative control shRNA?

- Fig. S3O: the functionality of the IRE of DMT1 has been questioned. How would IRP2 KD affect

DMT1? Also, is DMT1-IRE the isoform expressed in MCF7 cells and quantified in the study?

- Fig. S3P adds little value to the study.

- Fig. 4C: how to explain the gaussian effect of FAC? 500 μ M FAC could trigger oxidative stress and be toxic, did the authors perform viability assay?

- Fig. 4G: how does FAC affect NF2 expression and pull-down efficiency? Is it equal between FAC and control cells?

- Fig. 4I,J and lane 301: ambiguous phrasing, IncRIM overexpression has no effect on FTH1, the effect seems to be purely the one of FAC.

- Fig. S4E,F: in terms of fold change, IncRIM seems to have the same effect regardless of the presence or absence of FAC, especially in MCF7 cells, hence FAC does not alter the response of the cell to IncRIM fluctuation.

- Lane 306: " to promote cell proliferation". Cell proliferation has not been assayed in Figures 4 and S4. End the sentence at "Hippo-YAP signalling").

- Fig. 5D: iron staining is hard to see without DAB enhancement.

- Fig. 5F: expression changes in a bulk analysis of whole tumor samples could reflect changes in the cellular composition of the tissue, as opposed to changes in the cancer cells themselves. For example, the vascularization of the tissue seems to differ between control and IncRIM-KD tumors, and one cannot exclude that vascular cells contribute some of the western blot signals in 4F. Tissue staining would help to discriminate, if doable.

- Fig. 5H-I: cancer cells need iron to grow and as expected DFO decreases tumorigenesis regardless of whether IncRIM is overexpressed or not. These data do not show specifically that iron mediates the effect of IncRIM overexpression of tumorigenesis. A KD of TFRC and/or DMT1 would be more specific. Furthermore, DFO could have multiple effects for example by altering the tumor immune microenvironment.

- Fig. 5K: Iron staining hard to visualize.

- Fig. 6A, Perl's stain: what is the evidence that the iron signal is in tumor cells and not in immune cells. Fig. 6A does not give any information about the LIP, in contrast to what is stated in lane 340 (Perl's staining reveals mostly non-heme iron stored in ferritin degradation products).

- lane 347,348: there is some correlation between IncRIM expression, TFR1/DMT1 expression and tumor status, but no formal evidence that IncRIM promotes tumor development or aggressiveness by disturbing iron metabolism. This shall be rephrased.

Minor points.

- Lane 81: in Ref 15, iron loading does not activate but rather inhibits Wnt signaling.

- Lane 43: "the underlying mechanism has not yet been elucidated". There is already abundant literature on the topic, would be better to say "remains ill understood" or equivalent rather than "not elucidated".

- Iron 2+ and 3+ do not exist in biological systems, the valency state of iron shall be written as Fe(II) and Fe(III). Same for Fig. 2R and 6I.

- Calcein assay: for the non-expert reader, it would be helpful to specify (when the assay is mentioned for the first time) that the fluorescence signal is inversely correlated with the size of the LIP.
- lane 140: "to confirm the regulatory role of these lncRNAs in iron metabolism". Not to "confirm" but to assess.
- lane 144: explain CAPT + what has been done is polysome profiling, not ribosome profiling.
- lane 145: "inability to encode micropeptides" is a bit extreme as it could be a matter of sensitivity and/or context.
- lane 149: end of the sentence, specify "observed in multiple breast cancer CELL LINES".
- Fig 1Q: this figure is superfluous since the data is not really exploited
- lane 206: typo: "To further ANALYSE"
- lane 207-208: "constructed three different deletion mutants" (omit "fragments").
- Fig.6B: typo y-axis "Specomens".
- did the authors assess the status of STEAP proteins? Are they also stimulated by YAP?

Reviewer #3 (Remarks to the Author):

In the present study, the authors have reported the role of a novel lncRNA; lncRIM, in YAP-mediated iron metabolism. The authors demonstrated a positive correlation between YAP levels and intracellular iron levels in cell lines and breast cancer tissue samples. lncRIM is a target of the YAP/TEAD activator complex. lncRIM binds to NF2 on the membrane and prevents the interaction between NF2 and LATS1 kinase, thereby abolishing the LATS1-mediated phosphorylation and degradation of YAP. In this way, lncRIM promotes the nuclear translocation of YAP, thereby promoting YAP-mediated transcription activation of genes controlling iron metabolism. In general, this is an exhaustive study. Most of the experiments were conducted in breast cancer and kidney cancer cells, and the data is supported by experiments performed in mouse xenograft models as well in cancer patient tissue samples. The major drawback of the ms is that the authors have shown some sort of bias to connect the lncRIM and YAP pathway without performing an unbiased screen (see the details below). Also, several of the immunoblots seems to be modified (at least based on the pdf files that I have downloaded).

Specific comments

What was the basis for the authors to select only a subset of lncRNAs for RT-qPCR experiments shown in fig 1f? Did they perform an RNA-seq to identify the candidates initially, and then validate the top hits by RT-qPCR?

Did the experiment detail in fig 1h perform in presence of FAC? Since lncRIM is upregulated in FAC-treated cells, the iron-metabolism assay should be done under similar conditions.

what is the copy number of lncRIM? Authors argued that lncRIM by binding to NF2 quenches NF2 from interacting with LATS1. In order for this to happen, the copy number lncRIM should be comparable to the NF2 protein levels. This needs to be estimated.

Also, wherein the cytoplasm does IncRIM localize? Based on the model, it should co-localize with NF2 on the membrane.

Data presented in fig 2g should be quantified and also should be shown in the case of endogenous proteins. In addition, I see the reduced signal of both tagged-NF2 and LATS1 on the membrane in the absence of IncRIM. Does IncRIM also facilitate the localization of NF2 on the membrane? Again, in this context, it is important to show the localization of IncRIM in the cell.

Initial data indicates that both YAP and IncRIM positively influence the iron levels in BC cells. However, it is not clear to me why authors have presumed that both these molecules play in the same pathway, and therefore have decided to identify IncRIM interacting proteins that are part of the HIPPO pathway. This seems to me like a biased approach without a strong rationale. Ideally, IncRIM pull-down followed by mass spec should be conducted to identify the top hits.

There seem to be some undesirable modifications done on several of the immunoblots presented in the ms [some examples include Figs 2e (WCL:IB: HA) & f (IP: IB: NF2) and Fig S2b (IP-flag: IB: HA), Figs. S3d, f].

It is not clear why some of the experiments were done in HEK293T cells when the main focus of the paper is to understand the role of IncRIM in iron metabolism in breast cancer cells.

Does KD of IncRIM enhance NF2-LATS1 interaction?

fig 1B shows changes in the expression of iron metabolism genes in IncRIM-depleted cells. If IncRIM-depleted cells show such a dramatic increase in YAP phosphorylation, ultimately resulting in the inactivation of YAP, then one would expect reduced expression of a significant number of YAP target genes. Is that the case?

The model presented in fig 2r should be demonstrated by in vitro analyses. authors should test the binding kinetics of purified NF2 and LATS with varying concentrations of IncRIM.

Fig 1F panel wrote MCF7, whereas the text in the result section indicates HEK293T.

YAP shRNAs do not seem to efficiently deplete YAP (Fig s3a). Also, there seems to be some issue with the YAP-IB.

Fig s3b: How do the control and YAP-depleted cells show similar levels of CTGF and CYR61 mRNAs?

REVIEWER COMMENTS

Reviewer #1 (Remarks to the Author):

In their manuscript He et al. aim to investigate the possible correlation among long noncoding RNA, the Hippo-YAP pathway and iron metabolism, in a breast cancer contest. Overall the work presented is extremely well thought and executed. It elegantly combines the molecular mechanism and the functional aspects of the role of *lncRIM* in promoting breast cancer progression through modulation of iron metabolism. Worth noting are the experiments that uncover the molecular interaction between *lncRIM* and NF2. Despite it, this reviewer strongly believes that such a nice work would certainly benefit from more robust in vivo experiment. Even that xenograft transplantation is universally accepted as a model, orthotopic injections result in a more physiological tumour environment. Thus, performing orthotopic transplantation, using cells depleted for *lncRIM* and/or overexpressing it, will allow taking into account also the role of tumour microenvironment. This is required for further evaluation.

Response: We appreciate the reviewer's approval of our work, and thanks for the reviewer's constructive suggestion! Our study demonstrated one iron-trigger *LncRNA lncRIM* directly bound NF2 and wired up the Hippo-YAP pathway to modulate cellular iron metabolism via regulating the expression of DMT1 and TFR1, and ultimately promoted cell proliferation and breast cancer progression.

Following the reviewer's suggestion, we carried out orthotopic injection into nude mice. As shown in **Response Figs.1a-e (New Figs. 5h-k, S5j)**, orthotopic injection of over-expressed *LncRIM* MDA-MB-468 cell lines significantly promoted xenograft tumor growth indicated by the increased Ki67 and YAP, accompanied by increased cellular iron level, the high expression of DMT1 and TFR1, which was consistent with **New Fig. 5a-e**. Meanwhile, we also found that over-expression of *LncRIM* partially reversed diminished the cell proliferation and tumor growth caused by DFO treatment (**New Fig. 1n, S5e-h**). Thus, these data further validated the important role of *LncRIM* in regulating intercellular iron levels and tumor progression.

The tumor environment is one complex and dynamic environment consisting of cancer cells, immune cells, fibroblasts, and others that contribute to cancer progression (PMID:31350295). Emerging evidence also suggests that iron metabolism is also involved in the tumor environment (PMID:33679721). On the one hand, M2 macrophages reside in the TME can recycle the erythrocytes' iron via Ferroportin (FPN) or promote the expression of LCN2 of cancer cells to provide sufficient iron, and can also activate factors in tumor cells to regulate signaling pathways related to iron metabolism (PMID:21705499, PMID: 32675368, PMID:33628389). And the previous study demonstrated that breast tumor cells induce TAM into an iron-release phenotype (PMID:29399416). On the other hand, iron overload cancer cells can also in turn promote M2 macrophage expression in the TME (PMID:28286378), while cancer had better survival with M1 macrophages (PMID:31383898). Importantly, Sun et al

reported that tumor cells could directly compete for iron with tumor-associated macrophages, and lead to M2 macrophage polarization by induction of HIF-1 α (PMID:34389031). Besides, a previous study also showed that YAP activation can help TICs recruit M2 macrophages to promote cancer cell development (PMID:28223311). Interestingly, our data showed that over-expression of *LncRIM* slightly promoted M2 macrophage expression, however, double knockdown of DMT1 and TFR1 significantly reduced the M2 macrophages, and lead to M1 macrophages polarization in **Response Fig.1f (New Fig.S5m)**, which provides evidence to the potential link of lncRNAs-mediated cellular iron metabolism to the tumor environment. Besides, we are also very interested in deeply exploring the relationship between tumor environment and the iron metabolism of cancer cells in the next field.

Response Fig.1

(a-c) Nude mice were injected with control, *LncRIM* over-expressed, or double knockdown of DMT1 and TFR1 with over-expression of *LncRIM* MDA-MB-468 cell lines (a) tumor volumes (b) and tumor weights (c) were assessed. The data are presented as the mean \pm SD from n = 5 mice per group. (***)P < 0.001, One-way /Two-way ANOVA analysis).

(d and e) Representative IHC staining and enhanced DAB iron staining of randomly selected tumors from mice subcutaneously injected with the indicated stably transduced MDA-MB-468 cells (d). Scale bar, 100 μ m. The relative intensities were quantified by ImageJ (Fiji version 1.51 software) (e). The data are presented as the mean \pm s.d. from n = 5 xenograft tumor samples per group. (n.s., not significant; ***)P < 0.001, One-way ANOVA analysis).

(f) Flow cytometry analyses of M1 and M2-like tumor-associated macrophages from orthotopic injection tumor.(mean \pm SD, n=5 independent experiments, *P < 0.05, ***)P < 0.001,Students' t-test).

in Nature Communication.

Specific comments listed below need to be fully addressed:

1) In Figure 1m the effect of FAC in rescuing cell growth is really minimal. Showing the statistical significantly for all the columns would certainly make the data more robust.

Response: Thanks for your suggestion! We have re-analyzed the data and made all the statistical analysis of all the columns in the revised manuscript (**New Fig.1m**), which showed FAC stimulation partially reversed *LncRIM*-silence induced inhibition of proliferation.

2) In the section titled “*LncRIM* interacts with NF2 to inactive LATS1 kinase” you claim that overexpression of *LncRIM* significantly reduced the association of NF2 and LATS1 both in vivo and in vitro, as shown in Fig 2e-f and Fig S2b. However, no in vivo data are shown.

Response: We are sorry for this confusion. Here, the “*in vivo*” was used to refer to the experiment that was done with endogenous LATS1 and NF2 or MCF7 cells expressing HA-NF2 and SFB-LATS1 (**New Figs. 2e-h**), while “*in vitro*” referred to experiment that was done with purified protein. Thus, by using the purified GST-NF2, we performed GST pull-down experiment (**Response Fig. 2**) (**New Fig. S2g**), which also showed that over-expression of *LncRIM* significantly inhibited the interaction between LATS1 and NF2. We have added this data to the revised manuscript.

Response Fig.2

(a) GST pull-down assay was performed with GST-tagged NF2 and MCF7 cells lysates expressing Flag-LATS1 and over-expression of *LncRIM*. GST-NF2 was pulled down by GST beads. GST was used as the negative control.

3) In Fig 3c the silencing of YAP performed with the construct #2, does not seem to work as nice as the #1. In fact, overexpression on *LncRIM* in those conditions brings back TFR1 protein levels. That should be taken into account in the text.

Response: Thanks for pointing out this issue! **Fig. 3c is now New Fig. 3c**. Considering the role of *LncRIM* in cellular iron metabolism and binding to NF2 (**New Figs. 1k, 1l, 2a-e**), we were interested in examining the correlation between *LncRIM*, iron metabolism, and Hippo-YAP pathway. We apologized for the confusion of the previous Fig.3c, which may be because of the poor knockdown efficiency of YAP shRNA2. We have reconstructed cell lines with two different shRNA targeted at YAP and repeated this experiment (**Response Fig. 3**) (**New Figs. 3c-f**). The results suggested that *LncRIM* regulated the expression of DMT1, TFR1 and the cellular iron level dependently on the Hippo-YAP pathway. We also quantified the related western blot images and added this data to the revised manuscript.

Response Fig.3

(a) Immunoblot detection of the DMT1 and TFR1 expression in control and YAP knockdown MCF7 cells with or without over-expression of *LncRIM*.

(b) The cellular iron level in control and YAP knockdown MCF7 cells with or without over-expression of *LncRIM* was measured with Calcein-AM assay. Values were normalized to the control group. (n = 3 biological independent experiments, n.s., not significant, **P < 0.01; ***P < 0.001, Two-way ANOVA analysis).

(c and d) RT-qPCR detection of DMT1 (c) and TFR1 (d) expression in control and YAP knockdown MCF7 cells with or without over-expression of *LncRIM*. (mean ± SD, n=3 independent experiments, n.s., not significant; ***P < 0.001, Two-way ANOVA analysis).

4) In Fig 3i-k data the authors only show data on DMT1. Data on TFR1 levels in the presence of active or inactive YAP mutant are never shown. Panels should be added to the figure.

Response: Thanks for pointing out this issue! **Figs.3i-k is now New Figs.S3c-e.** We apologized for the unclear label in previous image. In fact, we have shown both the expression of TFR1 and DMT1. We are sorry again for this confusion.

5) Fig 3n would certainly benefit from an additional control, without the IRP2 KO, in order to assess the level of the rescue. The same for Fig S3g-h

Response: We appreciate the reviewer's suggestion! **Fig.3n is now New Fig.3k.** We further added the additional control groups and did more biological repeats. As shown in **Response Fig.4 (New Figs. 3k, 3l, and S3l, S3m)**, IRP2 knockdown significantly

decreased the expression of DMT1 and TFR1, however, over-expression of *LncRIM* and YAP still promoted the expression of DMT1 and TFR1 in both mRNA and protein level as well as the cellular iron level under the knockdown of IRP2, suggesting this *LncRIM*-Hippo axis acted effectively in a distinct manner of IRP2. We appreciated the reviewer's kind suggestion and provided relative quantification of the key immunoblot data.

Response Fig.4

(a) Immunoblot detection of DMT1 and TFR1 expression of control and IRP2 knockdown MCF7 cells with or without over-expression of *LncRIM* and YAP.

(b) The cellular iron level of control and IRP2 knockdown MCF7 cells with or without over-expression of *LncRIM* and YAP was measured with Calcein-AM assay. (mean ± SD, n=3 independent experiments; ***P < 0.001, Two-way ANOVA analysis).

(c and d) RT-qPCR detection of TFR1 (c) and DMT1 (d) expression of control and IRP2 knockdown MCF7 cells with or without over-expression of *LncRIM* and YAP. (mean ± SD, n=3 independent experiments; ***P < 0.001, Two-way ANOVA analysis).

6) In Fig 4b-c you state that the effect of FAC on LATS1 and YAP phosphorylation is dose dependent. However, from the images it is clear that higher concentrations of FAC bring the phosphorylation back. The text should be modified accordingly and an explanation should be provided.

Response: Figs.4b, c is now New Figs.4b, c. Thanks for pointing out this issue! As one of the fundamental elements, iron play important role in biological processes and tumor growth by wiring up some signaling pathways such as WNT signaling and JNK pathway (PMID:21666721, PMID:27546461). However, excessive iron is also reported to damage DNA and protein by producing ROS and is increasingly considered an

important mediator of cell death, such as apoptosis and ferroptosis (PMID:24346035, PMID:31105042, PMID:18355723). Moreover, studies also showed that YAP is inactive response to cellular apoptosis (PMID: 33086070), and YAP/TAZ deletion increases ROS buildup to promote oxidative stress-induced cell death (PMID:31063758). Thus, we then measured the cell viability under FAC stimulation of different concentration (**Response Fig.5**) (**New Fig. S4d**). The data showed that higher level of iron (>400 μ m) significantly led to decreased cell viability, while the lower concentration of iron could promote cell proliferation compared to the control cells, which is consistent with our previous immunoblot finding. We have modified the description in the revised manuscript and discussed it.

Response Fig.5

(a) Cell proliferation viability of MCF7 cells with FAC stimulation of different concentration was measured with MTT (mean \pm SD, n=3 independent experiment, **P < 0.01; ***P < 0.001, Two-way ANOVA analysis).

Reviewer #2 (Remarks to the Author):

He, Qu et al. investigate the interconnection between lncRNAs, YAP signaling, and iron metabolism in breast cancer. They identify lncRIM, a lncRNA whose expression is stimulated by iron and is associated with poor survival in breast cancer patients. They also show that YAP signalling increases cellular iron levels, likely due to direct stimulation of TFRC and DMT1 transcription by YAP/TEAD. The authors then connect the two findings and describe a positive feedback loop whereby lncRIM itself enhances YAP signaling to further increase iron levels in the cell. From a mechanistic point of view, the authors identify and map an interaction surface between lncRIM and the tumor suppressor NF2. This interaction impedes the binding of NF2 to LATS1, resulting in downstream activation of YAP. Tumor transplantation experiments in mice and clinical data from cancer patients suggest that this YAP-iron-lncRIM feedback loop may contribute to promoting tumor growth by increasing cellular iron availability in cancer cells.

Understanding how exactly cancer signalling pathways and iron metabolism crosstalk is a very interesting topic, yet it remains poorly understood. He, Qu et al. make an important contribution to the field and their work will be of broad interest to scientists interested in cancer signalling and cancer metabolism. The authors present a tremendous amount of data, and their detailed characterisation of the mechanism through which lncRIM affects YAP signalling and iron is rather advanced. As in every study, some aspects require improvement.

Response: We appreciated your recognition and favor of our work! And thanks for your suggestions!

The association between YAP/iron metabolism and tumor characteristics is largely based on correlations, without formal evidence of causality. This is normal when looking at clinical data. From an experimental viewpoint, the utilization of an iron chelator/donor to show that *lncRIM* promotes cell/tumor growth by increasing iron levels via YAP is not convincing enough. As discussed below, DFO can inhibit tumor growth or colony formation regardless of *lncRIM* status. In vivo, DFO might also influence tumor development indirectly. Furthermore, adding an iron donor does not improve colony formation in *lncRIM*-KD cells. This shows that *lncRIM* can influence tumor growth independently of iron. More specific approaches would be better. For example, if the stimulation of iron via YAP would be due to upregulation of TFRC/DMT1, preventing TFRC/DMT1 upregulation would be predicted to antagonize the effect of *lncRIM* on cell/tumor growth. In the absence of a clear demonstration, the authors shall at least temper their statements throughout the manuscript.

Response: Thanks for the reviewer's constructive suggestion! Here we uncovered one new mechanism in which *LncRIM* wired up the Hippo-YAP pathway to promote cellular iron levels by regulating the expression of DMT1 and TFR1, and ultimately affected tumor growth. Besides, this *LncRIM*-NF2 axis acted effectively in a distinct manner of IRP2 and ultimately promoted breast cancer progression (New Fig.1k,1l, 1q, 3c-f, 3k, 3l, S3l, S3m, 5a-d).

Following the reviewer's suggestion, we also carried out the orthotopic injection of over-expressed *LncRIM*, double knockdown of TFR1 and DMT1 with over-expression of *LncRIM* MDA-MB-468 cell lines. And the result showed knockdown of TFR1 and DMT1 significantly inhibited *LncRIM*-mediated cell proliferation and breast cancer growth as well as the cellular iron level (Response Fig.6) (New Figs. 5h-k, S5j-l), which suggested that preventing both TFR1 and DMT1 upregulation would antagonize the effect of *LncRIM* on cell proliferation and tumor growth. Together, the above data to some extent further supported the idea that *LncRIM* promoted breast cancer growth via upregulating cellular iron level.

Response Fig.6

(a-c) Nude mice were injected with control, *LncRIM* over-expressed, or double knockdown of DMT1 and TFR1 with over-expression of *LncRIM* MDA-MB-468 cell lines (a) Tumor volumes (b) and tumor weights (c) were assessed. The data are presented as the mean \pm SD from $n = 5$ mice per group. (***) $P < 0.001$, One-way/Two-way ANOVA analysis.

(d and e) Representative IHC staining and enhanced DAB iron staining of randomly selected tumors from mice subcutaneously injected with the indicated stably transduced MDA-MB-468 cell lines (d). Scale bar, 100 μ m. The relative intensities were quantified by ImageJ (Fiji version 1.51 software) (e). The data are presented as the mean \pm s.d. from $n = 5$ xenograft tumor samples per group. (n.s., not significant; ***) $P < 0.001$, One-way ANOVA analysis).

(f and g) Colony formation assay of control, *LncRIM* over-expressed and double knockdown of DMT1 and TFR1 with over-expression of *LncRIM* MCF7 cell lines (f). The data are presented as the mean \pm s.d., (g). ($n = 3$ biological independent experiments, ***) $P < 0.001$, Student's *t*-test).

The data based on Perl's staining without DAB enhancement is hard to judge given how weak the signal is (furthermore, localization of the signal is of importance but is not discussed).

Response: Thanks for this suggestion! We have done more biological repeats with enhanced DAB iron staining assay, which is usually used to detect some tissues with low iron content and can amplify the iron signal (PMID:33895792) to further support our conclusion of *LncRIM*-mediated cellular iron metabolism. We also added the statistical analysis (New Figs. 1d, 5d, 5j, S5g).

Similarly, it is sometimes hard to appreciate the immunoblot results from a quantitative perspective as the effects are in some cases rather subtle.

The authors may also want to consult a statistician to inquire whether all their statistical analyses are optimal. Using a student t-test for small samples is generally not appropriate, so is performing multiple t-tests to compare more than two groups.

Response: We appreciated the reviewer's suggestion! We have revised the statistical analysis in the revised manuscript. All data are presented as mean \pm SD. Statistical analysis was carried out using Image-J (Fiji version 1.51 software). The student's unpaired *t*-test was used to compare differences between control and a single test group. One-way or Two-way ANOVA analysis was used to detect statistical differences ($P < 0.05$) between multiple groups.

It would be interesting to discuss the data in relation to the recent work by Yuan et al. on the interconnection between the lncRNA *MAYA*, YAP signaling and iron (PMID: 34190396). In that study, YAP was found to prevent iron loading in a model of NAFLD.

Response: Thanks for this suggestion! lncRNA *MAYA* (MST1/2-Antagonizing lncRNA for YAP Activation) was firstly found and demonstrated its function in our previous work (PMID:28114269). We revealed the *MAYA* can form RNA-protein complex with LLGL2 and NSUN6 to methylate Hippo/MST1 at Lys59, which leads to activation of YAP and finally elicits osteoclast differentiation and bone metastasis. However, in the work directed by Yuan et al (PMID: 34190396), they uncovered that lncRNA *MAYA* inhibits YAP activation to promote PA-induced iron overload in non-alcoholic fatty liver disease (NAFLD). Considering the opposite role of *MAYA* on the Hippo pathway in these two different disease models, it may trigger other molecular involved in regulation of the Hippo-YAP pathway in non-alcoholic fatty liver disease (NAFLD) compared to breast cancer model, which requires more research to in-depth understanding the key role of selection of different disease models in mechanistic studies.

In our study, we uncovered one iron-trigger lncRNA *LncRIM* (lncRNA related to iron metabolism) wired up the Hippo-YAP pathway to regulate cellular iron metabolism, and finally promoted breast cancer growth (New Figs. 1k, 1l, 2a-h, 3c-g and New Fig.5). Besides, we also demonstrated that this *LncRIM*-NF2-DMT1/TFR1 axis acted

effectively in a distinct manner of the IRP2 system in cellular iron metabolism (**New Figs. 3k, 3l, S3l, S3m**).

Besides, the liver is known as one important iron site for storage organs (PMID:17014365). And the liver can also orchestrate systemic iron balance by producing hepcidin (PMID:30401708), thus, liver is susceptible to damage caused by iron deposition (PMID:31193082). Together, animal models may lead to differences in the regulation mechanism of iron metabolism.

For curiosity, it could be an option to say a few words on the lncRNA PACRG-AS1, which is also iron responsive and exerts a negative effect on iron.

Response: Thanks for this suggestion! By using FAC and DFO stimulation, we screened out 6 potential lncRNAs that responded to iron level, and by knocking down these 6 lncRNAs, we found our *LncRIM* has the most significant effect on the intracellular iron level among others. Besides, the data also showed that lncRNA *PACRG-AS1* also responded to FAC and DFO stimulation, while the silence of *PACRG-AS1* mildly increased the cellular iron level (**New Figs.1g,1h**). It may be possible for lncRNA *PACRG-AS1* to respond to the regulation of ferroptosis pathway and inhibit cell proliferation, which lead to the downstream negative signaling pathway or effector to play the compensatory effect (PMID:30944473, PMID:25759022).

I hope the specific comments below could help the authors improving their manuscript:

- Fig. 1A, S1A: The GSEA analysis shall be better explained: type of cells the data set was obtained from, why choosing LCN2 KD (given the minor role LCN2 plays in iron import compared to e.g. TFRC, etc.)

Response: Thank you for pointing out this issue! We have obtained our data (GEO number: GSE38369) from the GEO database (PMID:22767506). The GSE38369 was acquired using LCN2-knockdown MDA-MB-231 cells. LCN2-mediated iron transport is different from the TF-TFR1 pathway and is involved in diverse cell physiological processes (PMID:12453413, PMID:16377569). Meanwhile, several studies have reported that LCN2 is up-regulated in many cancer cells, and promotes tumor growth by promoting cellular iron accumulation, while knockdown of LCN2 inhibits cell proliferation and iron level (PMID: 32675368, PMID:12453413), suggesting the vital role of LCN2 in iron homeostasis and cancer progression. We have added this to the revised manuscript.

We performed GSEA analysis by using the GSEA MSigDB 7.0 (c6:oncogenic signature gene sets), and found that YAP CONSERVED SIGNATURE was highly

enriched, suggesting that the Hippo-YAP pathway may be involved in cellular iron metabolism. To further confirm this observation, we also carried out several experiments including **New Figs. 1b-e** to validate the correlation between iron metabolism and the Hippo-YAP signaling pathway. Together, these experiments illustrated the potential role of the Hippo pathway in cellular iron metabolism.

- Fig. 1B,C: What is the justification for using HEK cells that are not breast cancer cells?

Response: Thanks for your question! **Fig.1b, c is now New Fig. 1b, c.** We have done more biological repeats in MCF7 cells (**New Figs.1b, c**). And the result showed that active YAP significantly promoted the cellular iron level. Previous studies have shown the molecules of the Hippo-YAP pathway play the same biological regulation and mechanism in HEK293 cells as it in cancer cells (PMID:24012335, PMID:17974916, PMID:26045165). Meanwhile, in HEK293 cells, the Hippo-YAP pathway also responds to the various metabolic signals and presents similar changes of it in cancer cells (PMID:29100056, PMID:30472188, PMID:22863277). Thus, it is feasible to do some auxiliary experiments in HEK293 cells to support our findings. Thanks again for pointing out this issue, which can help us to better improve our manuscript.

- Calcein-AM is not so specific and may react with other divalent metals than iron, what is the evidence that what is measured is iron. Did the authors unquench calcein-AM in YAP-cells with a strong iron chelator such as SIH or equivalent?

Response: Thanks for this suggestion! The Calcein-AM assay is a widely used technique to assay the intracellular iron level (PMID:30449675, PMID:31267712, PMID:28819251, PMID:17233627). Increased iron concentration can quench the fluorescence of calcein, and the quenching efficiency can be up to 90%. However, other metals such as Ca^{2+} and Mg^{2+} are neither affected nor quenched the fluorescence of calcein like iron (PMID:25315476).

Further, we used deferiprone (DFP), one orally administered iron chelator in clinical to decrease cellular iron levels (PMID: 31267712, PMID:32975364, PMID:35471096) to treat MFC7 cells. As shown in **Response Fig.7**, active YAP (YAP-5SA) dramatically quenched the fluorescent of calcein, while the addition of DFP or DFO dramatically reversed the fluorescent to a similar level of control.

Response Fig.7

(a and b) Cellular iron level of control, YAP-5SA with or without the DFP (60 μM) or DFO (100 μM) was measured by Calcein-AM assay (mean \pm SD, n=3 independent experiment, n.s., not significant; **P < 0.01; ***P < 0.001, One-way ANOVA analysis).

- Fig. 1D,E: it is very hard to visualise the iron in the Perl's images. Furthermore, the iron signal in the first sample on the left seems localized in non-tumor cells. Were changes in iron levels in cells of the tumor micro-environment taken into consideration when scoring the samples?

Response: Thanks for pointing out this issue! **Fig.1d, e is now New Fig.1d, e.** We have done more biological repeats with enhanced DAB iron staining assay and counted the total iron level of the samples (**Response Fig. 8**) (**New Fig. 1d, e**). The data indicated that the activation of YAP was positively correlated with the iron level in breast cancer tissue.

we also test the expression of M1 and M2 macrophages in control and *LncRIM* over-expressed tumors. The result showed high expression of *LncRIM* slightly promoted the M2 macrophages (**Response Fig. 8c**) (**New Fig. S5m**). which provides evidence of the potential link of lncRNAs-mediated cellular iron metabolism to tumor environment. Besides, we are also very interested in deeply exploring the relationship between tumor environment and the iron metabolism of cancer cell in the next field.

Response Fig. 8

(a and b) The expression of YAP was positively correlated with iron level. Enhanced DAB iron staining and immunohistochemistry staining of YAP were performed by using breast cancer tissue arrays. (a) The region in each box is enlarged below (b). Enhanced DAB iron staining indicates a positive iron level. The region in each box is enlarged below. Scale bar, 200 μ m.

(c) Flow cytometry analyses of M1 and M2-like tumor-associated macrophages from orthotopic injection tumor. (mean \pm SD, n=5 independent experiment, n.s., not significant; *P < 0.05; ***P < 0.001, Student's *t*-test).

- lanes 131-134: phrasing is a bit convoluted and could be improved.

Response: Thanks for pointing out this issue! We have revised the description carefully in the revised manuscript.

- Fig. 1F: it is not clear how these data have been generated. Does it refer to previous work, in that case where is the data source? Or was it done specifically for this study, in which case the exact cell culture conditions must be specified. The main text

indicates that HEK cells were treated with FAC/DFO (lane 134), the figure indicates MCF7 cells. Which one is it?

Response: We are apologized for this confusing description! **Fig.1f is now New Fig.1f.** Herein, we screened out 6 potential lncRNAs responded to both FAC and DFO stimulation in MCF7 cells according to our previous study, where we have pinpointed 40 lncRNAs that were potentially required for YAP1-dependent transcription by transfecting the human Lincode® siRNA library into MCF7 cells that were engineered with a TEAD-driven luciferase reporter (PMID:28114269). We have correlated this in the revised manuscript and thanks again for pointing out this issue.

- Fig. 1M,N: the interpretation of these data may not be correct. Adding iron has no significant effect on colony formation when lncRIM is KD (1M) and DFO decreases colony formation regardless of the lncRIM status (the level is different, but not the response). These experiments show that the effect of lncRIM on this parameter (colony formation) involves iron-unrelated functions of this lncRNA.

Response: Thank you for pointing out this issue and we are sorry for this confusion! **Figs.1m, n is now New Figs.1m, n.** We have re-analyzed and made all the statistical analyses of all the columns in the revised manuscript to make this data more robust. Besides, we also found FAC stimulation partially reversed cell cycle arrest of *LncRIM*-silenced MCF7 cells (**Response Fig.9**) (**New Figs. S11**). Moreover, knockdown of *LncRIM* in mouse model also significantly inhibited tumor growth and iron level (**New Figs. 5a-d**). Besides, over-expression of *LncRIM* partially restored DFO treatment induced inhibition on cell proliferation, breast cancer growth and iron levels (**New Figs. 1n, S1m, S5e-g**).

Further, as shown in **New Figs. 5h-k, S5k, S5l**, the colony formation assay and orthotopic injection of mouse model also showed that knockdown of DMT1 and TFR1 significantly decreased the *LncRIM*-induced cell proliferation and tumor growth. To some extent, these results supported the idea that *LncRIM* promoted cell proliferation and breast cancer development by regulating cellular iron metabolism.

Response Fig.9

(a) Cell cycle arrest in control and *LncRIM*-silenced MCF7 cells with or without FAC (200 μ M) stimulation for 24hr was measured by flow cytometry analysis.

- Fig S1 O,P: the effect at the protein level looks rather weak. A quantification of biological replicates would be helpful, if possible/available.

Response: Thanks for this suggestion! **Fig.S1o, p is now Fig.S1q.** We did biological repeats and added biological statistical analysis (**Response Fig.10**).

Response Fig. 10

(a and b) Immunoblot detection of DMT1 and TFR1 expression in control and *LncRIM* over-expressed or *LncRIM* knockdown MCF7 cells and MDA-MB-468 cells.

- Fig.2: studies made with purified components, lysates, and cells overexpressing a protein of interest can sometimes yield false positive interactions. The main text could be more precise about how the experiments have been done. For example, make clear that 2A is done with in vitro transcripts incubated with cell extracts (lane 183: "determine whether CYTOSOLIC LncRIM interacts" sounds like endogenous LncRIM was pulled down), and that 2B is done with cells that over-express NF2.

Response: Thanks for this kind suggestion! We have revised the description carefully in the revised manuscript.

- lane 195: "both in vivo and in vitro". Which in vitro experiment does this sentence refer to? Is "in vivo" used here to indicate that the experiment was done with endogenous LATS1/NF2 (Fig. 2F)?

Response: We are sorry for this confusion. Here, the "in vivo" was used to refer to the experiment that was done with endogenous LATS1/NF2 or MCF7 cells expressing HA-NF2 and SFB-LATS1 (**New Figs. 2g-i**), while "in vitro" referred to experiment that was done with purified protein. Thus, by using the purified GST-NF2, we performed GST pull-down experiment (**Response Fig.11**) (**New Fig. S2g**), which also suggested

that over-expression of *LncRIM* inhibited the binding of LATS1 and NF2. We have added this description in the revised manuscript.

Response Fig. 11

(a) GST pull-down assay was performed with GST-tagged NF2(20nmol) and MCF7 cells lysates expressing Flag-LATS1 and over-expression of *LncRIM*. GST-NF2 was pulled down by GST beads. GST was used as the negative control. Immunoblot was performed to assess the expression of Flag-LATS1.

-Fig. 2G: the data is hard to read since *LncRIM* overexpression seems to suppress NF2.

Response: Thanks for pointing out this issue! **Fig.2g is now New Fig.2j.** We firstly examined the endogenous expression of NF2 on the membrane in control and *LncRIM* over-expressed MCF7 cells with the indicated membrane marker (**Response Figs.12a, b**) (**New Figs. S2d, e**). The results indicated that *LncRIM* had little effect on the membrane localization of NF2. For the previous Fig.2g, the confusion may due to the transfection efficiency of GFP-NF2. We are sorry for the quality of previous data. And we apologized for not finding different species of NF2 and LATS1 antibodies to carry out endogenous two-color fluorescent labeling experiments. Thus, we did more biological repeats with GFP-NF2 and pRFP-LATS1 in MCF7 cells (**Response Fig.12c, d**) (**New Figs. 2j, k**). Combined with **New Figs.2g-i** and **New Figs.S2f-j**, we concluded that over-expression of *LncRIM* significantly inhibited LATS1 and NF2 interaction.

Response Fig. 12

(a and b) The membrane location of NF2 with or without over-expression of *LncRIM* in MCF7 cells was detected by immunofluorescence staining (a). The membrane was used as a positive control (Red) and the NF2 was detected with Alexa Fluor 488. Scar bar, 10 μ m. The subcellular location of NF2 (F_{NF2}/F_{Mem}) was analyzed with Image J. (mean \pm SD, n=5 independent experiment, n.s., not significant, Student's *t*-test)

(c and d) Immunofluorescence staining was performed in MCF7 cells to examine the binding of LATS1 and NF2 with or without over-expression *LncRIM*. The data was quantized with Image J. (mean \pm SD, n=5 independent experiment; ***P < 0.001, Student's *t*-test).

- Fig 2G,H: why is the experiment done with HEK and not with MCF7 cells?

Response: Thanks for pointing out this issue! **Fig.2g, h is now New Fig.2i, j.** For the previous **Fig.2e, f**, we determined to further confirm the observation demonstrated in **New Fig. 2g, 2h, S2g, S2h, S2j**. Considering the convenience of HEK293T with high efficiency of transfection and the same biological regulation of the Hippo-YAP pathway as it in cancer cells (PMID:24012335, PMID:26045165), we thus chose HEK293T cells to do these experiments. To address this concern, we also repeated this experiment in MCF7 cells and added this data in the revised manuscript (**Response Fig.13**) (**New Figs. 2i-k and S2i**).

Response Fig. 13

(a and b) Immunofluorescent staining was performed to examine the interaction between LATS1 and NF2 in the control and *LncRIM* over-expressed MCF7 cells. The data was quantized with Image J. (mean \pm SD, n=5 independent experiment; ***P < 0.001, Student's *t*-test).

(c and d) Subcellular fractionation and immunoblot were performed in MCF7 cells to examine the LATS1 membrane association under control or the over-expression of *LncRIM*. (M: membrane; C: cytoplasm; T: total).

-Fig 2 O: no difference between EV and FL (but there is one between FL and S3 although data spread is bigger). The student t-test is not suitable when comparing several groups (1 way ANOVA would be more adequate).

Response: Thanks for this kind suggestion! **Fig. 2o is now New Fig. 2p.** We have re-analyzed the statistics between different groups by using One-way ANOVA analysis,

and we also checked carefully all the statistical analyses in the manuscript.

- Fig. 3A: the changes in Fig.3A are described as robust (lanes 234, 235: "robustly decreased or increased"), but they look rather mild.

Response: Thanks for pointing out this issue! **Fig. 3a is now New Fig.3a.** We have changed the description of "robustly decreased or increased" into "decreased or increased".

- Fig. S3A is unnecessary since 3C also shows the effect of sh-YAP alone.

Response: Thanks for this suggestion! We have deleted this data in the revised manuscript.

- Fig. 3F: the authors state that overexpression of lncRIM in YAP-silenced cells does not alter stimulate the expression of DMT1 and TFR1 up to the level of what is seen in YAP-normal cells. Same for iron levels (lane 241), however the figure shows a statistically significant effect. Please explain.

Response: Thanks for pointing out this issue! **Fig.3f is now New Fig. 3f.** We are very sorry for the quality of the previous Fig.3f, which may be due to the poor YAP knockdown efficiency. We have reconstructed cell lines with two different shRNA targeted at YAP, repeated this experiment, and also quantified the related western blot images (**Response Fig.14**) (**New Figs. 3c-f**). Besides, re-expression of YAP in YAP-silenced MCF7 cells significantly reversed the *LncRIM*-mediated upregulation of DMT1 and TFR1 as well as the YAP downstream genes (**New Figs.3g, 3h, S3a, S3b**).

Response Fig.14

(a) *LncRIM* regulated the expression of DMT1 and TFR1 in YAP-dependent manner. Immunoblot detection of the DMT1 and TFR1 expression in control and YAP knockdown MCF7 cells with or without over-expression of *LncRIM*.

(b) The cellular iron level in control and *LncRIM* knockdown MCF7 cells with or without over-expression of *LncRIM* was measured with Calcein-AM assay. Values were normalized to those in the control group (mean \pm SD; n = 3 biological independent experiments, n.s., not significant, **P < 0.01; ***P < 0.001, Two-way ANOVA analysis).

(c and d) RT-qPCR detection of DMT1 (c) and TFR1 (d) expression in control and YAP knockdown MCF7 cells with or without over-expression of *LncRIM*. (mean \pm SD, n=3 independent experiments, n.s., not significant; ***P < 0.001, Two-way ANOVA analysis).

-Fig. 3C to F: are YAP-control cells treated with a negative control shRNA?

Response: We are sorry for this description. The control cells were treated with control shRNA, and we have corrected “PLKO.1” into “Scramble” in the revised manuscript.

- Fig. S3C and lanes 247,248: CHIP-Seq reveals the presence of a YAP/TEAD4 binding site in the DMT1 promoter (please specify source data and cell type in figure legend), however it does not formally show that YAP "mediated the transcription of the DMT1 promoter".

Response: Thanks for this kind suggestion! **Fig.S3c is now New Fig. S3h.** We used the ChIP-seq data from the GEO dataset (GSE107013) (PMID:30082728), which was performed in MCF7 breast cancer cell lines. We also added the information in the figure legend. We have also labeled the promoter regions of DMT1 in **Response Fig.15 (New Fig. S3h).**

Response Fig.15

(a) ChIP-seq analysis of the YAP/TEAD binding elements in DMT1 promoter region.

- Fig. 3I: Ideally, a mutant reporter lacking specifically the identified TEAD4-binding site would be included in such experiment. However, the reporter assay is not convincing to start with because YAP overexpression stimulates luciferase expression regardless of the presence or not of the DMT1 promoter (same fold change).

Response: Thanks for this kind suggestion! We carried out the luciferase reporter experiment with mutated DMT1-promoter lacking the YAP/TEAD-DMT1 binding site to further assess the previous finding in **New Figs. 3i, 3j.** As shown in **Response Fig.16**

(**New Fig. S3i**), the result showed mutation of YAP/TEAD-binding sites in the DMT1 promoter significantly attenuated YAP/TEAD-induced DMT1 promoter luciferase activity. We have added this data in the revised manuscript. Moreover, combined with the ChIP-qPCR and bioinformatics analysis, these data together illustrated that DMT1 was one downstream gene of YAP (**New Figs. 3i, 3j, S3h**).

Response Fig.16

(a) Luciferase reporter assay detection in MCF7 cells expressing YAP- 5SA and wild DMT1-promoter or mutant DMT1 promoter. The values were normalized to those in the control group. (mean \pm SD, n = 3 biological independent experiments, n.s., not significant, *** $P < 0.001$, Two-way ANOVA analysis.)

- Analysis of DMT1: alternative utilization of promoters and polyadenylation sites generates four DMT1 mRNA isoforms. Please specify if the qRT-PCR used discriminates or not between these isoforms. Similarly, is the anti-DMT1 antibody against a specific isoform or not? Also, indicate where the TEAD4 binding site is located relative to the two DMT1 promoters.

Response: Thanks for the reviewer's suggestion! DMT1 mRNA has four different isoforms due to the different variants in the 3'-UTR (IRE+ and non-IRE), and the 5' end mRNA processing variant (1A and 1B) (PMID:12209011, PMID:9642100). We examined the expression of these four isoforms in MCF7 cells and MDA-MB-468 cells with the specific set of primers specifically target for these four isoforms according to previous paper (PMID:12209011). As shown in **Response Fig.17a (New Fig. S3j)**, the DMT1 isoform 1 (IRE-containing) was highly expressed in MCF7 cells and MDA-MB-468 cells among other isoforms. We also used IRE-specific targets with our previous targets to test the expression of DMT1 under control and *LncRIM* over-expressed MCF7 cells (**Response Fig. 17b**), which showed similar fold changes. Thus, it is reasonable of the change of DMT1 in our study.

The DMT1 antibody used in this study is targeted at the common central domain of these four isoforms, and the most reorganization site is ~70kda in many different cell lines (PMID:35172141, PMID:34732689). We provided a row membrane of the western blot (**Response Fig. 17c**).

There are two promoter regions upstream of DMT1, 51,021,800-51,022,201 and 51,023,800-51,029,201 respectively. Our Chip-PCR result and bioinformatics analysis showed the TEAD/YAP-DMT1 binding region was roughly 51,027,184-51,027,484, which is located at the longer promoter. Further, the luciferase report showed mutated DMT1 promoter lacking this YAP/TEAD-DMT1 binding site significantly attenuated the luciferase activity (**New Fig.S3i**), which further indicated the TEAD/YAP-DMT1 binding site was relative to the DMT1 longer promoter.

Response Fig.17

(a) RT-PCR detection of the four DMT1 isoforms expression in MCF7 cells and MDA-MB-468 cells (mean \pm SD, n=3 independent experiments, *** $P < 0.001$, Student's t -test).

(b) RT-PCR detection of DMT1 expression in control and *LncRIM* expressed MCF7 cells with previous DMT1 primers, or the primers target for DMT1-IRE isoform. (mean \pm SD, n=3 independent experiments; n.s., not significant, *** $P < 0.001$, Two-way ANOVA analysis).

(c) Immunoblot was performed to examine the DMT1 expression of MCF7 cells and MDA-MB-468 cells.

- Fig. 3N,O: data from control cells that have not been treated with IRP2-sh1 and/or *lncRIM* would be needed to appreciate the effect (or absence of effect) of these treatments.

Response: We appreciate the reviewer's suggestion. **Figs.3n, o is now New Fig.3k, l.** We have added the additional control group to repeat this experiment. As shown in **Response Fig.18 (New Figs. 3k, 3l, S3l, S3m)**, IRP2 knockdown decreased the expression of DMT1 and TFR1, however, over-expression of *LncRIM* and YAP still significantly promoted the expression of DMT1 and TFR1 in both mRNA and protein level as well as the cellular iron level under the knockdown of IRP2, suggesting that this *LncRIM*-NF2-DMT1/TFR1 axis acted effectively in a distinct manner of IRP2 system.

Response Fig.18

(a) Immunoblot detection of DMT1 and TFR1 expression in control and IRP2 knockdown MCF7 cells with or without over-expression of *LncRIM* and YAP.

(b) The cellular iron level of control and IRP2 knockdown MCF7 cells with or without over-expression of *LncRIM* and YAP was measured with Calcein-AM assay. (mean \pm SD, n=3 independent experiments; ***P < 0.001, Two-way ANOVA analysis).

(c and d) RT-qPCR detection of TFR1 (c) and DMT1 (d) expression in control and IRP2 knockdown MCF7 cells with or without over-expression of *LncRIM* or YAP (mean \pm SD, n=3 independent experiments; ***P < 0.001, Two-way ANOVA analysis).

- Fig. 3F: it would be good to have an uncropped image of this immunoblot. What is the evidence that this is IRP2? The antibody used is not specified. Also, IRPs are known to be present in excess and the KD of IRP2 seems very weak. A CRISPR-Cas9 approach would certainly be more efficient (if feasible with this cell line).

Response: Thanks for pointing out this issue and we apologize for the non-specificity of the IRP2 antibody used. We have re-constituted the IRP2 knockdown MCF7 cell lines and tested the knockdown efficiency (New Fig. 3k). Besides, we also provided row data in Source Data.

- Fig. S3N: does PLKO express a negative control shRNA?

Response: We apologized for this description! The control groups were treated with control shRNA, and we have revised the "PLKO.1" into "Scramble" in the revised manuscript.

- Fig. S3O: the functionality of the IRE of DMT1 has been questioned. How would IRP2 KD affect DMT1? Also, is DMT1-IRE the isoform expressed in MCF7 cells and quantified in the study?

Response: Thanks for pointing out this issue! **Fig.S3o is now New Fig. S3q.** As shown in **Response Fig.18**, DMT1 isoform1 (IRE-containing) was highly expressed in MCF7 cells and MDA-MB-468 cells. Cellular iron homeostasis previous is reported to be coordinately regulated by the IPR/IRP regulatory network system, which could bind to the conserved cis-regulatory hairpin structures (IRE) of DMT1 and TFR1 to improve their mRNA stability (PMID:20603012, PMID:15109490). And silence of IRP2 dramatically diminished the DMT1-IRE expression (PMID: 20125122, PMID:31092704). Consistently, our data also showed knockdown of IRP2 significantly decreased the expression of DMT1-IRE, while having no impact on the non-IRE DMT1(**Response Fig.19**) (**New Fig. S3k**). More importantly, we demonstrated that DMT1 was the downstream target of YAP, and this *LncRIM*-NF2 axis regulated the expression of DMT1 and TFR1 independently on the IRP2 system (**New Figs.3i-l, S3i**).

Response Fig.19

(a)The expression of DMT1-IRE isoform and non-IRE isoform in control and IRP2 knockdown MCF7 cells was detected by RT-qPCR. (mean \pm SD, n=3 independent experiments, n.s., not significant; ***P < 0.001, Student's *t*-test).

- Fig. S3P adds little value to the study.

Response: Thanks for this kind suggestion! We have deleted this data in the revised manuscript.

- Fig. 4C: how to explain the gaussian effect of FAC? 500 μ M FAC could trigger oxidative stress and be toxic, did the authors perform viability assay?

Response: Thanks for pointing out this issue! As one of the fundamental elements, iron is considered to play important role in the biological process including protein synthesis, cellular respiration, and DNA repair. Iron overload affects cell proliferation and tumor growth by wiring up some signaling pathways such as Wnt/ β signaling and the JNK pathway (PMID:22009536, PMID:21666721, PMID:27546461, PMID:21378396). However, excessive iron also damages DNA and protein by producing reactive oxygen species (ROS) and is increasingly considered as an important mediator of cell death (PMID:24346035, PMID:31063758). And this iron-addicted dependency makes cells

more vulnerable to iron-catalyzed death, which is named ferroptosis (PMID:31105042, PMID:18355723).

Besides, the previous study has shown that YAP is inactive response to cellular apoptosis (PMID: 33086070), and YAP/TAZ deletion increases ROS buildup to promote oxidative stress-induced cell death (PMID:31063758). Thus, we measured the cell viability under FAC stimulation of different concentrations (**Response Fig.20**) (**New Fig. S4d**), and the data suggested higher iron (>400 μ m) decreased cell viability, while lower concentration of iron could promote cell proliferation compared to control cells, which was consistent with our previous immunoblot finding. We have modified the description in the revised manuscript and discussed this.

Response Fig.20

(a) Cell proliferation viability of MCF7 cells with different concentration of FAC stimulation was measured with MTT (mean \pm SD, n=3 independent experiment, **P < 0.01; ***P < 0.001, Two-way ANOVA analysis).

- Fig. 4G: how does FAC affect NF2 expression and pull-down efficiency? Is it equal between FAC and control cells?

Response: Thanks for pointing out this issue! **Fig.4g is now New Fig.4g.** Here, RNA FISH result showed *LncRIM* was colocalized with NF2 on the cell membrane to impair NF2-LATS1 interaction (**Response Figs. 21a, b**) (**New Figs.2e-h**). Moreover, as shown in **Response Fig. 21c**, FAC stimulation had little effect on NF2 expression. As one downstream gene of YAP, iron overload promoted YAP activation and the expression of *LncRIM* (**Figs.4a-c, 4e, 4f**), which further reduced the LATS1-NF2 association (**New Fig.4h**). Consistently, the immunofluorescent result also showed that iron overload significantly increased the expression of *LncRIM* on the membrane and the interaction between *LncRIM* and NF2, while decreasing LATS1 membrane association. Meanwhile, the NF2 expression on the membrane was hardly affected (**Response Figs.21d, e**) (**New Figs. S4f, g**).

Response Fig. 21

(a and b) *LncRIM* RNA FISH was performed in MCF7 cells. Line scan of the relative fluorescence intensity of the signal (dotted line; left) is plotted to show the peak overlapping (right). The *LncRIM* probe was labeled with Cy3(Red) and the NF2 was detected with Alexa Fluor 488. Scale bar, 10 μ m.

(c) Immunoblot detection of NF2 expression under FAC stimulation of different concentration for 24hr.

(d and e) Immunofluorescence staining was performed to assess the interaction between LATS1 and NF2 in MCF7 cells expressing GFP-NF2 with or without FAC stimulation (200 μ M) for 24hr. Line scan of the relative fluorescence intensity of the signal (d) is plotted to show the peak overlapping (e). The *LncRIM* probe was labeled with Cy3, and the LATS1 was detected with Alexa Fluor 647. Scale bar, 20 μ m.

- Fig. 4I,J and lane 301: ambiguous phrasing, *lncRIM* overexpression has no effect on FTH1, the effect seems to be purely the one of FAC.

Response: Thanks for pointing out this issue! **Fig.4i, j is now New Fig.4i, j.** Once shuttled into the cytoplasm, iron can be immediately used or stored in ferritin, which is made up of two subunits of FTH1 and FTL. FTH1 can be used as an indicator of cellular iron and contributes to cancer growth (PMID:26461092). Here, we demonstrated that *LncRIM* affected basal cellular iron levels by regulating the expression of DMT1 and TFR1 while having little impact on the expression of FTH1 and FTL (**New Figs.1k,1l,1p, S1i**)

Cellular iron homeostasis balance is achieved by the expression of proteins related to iron uptake, iron storage, iron utilization, and iron efflux. Knockdown of *LncRIM* significantly decreased the expression of the iron-uptake protein, DMT1, and TFR1, thus the formation of ferritin would be reduced compared to control cells under iron stimulation. And we have changed the description in the revised manuscript. Thank you again for pointing out this issue to better help us revise our paper.

- Fig. S4E,F: in terms of fold change, *lncRIM* seems to have the same effect regardless of the presence or absence of FAC, especially in MCF7 cells, hence FAC does not alter the response of the cell to *lncRIM* fluctuation.

Response: Thanks for pointing out this issue! **Fig. S4e, f is now New Figs. S4i, j.** We demonstrated that iron-trigger *LncRIM*, one downstream gene of YAP, regulated the cellular iron levels by coordinating with the Hippo pathway (**New Fig. 1g, 1h, 1k, 3c-e, 4c-f**). And this *LncRIM*-NF2 axis in turn facilitated iron-mediated Hippo-YAP feedback loop signaling (**New Figs. 4i, 4j**). Interestingly, we found iron overload enhanced the expression of YAP downstream genes required longer stimulation than activating YAP (**Response Fig.22a**) (**New Fig. S4h**), which may be due to the signaling pathway cascade from upstream NF2 to downstream targets, and in turn active *LncRIM*-NF2 axis to play positive feedback regulation. We then stimulated the MCF7 cells and MDA-MB-468 cells with FAC for 48hr. The result showed that *LncRIM*-NF2 axis further enhanced the expression of *CTGF*, *CYR61* with iron stimulation upon *LncRIM* over-expression or attenuated the inhibition effect in *LncRIM* knockdown cells (**Response Figs.22b, c**) (**New Figs. S4i, S4j**).

Response Fig.22

(a) RT-qPCR was performed to examine the expression of YAP downstream targets *CTGF*, *CYR61* with FAC stimulation for different time point in MCF7 cells (mean \pm SD, n=3 independent experiment; **P < 0.01; ***P < 0.001, Student's *t*-test)

(b and c) The expression of YAP downstream targets *CTGF*, *CYR61* was detected by RT-qPCR in control and *LncRIM* over-expressed or *LncRIM* knockdown MCF7 cells and MDA-MB-468 cells with or without FAC stimulation for 48hr. (mean \pm SD, n=3 independent experiment; **P < 0.01; ***P < 0.001, One-way ANOVA analysis).

- Lane 306: " to promote cell proliferation". Cell proliferation has not been assayed in Figures 4 and S4. End the sentence at "Hippo-YAP signalling").

Response: Thanks for pointing out this issue! We performed a colony formation assay of control and YAP-silenced MCF7 cells with FAC stimulation (200 μ M). And the result showed that *LncRIM*-NF2 axis hyperactivated the YAP to promote cell proliferation (**Response Fig.23**) (**New Fig. S5a, S5b**).

Response Fig.23

(a and b) Colony number assay in control or YAP-silenced MCF7 cells under the FAC stimulation. (mean \pm SD, n = 3 biologically independent experiments, n.s. not significant; ***P < 0.001, Student's *t*-test).

- Fig. 5D: iron staining is hard to see without DAB enhancement.

Response: Thanks for pointing out this issue! We have done the biological replicates with enhanced DAB iron staining based on previous work (**Response Fig.24**) (**New Fig. 5d**) (PMID:33895792). The result also demonstrated that knockdown of *LncRIM* significantly suppressed cellular iron levels.

Response Fig. 24

(a and b) Enhanced DAB iron staining of randomly selected tumors from mice subcutaneously injected with the indicated stably transduced MDA-MB-468 cell lines (a). Scale bar, 100 μ m. The relative intensities of IHC were quantified by ImageJ(b) (Fiji version 1.51 software) (mean \pm SD, n=3 independent experiment; ***P < 0.001, Student's *t*-test).

- Fig. 5F: expression changes in a bulk analysis of whole tumor samples could reflect changes in the cellular composition of the tissue, as opposed to changes in the cancer cells themselves. For example, the vascularization of the tissue seems to differ between control and *LncRIM*-KD tumors, and one cannot exclude that vascular cells contribute some of the western blot signals in 4F. Tissue staining would help to discriminate, if doable.

Response: We appreciate the reviewer's suggestion! We have done the IHC staining of DMT1 and TFR1(**Response Fig.25**) (**New Figs. S5c, S5d**) in the revised manuscript. The result showed that knockdown of *LncRIM* dramatically decreased the expression of DMT1 and TFR1, which was consistent with the previous immunoblot image (**New Fig.5f**).

Response Fig. 25

(a and b) IHC staining of DMT1 and TFR1 randomly selected tumors from mice subcutaneously injected with the indicated stably transduced MDA-MB-468 cells (a). Scale bar, 100 μ m. The relative intensities of IHC were quantified by ImageJ (b) (Fiji version 1.51 software) (mean \pm SD, n=3 independent experiment; ***P < 0.001, Student's *t*-test).

- Fig. 5H-I: cancer cells need iron to grow and as expected DFO decreases tumorigenesis regardless of whether *lncRIM* is overexpressed or not. These data do not show specifically that iron mediates the effect of *lncRIM* overexpression of tumorigenesis. A KD of *TFRC* and/or *DMT1* would be more specific. Furthermore, DFO could have multiple effects for example by altering the tumor immune microenvironment.

Response: Thanks for this constructive suggestion! We have constructed *LncRIM* over-expressed, double knockdown of *TFR1* and *DMT1* with over-expression of *LncRIM* cell lines. And then carried out orthotopic injection within nude mice. As shown in **Response Figs. 26a-e (New Figs. 5h-k, S5j)**, double knockdown of *DMT1/TFR1* dramatically diminished *LncRIM*-mediated tumor growth, the *YAP* activation, and cellular iron level. Besides, the colony formation assay also showed the same finding (**Response Figs. 26f, g (New Figs. S5k, S5l)**). Together, this result to some extent further proved our idea that *LncRIM* promoted cell proliferation and tumor progression via regulating cellular iron metabolism.

Response Fig.26

(a-c) Nude mice were injected with control, *LncRIM* over-expressed, and double knockdown of DMT1 and TFR1 with over-expression of *LncRIM* MDA-MB-468 cell lines (a) Tumor volumes (b) and tumor weights (c) were assessed. The data are presented as the mean ± SD from n = 5 mice per group. (***)P < 0.001, Two-way/One-way ANOVA analysis).

(d and e) Representative IHC staining and enhanced DAB iron staining of randomly selected tumors from mice subcutaneously injected with the indicated stably transduced MDA-MB-468 cell lines (d). Scale bar, 100 μm. The relative intensities were quantified by ImageJ (Fiji version 1.51 software) (e). The data are presented as the mean ± s.d. from n = 5 xenograft tumor samples per group. (n.s., not significant; ***P < 0.001, One-way ANOVA analysis).

(f and g) Colony formation assay in control, *LncRIM* over-expressed and double knockdown of DMT1 and TFR1 with over-expression of *LncRIM* MCF7 cell lines (f). The data are presented as the mean ± s.d. (n = 3 biological independent experiments, ***P < 0.001, Students' t-test).

- Fig. 5K: Iron staining hard to visualize.

Response: Thanks for pointing out this issue! We did the biological repeats (**Response Fig.27**) (**New Fig.S5g**) with enhanced DAB iron staining.

Response Fig. 27

(a and b) Enhanced DAB iron staining of randomly selected tumors from mice subcutaneously injected with the indicated stably transduced MDA-MB-468 cells (a). Scale bar, 100 μ m. The relative intensities of IHC were quantified by ImageJ(b) (Fiji version 1.51 software) (mean \pm SD, n=3 independent experiment; ***P < 0.001, One-way ANOVA analysis).

- Fig. 6A, Perl's stain: what is the evidence that the iron signal is in tumor cells and not in immune cells. Fig. 6A does not give any information about the LIP, in contrast to what is stated in lane 340 (Perl's staining reveals mostly non-heme iron stored in ferritin degradation products).

Response: Thanks for pointing out this issue! As shown in **Response Fig.28**(**New Fig.6a**), double staining of Perl's blue and CD45, one normal indicator of immune cells, showed that the increased iron was mainly located in the breast cancer cells. We have added this datum in the revised manuscript.

Response Fig. 28

(a) Double staining of iron and CD45 shows cancer cells englobed iron. Scale bar, 100 μ m.

- lane 347,348: there is some correlation between lncRIM expression, TFR1/DMT1 expression and tumor status, but no formal evidence that lncRIM promotes tumor development or aggressiveness by disturbing iron metabolism. This shall be rephrased.

Response: Thanks for pointing out this issue! We came to the conclusion that *LncRIM*

promoted cell proliferation and breast cancer development by disturbing iron metabolism through many *in vivo* and *in vitro* experiments.

Firstly, we found that FAC stimulation could partially restore cell proliferation and cell cycle arrest in *LncRIM*-silenced cells (New Figs.1m, S1k, S1l). And over-expression of *LncRIM* partially diminished the inhibition of DFO on cell proliferation, tumor growth and the cellular iron level (New Figs. 1n, S1m, S5e-h), indicating iron homeostasis was important for *LncRIM*-mediated cell proliferation and tumor growth.

Here, we demonstrated that *LncRIM* regulated cellular iron metabolism via DMT1 and TFR1 (New Fig.1p). Further, mouse model showed that double knockdown of DMT1 and TFR1 dramatically decreased *LncRIM*-induced cell proliferation, tumor growth and cellular iron level indicated by the expression of YAP, Ki67 and enhanced DAB iron staining (New Figs.5h-k, S5k, S5l). Together, we came to the conclusion that *LncRIM* wired up the Hippo-YAP pathway to promote breast cancer development by regulating cellular iron metabolism. We have revised the description in the revised manuscript to make it clear.

Minor points.

- Lane 81: in Ref 15, iron loading does not activate but rather inhibits Wnt signaling.

Response: We are sorry for this description, and we have corrected this in the revised manuscript.

- Lane 43: “the underlying mechanism has not yet been elucidated”. There is already abundant literature on the topic, would be better to say “remains ill understood” or equivalent rather than “not elucidated”.

Response: Thanks for this kind suggestion! We have replaced the “*has not yet been elucidated*” with “*remain poorly understood*” in our revised manuscript.

- Iron 2+ and 3+ do not exist in biological systems, the valency state of iron shall be written as Fe (II) and Fe (III). Same for Fig. 2R and 6I.

Response: Thanks for this kind suggestion! We have corrected this description in our revised manuscript accordingly.

- Calcein assay: for the non-expert reader, it would be helpful to specify (when the assay is mentioned for the first time) that the fluorescence signal is inversely correlated with the size of the LIP.

Response: We appreciated this kind suggestion! We have added a detailed description of “*one widely used way to measure cellular iron level, with the fluorescence signal of*”

calcein is inversely correlated with the level of iron" in the text.

- lane 140: "to confirm the regulatory role of these lncRNAs in iron metabolism". Not to "confirm" but to assess.

Response: Thanks for this kind suggestion! We have changed the "confirm" into "assess" in the revised manuscript.

- lane 144: explain CAPT + what has been done is polysome profiling, not ribosome profiling.

Response: Thanks for this suggestion! We have changed the "ribosome profiling" into "polysome profiling" in the revised manuscript.

- lane 145: "inability to encode micropeptides" is a bit extreme as it could be a matter of sensitivity and/or context.

Response: We appreciate this kind suggestion! We changed the description of "inability to encode micropeptides" into "little ability of *LncRIM* in encoding protein".

- lane 149: end of the sentence, specify "observed in multiple breast cancer CELL LINES".

Response: Thanks for this kind suggestion! we have added "cell lines" at the end of our revised manuscript.

- Fig 1Q: this figure is superfluous since the data is not really exploited

Response: Thanks for pointing out this issue! For many lncRNAs, their function and the specific mechanism are related to their subcellular location (PMID:24967943, PMID:23498938). lncRNAs located at the nucleus are usually involved in the chromatin function and the function of membraneless nuclear bodies (PMID:33420484). And previous studies including ours have revealed multiple cytoplasmic lncRNA that can mediate organelle interaction, signaling transduction, and cellular metabolism (PMID:30220561, PMID:33398195, PMID:33953175). Thus, in the previous **Fig.1q**, we were interested in testing the cellular location of *LncRIM* for further study. We also performed RNA FISH to further validate the specific membrane location of *LncRIM* (**New Figs.2e, 2f**).

- lane 206: typo: "To further ANALYSE"2

Response: We appreciate this kind suggestion! We have corrected this in the revised manuscript.

lane 207-208: "constructed three different deletion mutants" (omit "fragments").

Response: Thanks for pointing out this issue! We corrected the description to “constructed three different deletion mutant fragments”

- Fig.6B: typo y-axis "Specomens".

Response: We appreciate your kind suggestion! We have corrected this into “Specimens” in the revised manuscript.

- did the authors assess the status of STEAP proteins? Are they also stimulated by YAP?

Response: Thanks for pointing out this issue! STEAP family is also critical for maintaining iron homeostasis (PMID:16609065). In the endosome, iron is freed to Fe (II) by STEAP and then released to the cytosol via DMT1. We examined the expression of STEAP in MCF7 cells stably expressing vector, YAP active mutant (5SA-YAP). The data showed that active YAP (5SA-YAP) slightly upregulates the expression of STEAP (**Response Fig.29a**). However, knockdown of *LncRIM* had no impact on the expression of STEAP (**Response Fig.29b**), suggesting that the regulation of YAP on STEAP may independent on our *LncRIM*-NF2 axis.

Response Fig. 29

(a) RT-qPCR detection of *STEAP* and *CTGF* in control and YAP-5SA over-expressed MCF7 cells. (mean \pm SD, n=3 independent experiment; **P < 0.01; ***P < 0.001, Student's *t*-test)

(b) RT-qPCR detection of *STEAP* in control and *LncRIM* knockdown MCF7 cells. (mean \pm SD, n=3 independent experiment; n.s., not significant, Student's *t*-test).

Reviewer #3 (Remarks to the Author):

In the present study, the authors have reported the role of a novel lncRNA; *lncRIM*, in YAP-mediated iron metabolism. The authors demonstrated a positive correlation between YAP levels and intracellular iron levels in cell lines and breast cancer tissue samples. *lncRIM* is a target of the YAP/TEAD activator complex. *lncRIM* binds to NF2 on the membrane and prevents the interaction between NF2 and LATS1 kinase, thereby abolishing the LATS1-mediated phosphorylation and degradation of YAP. In this way, *lncRIM* promotes the nuclear translocation of YAP, thereby promoting YAP-mediated transcription activation of genes controlling iron metabolism.

In general, this is an exhaustive study. Most of the experiments were conducted in breast cancer and kidney cancer cells, and the data is supported by experiments performed in mouse xenograft models as well in cancer patient tissue samples. The major drawback of the ms is that the authors have shown some sort of bias to connect

the lncRIM and YAP pathway without performing an unbiased screen (see the details below). Also, several of the immunoblots seems to be modified (at least based on the pdf files that I have downloaded).

Response: We are very grateful for your recognition of our work, and thank you for these creative suggestions, which will help us to better improve our research. We have revised manuscript as you suggested.

Specific comments

What was the basis for the authors to select only a subset of lncRNAs for RT-qPCR experiments shown in fig 1f? Did they perform an RNA-seq to identify the candidates initially, and then validate the top hits by RT-qPCR?

Response: We apologized for this confusing description. **Fig.1f is now New Fig. 1f.** In this study, we uncovered one novel mechanism in which oncogenic lncRNA *LncRIM* regulated cellular iron metabolism dependent on the Hippo-YAP pathway step by step through a series of unbiased bioinformatic screening and experimental analysis. Following is the detailed procedure:

- Firstly, by using the GSEA analysis (GEO number: GSE38369) (PMID:22767506), Calcein-AM assay, and clinical breast cancer tissue (**New Figs.1a-e, S1a**), we found the potential link between the Hippo pathway and cellular iron metabolism.
- Then, our previous study (PMID:28114269) have pinpointed 40 lncRNAs that were potentially required for YAP-dependent transcription by transfecting the human Lincode® siRNA library into MCF7 cells that were engineered with a TEAD-driven luciferase reporter. By overlapping this RNA-seq database and FAC/DFO stimulation, we obtained a cytoplasmic lncRNA *Loc729013*, re-named *LncRIM* (LncRNA Related to Iron Metabolism), which obviously responded to alteration of iron concentration and also regulated cellular iron level among others (**New Figs.1f-h, 1k,1l, S1h-j**). And we also found *LncRIM* regulated the YAP downstream targets (**New Fig.S1q**).
- Furthermore, we also confirmed that *LncRIM* directly bound NF2 to impair NF2-LATS1 interaction (**New Fig.2**), and regulated the expression of DMT1 and TFR1 dependent on the Hippo-YAP pathway (**New Fig. 3**).

We have explained it clearly in the revised manuscript. Thanks again for pointing out this issue!

Did the experiment detail in fig 1h perform in presence of FAC? Since lncRIM is upregulated in FAC-treated cells, the iron-metabolism assay should be done under similar conditions.

Response: Thanks for pointing out this issue! **Fig.1h is now New Fig.1h.** In the **New Fig.1f, 1g,** we have screened out 6 promising Hippo-YAP related lncRNAs that responded to cellular iron level, which indicated the potential role of these lncRNAs in turn regulate cellular iron metabolism. To further test this hypothesis, we then performed the Calcein-AM assay with siRNA targeted for these 6 lncRNAs in MCF7 cells, where knockdown of lncRNA *Loc729013*, which is renamed as *LncRIM*, robustly decreased the level of basal cellular iron among others (**New Figs.1h, 1k, 1l, S1h-j**). Consistent with this, knockdown of *LncRIM* also dramatically decreased the iron level in the tumor as shown in **New Fig. 5d**. We have clearly described this in the revised manuscript.

what is the copy number of lncRIM? Authors argued that lncRIM by binding to NF2 quenches NF2 from interacting with LATS1. In order for this to happen, the copy number lncRIM should be comparable to the NF2 protein levels. This needs to be estimated.

Response: Thanks for this experimental advice. For the copy number of *LncRIM*, we have determined that there are roughly 690 copies of the *LncRIM* per MCF7 cell and 502 copies per MDA-MB-468 cell (**Response Fig.30a, b**) (**New Fig. S2b, S2c**), which was of relatively high abundance compared with that of several known functional lncRNAs: *LINK-A* has roughly 150 copies per MDA-MB-231 cell and *CamK-A* has roughly 937 per MDA-MB-231 cell (PMID:28218907, PMID:30220561). Also, we have carefully revised our manuscript by adding current knowledge about *LncRIM* and new experiments to address the reviewers' concerns.

For the model of competition: NF2 molecule number per MCF7 cell was determined as around 830 thousand (**Response Fig.30c, d**), we concluded that *LncRIM*/NF2 ratio in the membrane was about 1: 1200 in normal conditions and about 1:400 under FAC stimulation with the expression of *LncRIM* increased about 3-fold changes (**New Fig. S4f**). Besides, we also showed that the approximately 1:1 molar ratio of *LncRIM* (25nM) and NF2 (20nM) was sufficient to interfere with the NF2-LATS1 complex (**New Figs. S2j, Sk**).

Combining with the experimental evidence above, we thought *LncRIM* could be sufficient to function with NF2 in reasonable effect size in cellular iron metabolism and cancer progression.

Response Fig. 30

(a and b) *LncRIM* copy number was determined in MCF7 and MDA-MB-468 cell lines(a). Data are presented as mean values \pm SD, n=3 biologically independent experiments (b).

(c and d) The absolute concentration of the purified GST-NF2(10 μ l) protein was determined by Coomassie staining with BSA standard curve (0.1-2.0 μ g)(c). Around five million cells (MCF7, MCF10A separately) were collected by cell counting and 10% of them was input to SDS-PAGE gels for anti-NF2 immunoblot detection. NF2 Standard curve (0.005-0.100 μ g) was used to determine the endogenous NF2 mass and the corresponding molecule number per cell by measuring the bands' relative grey values (d).

Also, wherein the cytoplasm does *LncRIM* localize? Based on the model, it should co-localize with NF2 on the membrane.

Response: We thank the reviewer for this suggestion! As shown in **New Fig.1q**, we validated that *LncRIM* was mostly located in the cytoplasm. Then, we designed the probe to precisely validate the accurate location of *LncRIM*. As seen in **Response Fig. 31 (New Figs. 2e, f)**, RNA fluorescence in situ hybridization (RNA FISH) data showed the *LncRIM* was co-localized with NF2 on the cell membrane in MCF7 cells.

Response Fig. 31

(a and b) *LncRIM* RNA FISH and immunofluorescence staining of endogenous NF2 were performed in MCF7 cells. Line scan of the relative fluorescence intensity of the signal (dotted line; left) is plotted to show the peak overlapping (right). The *LncRIM* probe was labeled with Cy3 and the NF2 was detected with Alexa Fluor 488. Scale bar, 10 μ m.

Data presented in fig 2g should be quantified and also should be shown in the case of endogenous proteins. In addition, I see the reduced signal of both tagged-NF2 and LATS1 on the membrane in the absence of *LncRIM*. Does *LncRIM* also facilitate the

localization of NF2 on the membrane? Again, in this context, it is important to show the localization of LncRIM in the cell.

Response: Thanks for this constructive suggestion! **Fig.2g is now New Figs.2j.** We are sorry for the confusion in previous Fig.2g, which may be due to the different transfection efficiency of GFP-NF2. We have repeated this experiment in MCF7 cells (**Response Fig.32a, b**) and quantified the data of F_{LATS1}/F_{NF2} . We apologized for not finding suitable antibodies of endogenous NF2 and LATS1 with different fluorescent secondary antibodies. Thus, we tested the endogenous NF2-LATS1 interaction in *LncRIM* over-expressed MCF7 cells by using subcellular fractionation (**Response Fig.32c**) (**New Fig.S2i**).

Besides, our data showed that over-expression of *LncRIM* had no impact on NF2 membrane localization (**Response Fig.32d, e**) (**New Figs. S2d, e**). Further, iron stimulation increased the expression of *LncRIM* on membrane and the interaction between *LncRIM* and NF2, while further decreasing LATS1 membrane interaction (**Response Fig. 32 f, g**) (**New Figs.S4f, S4g**). Again, thank you for these suggestions which help better improve our revised manuscript.

Response Fig. 32

(a and b) Immunofluorescence staining was performed in MCF7 cells expressing GFP-NF2 and pRFP-LATS1 with or without over-expression of *LncRIM*. The data was quantified with Image J. (b) (mean ± SD, n=5 independent experiment; ***P < 0.001, Student's *t*-test).

(c) Subcellular fractionation was performed to assess endogenous NF2 and LATS1 expression in control and *LncRIM* over-expressed MCF7 cells.

(d and e) Immunofluorescence staining was performed to assess the endogenous NF2 membrane location in control and *LncRIM* over-expressed MCF7 cells. The data was quantified with Image J. (mean ± SD, n=5 independent experiment; n.s., not significant, Student's *t*-test).

(f and g) *LncRIM* RNA FISH and immunofluorescence staining were performed in MCF7 cells expressing GFP-NF2 with or without FAC stimulation (200μM). Line scan of the relative fluorescence intensity of the signal (f) is plotted to show the peak overlapping (g). The *LncRIM* probe was labeled with Cy3 and the LATS1 was detected with Alexa Fluor 647. Scale bar, 20 μm.

Initial data indicates that both YAP and *lncRIM* positively influence the iron levels in BC cells. However, it is not clear to me why authors have presumed that both these molecules play in the same pathway, and therefore have decided to identify *lncRIM* interacting proteins that are part of the HIPPO pathway. This seems to me like a biased approach without a strong rationale.

Ideally, *lncRIM* pull-down followed by mass spec should be conducted to identify the top hits.

Response: Thanks for this pointing out this issue! In this study, we uncovered one novel oncogenic *lncRIM*-mediated mechanism of cellular iron metabolism. And through a large number of unbiased bioinformatics screening and *in vitro* and *in vivo* experiments, we gradually revealed the crosstalk between *lncRIM*, Hippo signaling pathway, and cellular iron metabolism in breast cancer progression. Following is the detailed procedure:

- Firstly, by using the GSEA analysis, Calcein-AM assay, and IHC of clinical breast cancer tissue (**New Figs. 1a-e, S1a**), we found YAP was significantly inactive under the deregulation of cellular iron levels, and active YAP obviously affected cellular iron level, suggesting the potential link between the Hippo pathway and cellular iron metabolism.
- Then, in our previous study (PMID:28114269), we have pinpointed 40 lncRNAs that were potentially required for YAP-dependent transcription by transfecting the human Lincode® siRNA library into MCF7 cells that were engineered with a TEAD-driven luciferase reporter. By overlapping from this RNA-seq database and FAC/DFO stimulation, we unbiasedly obtained one cytoplasmic lncRNA *Loc729013*, re-named *lncRIM* (lncRNA Related to Iron Metabolism), which obviously responded to alteration of iron concentration and regulated cellular iron level among others (**New Figs.1f-h, 1k,1l, S1h**). And we also found *lncRIM* regulated the YAP downstream targets and the expression of DMT1 and TFR1 (**New Figs. 1p, S1o-q**).
- To further clarify the connection between cellular *lncRIM*-mediated iron metabolism and the Hippo pathway, we then performed RNA pull-down, and found *lncRIM* only directly bound to NF2 not other molecules of the Hippo pathway (LATS1, MOB1, MST1, YAP). And this *lncRIM*-NF2 axis significantly inhibited NF2-LATS1 interaction (**New Figs.2, S2a**).
- Moreover, we also demonstrated that *lncRIM*-NF2 axis regulated cellular iron metabolism dependent on the Hippo-YAP pathway, and DMT1 and TFR1 are both the target genes of YAP (**New Figs. 3c-k, S3i**).

Overall, we came to this conclusion step by step that iron-trigger lncRNA *lncRIM* regulated cellular iron metabolism dependent on the Hippo-YAP pathway

effectively, in a distinct manner of IRP2 system. We have made it clear in the revised manuscript. Thanks again for pointing out this issue!

There seem to be some undesirable modifications done on several of the immunoblots presented in the ms [some examples include Figs 2e (WCL:IB: HA) & f (IP: IB: NF2) and Fig S2b (IP-flag: IB: HA), Figs. S3d, f].

Response: We apologize for this confusion! **Figs.2e, 2f is now New Figs.2g, 2h. Fig.S2b is now New Fig.S2f. And Figs.S3d, S3f is now New Figs.S3f, 3k.** We have provided the raw data of these immunoblots in **Source data**.

It is not clear why some of the experiments were done in HEK293T cells when the main focus of the paper is to understand the role of *lncRIM* in iron metabolism in breast cancer cells.

Response: Thanks for pointing out this issue! For the previous **Figs. 1b, c, 2g, 2h, 3k**, we have done more biological repeats in MCF7 cells (**New Figs.1b, 1c, 2i-k, S3i**) and added these data in the revised manuscript. Previous studies have shown that molecules of the Hippo-YAP pathway play the same biological regulation and mechanism in HEK293 cells as in cancer cells (PMID:24012335, PMID:17974916, PMID:26045165). Meanwhile, in HEK293 cells, the Hippo-YAP pathway also responds to the various upstream metabolic signals and presents a similar change in cancer cells (PMID:29100056, PMID:30472188, PMID:22863277).

We demonstrated that *LncRIM* inhibited NF2-LATS1 association in both HEK293T and MCF7 cells (**New Fig. 2g, 2h, S2f**), and verified that DMT1 was the downstream target of YAP in these two cell lines (**New Fig.3i, 3j, S3c-e, S3i**). Moreover, iron stimulation enhanced the role of *LncRIM*-NF2 axis in inhibiting LATS1-NF2 association and further promoted the interaction between *LncRIM* and NF2 (**New Figs.4e-h, S4f, S4g**), which presented the same finding in both HEK293T cells and MCF7 cells. Thus, we thought it was reasonable to do some auxiliary experiments in HEK293T cells to further support our findings.

Does KD of *lncRIM* enhance NF2-LATS1 interaction?

Response: Thanks for this experimental advice! As seen in the **Response Fig.33 (New Fig. S2h)**, knockdown of *LncRIM* significantly increased the interaction between NF2 and LATS1, suggesting the important role of *LncRIM* in regulating NF2-LATS1 interaction.

Response Fig. 33

(a) An endogenous coimmunoprecipitation (Co-IP) by using IgG and NF2 antibodies was performed to test NF2 and LATS1 interaction in control and *LncRIM* knockdown MCF7 cells. The NF2 antibody was pulled down with Protein A/G beads. IgG was used as the negative control.

fig 1B shows changes in the expression of iron metabolism genes in *lncRIM*-depleted cells. If *lncRIM*-depleted cells show such a dramatic increase in YAP phosphorylation, ultimately resulting in the inactivation of YAP, then one would expect reduced expression of a significant number of YAP target genes. Is that the case?

Response: Thanks for pointing out this issue! As shown in **Response Fig. 34 (New Fig. S1q)**, knockdown of *LncRIM* significantly decreased the expression of YAP downstream targets. We have added this data to the revised manuscript. Thanks again for this advice!

Response Fig. 34

(a) RT-qPCR detection of the expression of YAP downstream targets in control and *LncRIM*-silenced MCF7 cells.

The model presented in fig 2r should be demonstrated by in vitro analyses. authors should test the binding kinetics of purified NF2 and LATS with varying concentrations of *lncRIM*.

Response: We appreciate the reviewer's constructive suggestions! The N-terminus of LATS1 (LATS1-NT) contributes to the binding to NF2 (PMID:26045165). By using purified GST-LATS1-NT and MBP-His-NF2 proteins, we performed GST-pull down to assess the binding kinetics of NF2 and LATS1 under different concentration of in vitro transcribed biotinylated *LncRIM* (**Response Fig. 35**) (**New Figs. S2j, S2k**). And the result also showed that *LncRIM* significantly inhibited the interaction between LATS1 and NF2 in a dose-dependent. We appreciated the reviewer's kind suggestion and provided this data as suggested.

Response Fig. 35

(a and b) GST-pull down was performed to test the interaction between LATS1 and NF2 under different concentration of in vitro-transcribed *LncRIM* by using 20nmol purified GST-LATS1-NT and 20nmol His-MBP-NF2 proteins.

Fig 1F panel wrote MCF7, whereas the text in the result section indicates HEK293T.

Response: We are sorry for this writing and thanks for pointing out this issue! Indeed, we screened out the potential lncRNAs responded to iron stimulation in MCF7 cells. We have revised this in the revised manuscript.

YAP shRNAs do not seem to efficiently deplete YAP (Fig s3a). Also, there seems to be some issue with the YAP-IB.

Response: Thanks for pointing out this issue! We have re-constructed the YAP knockdown cell lines for two different shRNA in MCF7 cells (**New Fig. 3c**). Also, we provided the raw data in the **Source data**.

Fig s3b: How do the control and YAP-depleted cells show similar levels of CTGF and CYR61 mRNAs?

Response: Thanks for pointing out this issue! We apologized for the quality of previous data. Herein, *LncRIM* was proved to regulate the expression of DMT1 and TFR1 depending on the Hippo-YAP pathway, and DMT1 and TFR1 both were downstream targets of YAP (**New Figs.3c-e, 3i, 3j, S3i**). Besides, *LncRIM* also regulated YAP activation and downstream targets (**New Figs.S1q, 3a, 3b**). Moreover, re-expression of YAP significantly restored the expression of DMT1 and TFR1, and the cellular iron level to a similar level to that of *LncRIM* over-expressed so as the cellular iron level (**New Figs.3g, 3h**). Consistent with this, the level of *CTGF* and *CYR61* also present a similar change. We have repeated this experiment by using effective YAP knockdown cell lines (**Response Fig. 36**) (**New Figs. 3g, 3h, S3a, S3b**).

Response Fig. 36

(a and b) Immunoblot and Calcein-AM assay were performed to examine the expression of DMT1 and TFR1, and cellular iron level in control and YAP-silenced MCF7 cell lines with over-expression of *LncRIM* or YAP. (mean \pm SD, n=3 independent experiment; n.s., not significant, ***P < 0.001, One-way ANOVA analysis).

(c and d) RT-qPCR detection of *CYR61*, *CTGF* in control and YAP-silenced MCF7 cell lines with over-expression of *LncRIM* or YAP. (mean \pm SD, n=3 independent experiment; n.s., not significant, ***P < 0.001, One-way ANOVA analysis).

REVIEWER COMMENTS

Reviewer #1 (Remarks to the Author):

The authors have adequately addressed to the concerns raised previously by this reviewer. Based on this the manuscript is now suitable for publication in Cell Death and Disease.

Reviewer #2 (Remarks to the Author):

It is sometimes difficult to navigate through the huge rebuttal that the authors have generated. Changes in the text have been made and those are not highlighted. On the opposite, changes mentioned in the rebuttal are not highlighted in the text and are hard to find. In any case, the authors have responded to most of my comments in a satisfactory manner. Some clarification is still needed, as detailed below.

Rebuttal page 8:

The KD of TFR1 and DMT1 does not reduce tumor growth, colony formation, and ki67 expression to the level observed in EV cells. This simply reflects the essentiality of the nutrient iron for cell growth in general, independently of LncRIM. Yet, the TFR1/DMT1 KD approach is more specific than DFO treatment and is a good addition to the manuscript.

Figure 5h-k:

- I guess there is a color inversion in the legend to Fig. 5i (LncRIM vs LncRIM-double KD).
- indicate the statistical significance on the figure (as done in 5b).
- verify typos (space) for the Y-axis titles.

It would be good to have inserts with magnification of the signal in 5j. This would help the reader appreciate for example whether TFR1 is localized at the plasma membrane or DMT1 in endosomes, as expected.

Rebuttal page 10 - Fe staining.

I find it still difficult to appreciate the location of the iron signal in the tissue. This may be due to the resolution of the PDF. Inserts with iron magnification could be added to 5d,j and S5g. Hopefully, high-resolution images allowing deep zooming will be provided with the article.

Rebuttal page 10, 11 - Role of MAYA/YAP in iron metabolism in the context of NAFLD.

The answer to my comment is convoluted and hard to understand. I was simply asking to refer to the work by Yuan et al. in the text, as it would be interesting for the reader to realize that YAP and iron could be interconnected in different ways depending on context.

Rebuttal page 13,14 - iron in microenvironment cells:

The data in response Fig. 8 are very intriguing, and suggest that the iron status of the cancer cell shapes the macrophage microenvironment, and/or vice versa. Very interesting, but my question was simply about the localization of iron in Fig. 1d (or new Fig. 1d).

Rebutal page 14 – interpretation Fig.1m,n (Response Fig. 9):

- The authors say they “re-analyzed” the data in Fig.1m. The data now look different from what they were in the first version of the manuscript, and differences between FAC and vehicle treated cells become statistically significant! What has changed?
- The interpretation of figure 1n is still uncertain. What matters is if the fold change in DFO vs vehicle-treated cells is significantly different when LncRIM is overexpressed (visually the effect of DFO seems slightly attenuated but is it statistically significant?).
- New Fig. S1l (cell cycle analysis). There is no indication of whether the differences observed are statistically significant or not.

Rebuttal page 15: western blots in former Fig. S1o,p.

Fig.S1o,p is not new Fig.S1q as stated by the author but it is Fig. S1p!

The authors said they did statistical analyses. I can see ratio indicating they have QUANTIFIED the western blot signals, but I don't see anything related to STATISTICAL SIGNIFICANCE. Please clarify.

Rebuttal page 19 - Chip-seq data DMT1 (Response Fig. 15):

The authors state they added the GEO dataset reference to the figure legend but I don't see it.

Rebuttal page 20 - DMT1 luciferase reporter assays (response Fig. 16):

Fig. 3i alone is not convincing, as commented in my previous review. Fig. S3i is more convincing. How was the mutant generated exactly? How was basal expression of the reporter affected by the mutation? Was it in a similar range for the two types of constructs?

Rebuttal page 20,21 - DMT1 isoforms (response Fig. 17):

Figure S3j,k: This type of comparison relies on distinct qPCR assays. For it to be valid, one needs to make sure cDNA synthesis is equal between isoforms (e.g. no GC rich structure impairing RT in one isoform) and that primer efficiency is the same (or corrected for). Was this the case?

Rebuttal page 21,22:

Response Fig. 18 shows a nice piece of data. Yet, the induction of TFR1 and DMT1 by IncRIM and YAP seems partially blunted in IRP2-KD cells. As the KD is only partial, this raises the question of whether the response to IncRIM/YAP could be more profoundly affected if IRP2 was totally gone. The authors shall acknowledge this limitation and temper their statement about the role of IRP2.

The exact reference of the IRP2 antibody is still not provided.

Rebuttal page 24 (Response Fig. 20):

My remark was related to Fig 4C but the text does not link Fig. S4D to Fig. 4C.

Rebuttal page 24,25 (Response Fig. 21):

It is written that Fig. 4g is now new Fig.4g, which is exactly the same. I guess the authors mean new Fig. S4f?

These data do not answer my question. Co-localization is not evidence of interaction. If the authors cannot provide formal evidence that FAC has no effect on NF2 pull-down efficiency, they could at least mention in the text that such effect cannot be formally ruled out.

Rebuttal page 25:

The data in Fig. 1p are RNA data and have limited value given that ferritin is essentially regulated at the post-transcriptional level via IRPs and ferritinophagy.

More importantly, a reduction of DMT1 and TFR1 in IncRIM-KD cells would unlikely explain the reduction in ferritin upon iron stimulation since cells can take up iron in a non-transferrin-bound form such as FAC independently of the TFR1-DMT1 system. This explanation is not valid.

Lane 352: iron lead to the formation of ferritin, not FTH1 (ferritin is a heteropolymer made of H and L chains).

Rebuttal page 26 (Response Fig. 22):

Fig. S4H: strange to have the 48h bar before the 36h bar, or is it a color inversion in the legend?

Fig. S4i,j: it looks like (visually-speaking) IncRIM has an effect on the iron response (indeed seems superior after 48h compared to previous data). However, the statistics assess differences between IncRIM KD (or IncRIM overexpression) and the respective control cells separately in normal iron conditions or in high iron conditions. The statistics do not assess whether the RESPONSE (fold change) to iron is significantly different in IncRIM-KD or overexpressing cells.

Rebuttal page 27 (Response Fig. 24):

Fig. 5d: it is very hard to visualize the signal (the counterstain is pretty strong). It would be good to have an insert with higher magnification to see where the iron signal is localized.

Rebuttal page 27, 28 (Response Fig. 25):

Fig. S5c: the signal is hard to visualize, it would be good to have an insert with high magnification to

see whether the proteins localize where they should (endosome for DMT1, membrane for TFRC).

Rebuttal page 28 (Response Fig. 26):

Fig. 5h-k and S5j: this is a nice addition to the manuscript. As for other histology data (5j), it would be good to have inserts with high magnification to better appreciate the staining.

Rebuttal page 30 (Response Fig. 27 and 28):

Fig. S5G and 6a: same comment as above regarding inserts with higher magnification. It is very hard to visualize the cells in Fig. 6A in particular.

Rebuttal page 31, minor point 1: lane 80, I would suggest to remove "overdose of".

Rebuttal page 32, minor point 6: lane 161, CAPT still not explained.

Lanes 234 and 241: the word "fragments" can be omitted in these 2 sentences.

REVIEWER COMMENTS

Reviewer #1 (Remarks to the Author):

The authors have adequately addressed to the concerns raised previously by this reviewer. Based on this the manuscript is now suitable for publication in Cell Death and Disease.

Response: We appreciate the reviewer's approval of our work!

Reviewer #2 (Remarks to the Author):

It is sometimes difficult to navigate through the huge rebuttal that the authors have generated. Changes in the text have been made and those are not highlighted. On the opposite, changes mentioned in the rebuttal are not highlighted in the text and are hard to find.

In any case, the authors have responded to most of my comments in a satisfactory manner. Some clarification is still needed, as detailed below.

Response: We appreciated your effort to review our revision and thanks for your suggestions! We have carefully revised our manuscript accordingly and included our point-by-point response letter. Those changes in the revised manuscript are highlighted in red. And we also polished our revised manuscript in Springer nature editing. Besides, the figure in this revised manuscript is called new figure. We hope that below responses would alleviate your concerns.

Rebuttal page 8:

The KD of TFR1 and DMT1 does not reduce tumor growth, colony formation, and ki67 expression to the level observed in EV cells. This simply reflects the essentiality of the nutrient iron for cell growth in general, independently of *lncRIM*. Yet, the TFR1/DMT1 KD approach is more specific than DFO treatment and is a good addition to the manuscript.

Response: Thanks for your suggestion in the last version to help better revise our manuscript! The results (**New Fig. 5h-k**) showed that knocking down both TFR1 and DMT1 partially inhibited *LncRIM*-mediated tumor growth, which further confirmed the idea that iron stimulation activated YAP and *LncRIM*-NF2 interaction to upregulate the downstream DMT1 and TFR1 expression to promote cell proliferation. In addition, colony formation and MTT assays also suggested oncogenic *LncRIM* partially dependent on the cellular iron level to promote cell proliferation (**New Fig. 1m, 1n and S1k, S1m**). Thus, *in vitro* experiments and *in vivo* mouse model together to some extent suggested that iron homeostasis was important in *LncRIM* mediating cell proliferation and tumor growth.

Figure 5h-k:

- I guess there is a color inversion in the legend to Fig. 5i (lncRIM vs lncRIM-double KD).

- indicate the statistical significance on the figure (as done in 5b).

- verify typos (space) for the Y-axis titles.

It would be good to have inserts with magnification of the signal in 5j. This would help the reader appreciate for example whether TFR1 is localized at the plasma membrane or DMT1 in endosomes, as expected.

Response: Thanks for pointing out this issue and your suggestion! **Fig. 5i is now New Fig. 5i.** We have corrected the color inversion, added the statistical significance in revised manuscript and check space for the Y-axis titles in the revised manuscript. Following your suggestion, we have also enlarged the immunohistochemical image of TFR1, DMT1 and the enhanced DAB iron staining (**Response Fig.1a**). In previous experiments, the DMT1 and TFR1 antibodies are both from Proteintech (Catalog:20507-1-AP and Catalog:10084-2-AP). To further precisely validate the location of TFR1, we also purchased TFR1 antibody (Catalog: A5865) from ABclonal and performed IHC again (**Response Fig.1b**). Although IHC cannot clearly see the endosome, we can identify the plasma membrane location of TFR1 and cytoplasm location of DMT1.

Response Fig. 1

(a) IHC staining image (Left) of TFR1, DMT1 and enhanced DAB iron staining, and the enlarge images (Right). Scale bar, 100 μ m.

(b) IHC staining of TFR1 (Left) and the enlarge images (Right). Scale bar, 100 μ m.

Rebuttal page 10 - Fe staining.

I find it still difficult to appreciate the location of the iron signal in the tissue. This may be due to the resolution of the PDF. Inserts with iron magnification could be added to 5d,j and S5g. Hopefully, high-resolution images allowing deep zooming will be provided with the article.

Response: Thanks for your suggestion! As shown in **Response Fig.2**, we have enlarged the iron staining of **New Fig. 5d, 5j and S5g** to help clearly identify the iron location. And these results showed that knocking down of *LncRIM* or DFO treatment both significantly reduced the cellular iron level, while overexpressed *LncRIM* increased the cellular iron level.

Response Fig. 2

(a and b) Enhanced iron staining of in Fig.5d and 5j (Left), and the enlarge image (Right). Scale bar, 100 μ m.

(c) Enhanced iron staining of in Fig.S5g (Top), and the enlarge image (Below). Scale bar, 100 μ m.

Rebuttal page 10, 11 - Role of MAYA/YAP in iron metabolism in the context of NAFLD. The answer to my comment is convoluted and hard to understand. I was simply asking to refer to the work by Yuan et al. in the text, as it would be interesting for the reader to realize that YAP and iron could be interconnected in different ways depending on context.

Response: Thanks for your suggestion! We have referred to this work by Yuan et al, and briefly discussed this result and our result in the discussion section (Lanes 455-459).

Rebuttal page 13,14 - iron in microenvironment cells:

The data in response Fig. 8 are very intriguing, and suggest that the iron status of the cancer cell shapes the macrophage microenvironment, and/or vice versa. Very interesting, but my question was simply about the localization of iron in Fig. 1d (or new Fig. 1d).

Response: Thank you very much for this suggestion, which helped us better revise the manuscript. And the **New Fig.1d** further clearly showed that activation of YAP was positively correlated with the iron level in breast cancer tissue. And we also discussed the tumor microenvironment and *LncRIM*-mediated iron metabolism in the revised manuscript (Lanes:475-488).

Rebuttal page 14 – interpretation Fig.1m,n (Response Fig. 9):

- The authors say they “re-analyzed” the data in Fig.1m. The data now look different from what they were in the first version of the manuscript, and differences between FAC and vehicle treated cells become statistically significant! What has changed?

Response: Thanks for pointing out this issue! **Fig.1m is now New Fig.1m.** We processed the images and counted the number of clones by using Image J software. In the first version, the number of clones calculated by the software was different from the real situation due to the variation of background brightness/contrast setting, so we made corrections to ensure the accuracy of the data in **New Fig.1m.** In addition, there are three experimental groups with two variables (*LncRIM* and FAC), the student’s *t*-test used in the first revision is not applicable to this scenario. Therefore, we’ve changed the statistical method and re-analyzed the data using the Two-way ANOVA analysis, which is one of the reasons for the change in significance.

- The interpretation of figure 1n is still uncertain. What matters is if the fold change in DFO vs vehicle-treated cells is significantly different when *LncRIM* is overexpressed (visually the effect of DFO seems slightly attenuated but is it statistically significant?).
- New Fig. S11 (cell cycle analysis). There is no indication of whether the differences observed are statistically significant or not.

Response: Thanks for pointing out this issue! **Fig.1n is now New Fig. 1n.** In the **New Fig.1n,** we treated cells with DFO and analysis showed that *LncRIM* overexpression partially reduced the inhibition of DFO on cell proliferation. Further, we found that in the EV group, DFO treatment decreased cell proliferation by around 77.2%, 70.1% and 65.6% in three independent biological repeats, while in the *LncRIM* overexpressed group, DFO treatment decreased cell proliferation about 47.9%, 51.3%, 53.3% (**Response Fig. 3a**), which indicated that the fold change in DFO vs vehicle-treated cells is significantly different with *LncRIM* overexpression. In addition, the mouse model with DFO injection or double knockdown of DMT1 and TFR1 also further illustrated the important role of *LncRIM* mediated cellular iron metabolism in cell proliferation (**New Fig.5h, S5g**). In addition, we also added the statistical analysis of **New Fig. S11** in the revised manuscript.

Response Fig. 3

(a) Analysis of cell proliferation reduction percentage of EV and *LncRIM* over-expressed groups with or without DFO treatment. mean ± s.d., n=3 biological independent experiments, n.s., not significant; **P < 0.01, Student’s *t*-test.

Rebuttal page 15: western blots in former Fig. S1o,p.

Fig.S1o,p is not new Fig.S1q as stated by the author but it is Fig. S1p!

The authors said they did statistical analyses. I can see ratio indicating they have

QUANTIFIED the western blot signals, but I don't see anything related to STATISTICAL SIGNIFICANCE. Please clarify.

Response: Thanks for pointing out this issue and we apologize for this incorrect description! As you suggested, we did biological statistical analysis of three independent biological repeats of **New Fig. S1p**. The results showed that overexpression of *LncRIM* increased the expression of DMT1 and TFR1, while knocking down of *LncRIM* decreased the expression of both the DMT1 and TFR1 (**Response Fig. 4**).

Response Fig.4

(a and b) Statistical Analysis of the expression level of DMT1 and TFR1 in control and *LncRIM* overexpressed or knocked down MCF7 and MDA-MB-468 cells. mean \pm s.d., n=3 biological independent experiments, **P < 0.01; ***P < 0.001, Student's *t*-test.

Rebuttal page 19 - Chip-seq data DMT1 (Response Fig. 15):

The authors state they added the GEO dataset reference to the figure legend but I don't see it.

Response: Thanks for pointing out this issue! We have added the GEO dataset (GSE107013) in the **Supplementary figure 3 legend** in the revised manuscript.

Rebuttal page 20 - DMT1 luciferase reporter assays (response Fig. 16):

Fig. 3i alone is not convincing, as commented in my previous review. Fig. S3i is more convincing. How was the mutant generated exactly? How was basal expression of the reporter affected by the mutation? Was it in a similar range for the two types of constructs?

Response: Thanks for this question! By using GEO database (GSE107013) and MEME (<https://meme-suite.org/>), we analyzed the potential binding motif of YAP/TEAD4 at DMT1 promoter (**Response Fig. 5a**) (**New Fig. S3h, i**). Further, we designed several pairs of primers in different regions of DMT1 promoter and found the specific binding region by using ChIP-PCR experiment (**New Fig. 3i**). Then we constructed DMT1 promoter mutant by deleting this binding motif (CATTCT), and the result showed that

mutation of binding site significantly attenuated YAP/TEAD-induced DMT1 promoter luciferase activity (**New Fig. S3j**).

Here, we adopted the double luciferase reporter gene assay. The ratio of Firefly Luciferase value to Renilla Luciferase value in the same sample well Firefly/Renilla Luciferase value was taken as the relative expression amount of luciferase, so the experimental error was excluded. Then we quantized the control group into one, and compared the experimental group data with the control group to get fold change. Compared to the WT DMT1-promoter, mutation decreased the basal expression of firefly luciferase reporter, and the fluorescent expression range of these two constructs was about 10^5 - 10^6 .

Response Fig. 5

(a) YAP/TEAD-binding elements in the DMT1 promoter region.

Rebuttal page 20,21 - DMT1 isoforms (response Fig. 17):

Figure S3j,k: This type of comparison relies on distinct qPCR assays. For it to be valid, one needs to make sure cDNA synthesis is equal between isoforms (e.g. no GC rich structure impairing RT in one isoform) and that primer efficiency is the same (or corrected for). Was this the case?

Response: Thanks for this question! Here, we carried out RT-qPCR assay to test the expression of these four DMT1 isoforms (IRE, non-IRE, 1A and 1B), and found DMT1-IRE isoform was significantly high expression in breast cancer cells (**New Fig. S3k**). Our actual situation is slightly different from your description. In this experiment, total RNA rather than specific fragments was extracted from MCF7 and MDA-MB-468 cells, and then we carried out RT to generate cDNA. Then we took the same amount cDNA and conduct quantitative PCR by using the specific primers with roughly the same efficiency of DMT1 isoforms published in the previous study to compare their expression (PMID:12209011, PMID:9642100).

Rebuttal page 21,22:

Response Fig. 18 shows a nice piece of data. Yet, the induction of TFR1 and DMT1 by *LncRIM* and YAP seems partially blunted in IRP2-KD cells. As the KD is only partial, this raises the question of whether the response to *LncRIM*/YAP could be more profoundly affected if IRP2 was totally gone. The authors shall acknowledge this limitation and temper their statement about the role of IRP2.

The exact reference of the IRP2 antibody is still not provided.

Response: Thanks for pointing out this issue! We have modified the description about the role of IRP2 and our *LncRIM*-NF2 axis in the revised manuscript (Lanes: 303-308, 312-313, 492-496, 505-511). Besides, the IRP2 antibody in this study was purchased from the Santa Cruz Biotechnology (sc-33682) (PMID:35504898, PMID: 35654137, PMID:35977492) and we added it to the revised manuscript of **Antibodies**.

Rebuttal page 24 (Response Fig. 20):

My remark was related to Fig 4C but the text does not link Fig. S4D to Fig. 4C.

Response: Thanks for your constructive suggestion to help better revise our manuscript! **Fig.4c, S4d is now New Fig. 4c, S4e respectively**. We are sorry for the confusion description and we have changed this in the revised manuscript (Lanes: 337-345).

Rebuttal page 24,25 (Response Fig. 21):

It is written that Fig. 4g is now new Fig.4g, which is exactly the same. I guess the authors mean new Fig. S4f? These data do not answer my question. Co-localization is not evidence of interaction. If the authors cannot provide formal evidence that FAC has no effect on NF2 pull-down efficiency, they could at least mention in the text that such effect cannot be formally ruled out.

Response: Thanks for pointing out this issue and sorry for this confusion! **Fig.4g, S4f is now New Fig.4g, S4f respectively**. **New Fig. 4g** was a RIP assay, and we performed this experiment to test the effect of iron on *LncRIM* and NF2 interaction. In this experiment, 200 μ M FAC was added during the cell culture, and after collecting cells we used NF2 antibody to immunoprecipitate endogenous NF2, followed by RNA extraction and RT-qPCR. And the result showed that FAC stimulation significantly increased the enrichment of NF2 to *LncRIM*. Besides, immunofluorescent result (**New Fig. S4f**) also showed that iron overload significantly increased the expression of *LncRIM* on the membrane and binding to NF2, while decreasing LATS1 membrane association.

Here, we demonstrated that *LncRIM* competitively binds to NF2 and therefore inhibited NF2-LATS1 interaction (**New Fig. 2e, f, m and S2m**). Moreover, we also illustrated that *LncRIM* was downstream target of YAP, and iron stimulation significantly activated YAP and up-regulated the expression of *LncRIM* on cell membrane, as well as increased the *LncRIM*-NF2 co-localization (**New Fig. 4a, c, e, f, and S4f**). In addition, FAC and *LncRIM* did not affect NF2 expression (**New Fig.S2d, S2i and S4f**). Besides, we performed a western blot assay after the lysates incubated with NF2 antibody, and the result showed that FAC had little effect on NF2 pull-down efficiency (**Response Fig. 6**). Thank you again for this suggestion, and we have mentioned this issue in the revised manuscript (Lanes:471-474).

Response Fig.6

(a) Immunoblot detection of NF2 expression after immunoprecipitated by the NF2 antibody in MCF7 cells with or without FAC stimulation. IgG was used as the negative control.

Rebuttal page 25:

The data in Fig. 1p are RNA data and have limited value given that ferritin is essentially regulated at the post-transcriptional level via IRPs and ferritinophagy. More importantly, a reduction of DMT1 and TFR1 in *LncRIM*-KD cells would unlikely explain the reduction in ferritin upon iron stimulation since cells can take up iron in a non-transferrin-bound form such as FAC independently of the TFR1-DMT1 system. This explanation is not valid.

Lane 352: iron lead to the formation of ferritin, not FTH1 (ferritin is a heteropolymer made of H and L chains).

Response: Thanks for pointing out this issue! In the **New Fig. 1h, k, l and S1h, i**, we found that *LncRIM* regulated cellular iron metabolism. Then in the **New Fig.1p**, we determined to test whether *LncRIM* affected the expression of specific iron metabolism related genes, and the result showed that knocking down of *LncRIM* decreased the expression of iron-uptake related genes, DMT1 and TFR1 in both RNA and protein level (**New Fig. 1p** and **S1p**). Also, we found that *LncRIM* had little effect on the expression of ferritin in the protein level (**New Fig. 4i, j**).

Apart from the TF-TFR1 system in iron uptake, there are also some other pathways to acquire iron. For example, lipocalin2 (LCN2)-dependent endocytosis of an iron-laden siderophore via the SLC22A17 lipocalin receptor (PMID: 16377569, PMID:18418376); two ZRT/IRT-like family proteins ZIP14 and ZIP8 mediated uptake of non-transferrin-bound iron (NTBI) (PMID:22318508, PMID:26725301). Then we assessed the expression of LCN2, ZIP14 and ZIP8 after *LncRIM* was knocked down. The result showed that *LncRIM* had little effect on the expression of other non-transferrin-bound systems (**Response Fig. 7**). Here, we illustrated that *LncRIM*-NF2 coordinated Hippo-YAP pathway regulated cellular iron metabolism *in vivo* and *in vitro* (**New Fig. 2e, 3c,3g, 5a, 5h**). And previous study has shown that iron treatment would significantly increase the FTH1 and FTL protein levels (PMID:26461092). Thus, *LncRIM* knockdown to some extent was responsible for the reduced formation of ferritin under FAC stimulation.

And we have corrected the *fth1* to ferritin through revised manuscript. Thanks for your suggestion again!

Response Fig. 7

(a) RT-qPCR analysis of the expression of *ZIP14*, *SLC22A17*, *LCN2* and *ZIP8* in control and LncRIM knockdown MCF7 cells. mean \pm s.d., n=3 biological independent experiments, n.s., not significant; ***P < 0.001, Student's *t*-test.

Rebuttal page 26 (Response Fig. 22):

Fig. S4H: strange to have the 48h bar before the 36h bar, or is it a color inversion in the legend?

Response: Thanks for pointing out this issue! For **Fig.S4h (New Fig.S4h)**, we have corrected the color of 36h and 48h in the revised manuscript.

Fig. S4i,j: it looks like (visually-speaking) LncRIM has an effect on the iron response (indeed seems superior after 48h compared to previous data). However, the statistics assess differences between LncRIM KD (or LncRIM overexpression) and the respective control cells separately in normal iron conditions or in high iron conditions. The statistics do not assess whether the RESPONSE (fold change) to iron is significantly different in LncRIM-KD or overexpressing cells.

Response: Thanks for pointing out this issue! We have analyzed the response (fold change) to iron in *LncRIM* knockdown and over-expressed cells. As shown in **Response Fig.8**, the fold change to iron of *LncRIM* overexpressed MDA-MB-468 and MCF7 cells is higher than in control cells. However, we found in *LncRIM* knocked down cells, the response to iron sometimes was a little higher than in control cells, which may due to the compensatory effect (PMID:30944473, PMID:25759022), and this needs to be explored further.

Response Fig.8

(a and b) Statistical Analysis of the fold change to iron in control and *LncRIM* over-expressed MCF7 and MDA-MB-468 cells. mean \pm s.d., n=3 biological independent experiments, *P < 0.05, Student's *t*-test.

Rebuttal page 27 (Response Fig. 24):

Fig. 5d: it is very hard to visualize the signal (the counterstain is pretty strong). It would be good to have an insert with higher magnification to see where the iron signal is localized.

Response: Thanks for your suggestion! **Fig. 5d** is now **New Fig. 5d**. We have enlarged the iron staining to help clearly identify the iron location (**Response Fig. 2**). And this result showed that knocking down of *LncRIM* significantly decreased the cellular iron level.

Rebuttal page 27, 28 (Response Fig. 25):

Fig. S5c: the signal is hard to visualize, it would be good to have an insert with high magnification to see whether the proteins localize where they should (endosome for DMT1, membrane for TFRC).

Response: Thanks for your suggestion! **Fig. 5c** is now **New Fig. S5c**. We have enlarged the immunohistochemistry of DMT1 and TFR1 (**Response Fig. 9a**). Besides, considering the specificity of the TFR1 antibody, we also purchased another TFR1 antibody from ABclonal (Catalog: A5865), and performed IHC assay again (**Response Fig. 9b**). Although IHC cannot clearly see the endosome, we can identify the plasma membrane location of TFR1 and cytoplasm location of DMT1.

Response Fig. 9

(a) IHC staining of TFR1 and DMT1 (Top), and the enlarge images (Below). Scale bar, 100 μ m.

(b) IHC staining of TFR1 (Top) and the enlarge images (Below). Scale bar, 100 μ m.

Rebuttal page 28 (Response Fig. 26):

Fig. 5h-k and S5j: this is a nice addition to the manuscript. As for other histology data (5j), it would be good to have inserts with high magnification to better appreciate the staining.

Response: Thanks for your suggestion! We have enlarged the IHC staining of TFR1 and DMT1, and re-performed IHC staining of TFR1 by using another TFR1 antibody from ABclonal (**Response Fig.1**).

Rebuttal page 30 (Response Fig. 27 and 28):

Fig. S5G and 6a: same comment as above regarding inserts with higher magnification. It is very hard to visualize the cells in Fig. 6A in particular.

Response: Thanks for your suggestion! **Fig. S5g** and **6a** are now **New Fig. S5g** and **6a** respectively. As shown in **Response Fig. 10**, we have enlarged the iron staining to help clearly identify iron staining. These results further demonstrated that overexpression of *LncRIM* partially restored the cellular iron level with DFO treatment, and *LncRIM*-mediated changes in iron levels were mostly located in breast cancer cells, not in other cells.

Response Fig. 10

(a and b) Iron staining of S5g(Top), and 6a (Left), and the enlarge image (Below and Right respectively). Scale bar, 100 μ m.

Rebuttal page 31, minor point 1: lane 80, I would suggest to remove “overdose of” .

Response: Thanks for this suggestion! We have removed “*overdose of*” in the revised manuscript (Lanes:78-80).

Rebuttal page 32, minor point 6: lane 161, CAPT still not explained.

Response: Thanks for pointing this out! The Prediction by the CPAT (Coding Potential Assessment Tool) (<http://lilab.research.bcm.edu/>) showed the coding probability of *LncRIM* (NR_034137.1) is 0.0424, much lower than the cutoff value (0.364), indicating that *LncRIM* lacked the potential ability of protein-encoding. We have added this in the revised manuscript (Lanes:160-162).

Lanes 234 and 241: the word “fragments” can be omitted in these 2 sentences.

Response: Thanks for this suggestion! We have omitted this word in the revised manuscript (Lanes:240-247).

Reviewer #3 (Remarks to the Author):

In the present study, the authors have reported the role of a novel lncRNA; lncRIM, in YAP-mediated iron metabolism. The authors demonstrated a positive correlation between YAP levels and intracellular iron levels in cell lines and breast cancer tissue samples. lncRIM is a target of the YAP/TEAD activator complex. lncRIM binds to NF2 on the membrane and prevents the interaction between NF2 and LATS1 kinase, thereby abolishing the LATS1-mediated phosphorylation and degradation of YAP. In this way, lncRIM promotes the nuclear translocation of YAP, thereby promoting YAP-mediated transcription activation of genes controlling iron metabolism.

In general, this is an exhaustive study. Most of the experiments were conducted in breast cancer and kidney cancer cells, and the data is supported by experiments performed in mouse xenograft models as well in cancer patient tissue samples. The major drawback of the ms is that the authors have shown some sort of bias to connect the lncRIM and YAP pathway without performing an unbiased screen (see the details below). Also, several of the immunoblots seems to be modified (at least based on the pdf files that I have downloaded).

Response: We are very grateful for your recognition of our work, and thank you for these creative suggestions, which will help us to better improve our research. Here, we found and validated one novel mechanism of lncRNA *lncRIM* in cellular iron metabolism through a series of unbiased bioinformatic screening and experimental analysis step by step (Also see the details below in point-to-point response to specific comments):

- Initially, by using bioinformatic screening (GEO number: GSE38369) and calcein-AM assay, we discovered the potential crosstalk between iron metabolism and the Hippo pathway (**New Fig.1a, 1b S1a**). Many studies have revealed the vital role of signaling pathway related lncRNAs in mediating metabolism regulation, we then focused on whether Hippo signaling related lncRNAs involved the tandem role. By using previous database where we have pinpointed 40 lncRNAs that were potentially required for YAP-dependent transcription (PMID:28114269), we found and demonstrated one YAP-related lncRNA *lncRIM* response to FAC/DFO stimulation and in turn regulated cellular iron metabolism (**New Fig.1h, 1k, 1l, S1b, S1h-j**). And *lncRIM* is associated with poor survival in breast cancer patients and iron homeostasis was vital for *lncRIM* modulated cell proliferation (**New Fig.1i,1j,1m,1n,S1k**).
- Then, combined the potential link between *lncRIM*-mediated iron metabolism and the Hippo pathway, we identified and mapped an interaction surface and co-localization between *lncRIM* and the tumor suppressor NF2. This interaction impedes the binding of NF2 to LATS1, resulting in downstream activation of YAP (**New Fig. 2a-h, S2a, S2f, S2g, S2i**). And *lncRIM* wires up the Hippo signaling to increase cellular iron levels via direct stimulation of TFR1 and DMT1 transcription by YAP/TEAD (**New Fig. 3c-f, 3i, 3j, S3j**). Besides, we also examined the

relationship between *LncRIM*-NF2 axis and the classic IRP/IRE system from different perspectives (**New Fig. 3k, 3l, S3m-r**), which to some extent illustrated that this *LncRIM*-NF2 axis exerts biological effect effectively and may differ from the IPR2.

- In addition, we connected these findings and describe a positive feedback loop whereby iron stimulation further activates YAP to increase *LncRIM* expression and *LncRIM*-NF2 association to promote cell proliferation and tumor growth (**New Figs.4a, 4b, 4g-k, S4f**). Moreover, colony number experiments and tumor transplantation experiments in mice including knockdown of *LncRIM*, *LncRIM* overexpression, double knockdown of TFR1 and DMT1 also together further validated that this *LncRIM*-NF2 feedback loop contributes to promoting tumor growth by increasing cellular iron level and iron homeostasis was important for *LncRIM*-induced cell proliferation and tumor growth (**New Fig. 1m, 1n, 5a-d, 5h-j, S5e-g**).

Thanks for your effort to review our manuscript again! As you suggested, we have revised our figure and manuscript, and have done more biological repeats of some data. Besides, we also provided source data of immunoblots in the **Uncropped gel images**.

Specific comments

What was the basis for the authors to select only a subset of lncRNAs for RT-qPCR experiments shown in fig 1f? Did they perform an RNA-seq to identify the candidates initially, and then validate the top hits by RT-qPCR?

Response: Thanks for pointing out this issue and we apologized for this confusing description, we have clearly explained it in the revised manuscript! **Fig.1f** is now **New Fig. 1f**. In this study, we uncovered one novel mechanism in which oncogenic lncRNA *LncRIM* regulated cellular iron metabolism effectively by directly bind to NF2 and dependent on the Hippo-YAP pathway step by step through a series of unbiased bioinformatic screening and experimental analysis. Following is the detailed procedure:

- Firstly, by using the GSEA analysis (GEO number: GSE38369) (PMID:22767065), Calcein-AM assay, and clinical breast cancer tissue (**New Fig.1a-e, S1a**), we found YAP activation was positively correlated with iron level in breast cancer, which indicated the potential link between the Hippo pathway and cellular iron metabolism.
- Then, considering the vital role of signaling pathway related lncRNAs in mediating metabolism regulation and combing with above results, we tried to screen out whether there was promising Hippo pathway related lncRNAs that involve in iron metabolism. And our previous study database (PMID:28218907) pinpointed 40 lncRNAs that were potentially required for YAP-dependent transcription by transfecting the human Lincode® siRNA library into MCF7 cells that were

engineered with a TEAD-driven luciferase reporter. By overlapping this RNA-seq database and FAC/DFO stimulation, we obtained a cytoplasmic lncRNA *Loc729013*, re-named *LncRIM* (LncRNA Related to Iron Metabolism), which obviously responded to alteration of cellular iron concentration and also regulated cellular iron level among others (New Fig.1f-h, 1k, 1l, S1b). Besides, we also found *LncRIM* regulated the YAP downstream targets (New Fig. S1q).

- Furthermore, we also confirmed that *LncRIM* co-localized with NF2 on cell membrane to impair NF2-LATS1 interaction (New Fig. 2), and upregulated the transcription of DMT1 and TFR1 (New Fig. 3h-j, S3c-f, S3j). Besides, *LncRIM*-NF2 regulated cellular iron metabolism dependent on the Hippo pathway effectively, in a distinct manner of IRP2 (New Fig. 3c-f, 3k, 3l).
- Besides, tumor transplantation experiments in mice also showed that knockdown of *LncRIM* significantly decreased iron level and tumor growth (New Fig. 5d, e), which also further demonstrated the important role of *LncRIM* mediated cellular iron metabolism in breast cancer development.

We have explained this issue clearly in the revised manuscript (Lanes:147-160). Thanks again for pointing out this issue!

Did the experiment detail in fig 1h perform in presence of FAC? Since *LncRIM* is upregulated in FAC-treated cells, the iron-metabolism assay should be done under similar conditions.

Response: Thanks for pointing out this issue and we are sorry for this confusion! Fig.1h is now New Fig.1h. In the New Fig.1f,1g, we have screened out 6 promising YAP related lncRNAs that responded to cellular iron level, which indicated the potential role of these 6 lncRNAs in turn regulate cellular iron metabolism. To further test this hypothesis, we then performed the Calcein-AM assay with siRNA targeted for these 6 lncRNAs in MCF7 cells, where knockdown of lncRNA *Loc729013*, which is renamed as *LncRIM*, robustly decreased the level of basal cellular iron among others (New Fig.1h, 1k, 1l, S1b). Consistent with this, knockdown of *LncRIM* also dramatically decreased the iron level in the tumor as shown in New Fig. 5d. We have clearly described this in the revised manuscript (Lanes: 147-156, 383-385).

what is the copy number of *LncRIM*? Authors argued that *LncRIM* by binding to NF2 quenches NF2 from interacting with LATS1. In order for this to happen, the copy number *LncRIM* should be comparable to the NF2 protein levels. This needs to be estimated.

Response: Thanks for this experimental advice. For the copy number of *LncRIM*, we have determined that there are roughly 690 copies of the *LncRIM* per MCF7 cell and 502 copies per MDA-MB-468 cell (Response Fig.11a, b) (New Fig. S2b, c), which was of relatively high abundance compared with that of several known functional lncRNAs: *LINK-A* has roughly 150 copies per MDA-MB-231 cell and *CamK-A* has

roughly 937 per MDA-MB-231 cell (PMID:28218907, PMID:30220561). Also, we have carefully revised our manuscript by adding current knowledge about *LncRIM* and new experiments to address the reviewers' concerns (Lanes:216-221).

For the model of competition: NF2 molecule number per MCF7 cell was determined as around 830 thousand (**Response Fig.11c, d**), we concluded that *LncRIM*/NF2 ratio in the membrane was about 1: 1200 in normal conditions and about 1:400 under FAC stimulation with the expression of *LncRIM* increased about 3-fold changes (**New Fig. 4c**). Besides, we also showed that the similar ratio of *LncRIM* (25nM) and NF2 (20nM) was sufficient to interfere with the NF2-LATS1 complex (**New Fig. S2j, k**).

Combined with the experimental evidence above, we thought *LncRIM* could be sufficient to function with NF2 in reasonable effect size in cellular iron metabolism and cancer progression.

Response Fig. 11

(a and b) *LncRIM* copy number was determined in MCF7 and MDA-MB-468 cell lines (a). Different concentration of *LncRIM* plasmid was used as control to analyze the copy number, and we extracted RNA of 6 well MCF7 and MDA-MB-468 cells with 20ul DEPC to dissolve. Data are presented as mean values \pm SD, n=3 biologically independent experiments (b).

(c and d) The absolute concentration of the purified GST-NF2 (10μl) protein was determined by Coomassie staining with BSA standard curve (0.1-2.0μg) (c). Around five million cells (MCF7, MCF10A separately) were collected by cell counting and 10% of them was input to SDS-PAGE gels for anti-NF2 immunoblot detection. NF2 Standard curve (0.005-0.100μg) was used to determine the endogenous NF2 mass and the corresponding molecule number per cell by measuring the bands' relative grey values (d).

Also, wherein the cytoplasm does *LncRIM* localize? Based on the model, it should co-localize with NF2 on the membrane.

Response: We thank the reviewer for this suggestion! As shown in **New Fig.1q**, we validated that *LncRIM* was mostly located in the cytoplasm. Then, we designed the probe to precisely validate the accurate location of *LncRIM*. As seen in **Response Fig. 12 (New Fig. 2e, f)**, RNA fluorescence in situ hybridization (RNA FISH) result showed the *LncRIM* was co-localized with NF2 on the cell membrane in MCF7 cells.

Response Fig. 12

(a and b) *LncRIM* RNA FISH and immunofluorescence staining of endogenous NF2 were performed in MCF7 cells. Line scan of the relative fluorescence intensity of the signal (dotted line; left) is plotted to show the peak overlapping (right). The *LncRIM* probe was labeled with Cy3 (red) and the NF2 (green) was detected with Alexa Fluor 488. Scale bar, 10 μ m.

Data presented in fig 2g should be quantified and also should be shown in the case of endogenous proteins. In addition, I see the reduced signal of both tagged-NF2 and LATS1 on the membrane in the absence of *LncRIM*. Does *LncRIM* also facilitate the localization of NF2 on the membrane? Again, in this context, it is important to show the localization of *LncRIM* in the cell.

Response: Thanks for this constructive suggestion! **Fig.2g** is now **New Fig.2j**. We are sorry for the confusion in previous Fig.2g, which may be due to the different transfection efficiency of GFP-NF2. We have repeated this experiment in MCF7 cells and quantified the interaction of F_{Lats1}/F_{NF2} in control and *LncRIM* overexpressed cells (**Response Fig.13a, b**) (**New Fig. 2j, k**). And the results indicated that overexpression of *LncRIM* significantly inhibited LATS1-NF2 interaction on the cell membrane. Besides, we apologized for not finding suitable antibodies of both endogenous NF2 and LATS1 with different fluorescent secondary antibodies. Thus, we tested the endogenous NF2-LATS1 interaction in *LncRIM* overexpressed MCF7 cells by using subcellular fractionation (**Response Fig. 13c**) (**New Fig. S2i**), which also showed that overexpression of *LncRIM* inhibited the LATS1 membrane location and NF2-LATS1 interaction.

Besides, fluorescence and subcellular fractionation results showed that overexpression of *LncRIM* had little impact on NF2 membrane localization (**Response Fig.13c-e**) (**New Fig. S2d, e, i**). Further, iron stimulation increased the expression of *LncRIM* on membrane and the interaction between *LncRIM* and NF2, while further decreasing LATS1 membrane interaction (**Response Fig. 13f, g**) (**New Fig. S4f, g**), which together illustrated that Again, thank you for these suggestions which help better improve our revised manuscript.

Response Fig. 13

(a and b) Immunofluorescence staining was performed in MCF7 cells expressing GFP-NF2 and pRFP-LATS1 with or without over-expression of *LncRIM*. The data was quantified with Image J. (b) (mean \pm SD, n=5 independent experiment; ***P < 0.001, Student's *t*-test).

(c) Subcellular fractionation was performed to assess endogenous NF2 and LATS1 expression in control and *LncRIM* over-expressed MCF7 cells.

(d and e) Immunofluorescence staining was performed to assess the endogenous NF2 membrane location in control and *LncRIM* over-expressed MCF7 cells. The data was quantified with Image J. (mean \pm SD, n=5 independent experiment; n.s., not significant, Student's *t*-test).

(f and g) *LncRIM* RNA FISH and immunofluorescence staining were performed in MCF7 cells expressing GFP-NF2 with or without FAC stimulation (200 μ M). Line scan of the relative fluorescence intensity of the signal (f) is plotted to show the peak overlapping (g). The *LncRIM* probe was labeled with Cy3 and the LATS1 was detected with Alexa Fluor 647. Scale bar, 20 μ m.

Initial data indicates that both YAP and *LncRIM* positively influence the iron levels in BC cells. However, it is not clear to me why authors have presumed that both these molecules play in the same pathway, and therefore have decided to identify *LncRIM* interacting proteins that are part of the HIPPO pathway. This seems to me like a biased approach without a strong rationale.

Ideally, *LncRIM* pull-down followed by mass spec should be conducted to identify the top hits.

Response: Thanks for this pointing out this issue! In this study, we uncovered one novel oncogenic *LncRIM*-mediated mechanism of cellular iron metabolism. And through a large number of unbiased bioinformatics screening and *in vitro* and *in vivo* experiments, we gradually revealed the crosstalk between *LncRIM*, Hippo signaling pathway, and cellular iron metabolism in breast cancer progression. Following is the detailed procedure:

- Firstly, by using the GSEA analysis, calcein-AM assay, and IHC of clinical breast cancer tissue (**New Fig. 1a-e, S1a**), we found YAP was significantly inactive under the deregulation of cellular iron levels and the potential link between the Hippo pathway and cellular iron metabolism.
- Then, by using our previous RNA-seq database (PMID:28218907), in which we have pinpointed 40 lncRNAs that were potentially required for YAP-dependent transcription by transfecting the human Lincode® siRNA library into MCF7 cells that were engineered with a TEAD-driven luciferase reporter, we unbiasedly obtained the cytoplasmic lncRNA *LncRIM*, which obviously responded to alteration of iron concentration, and in turn regulated cellular iron level through DMT1 and TFR1 (**New Fig. 1f-h, 1k, 1l, 1p, S1h, S1n-p**). Besides, we also found YAP-related *LncRIM* affected both the expression of YAP downstream targets (**New Fig. S1q**). Thus, above data together indicated the potential link between cellular iron metabolism, *LncRIM* and the Hippo pathway.
- Cytoplasmic location lncRNAs are thought to form RNA-protein complex to regulate various physiological process (PMID:24105322). To further clarify the connection between *LncRIM*-mediated iron metabolism and the Hippo pathway, we then performed in vivo and in vitro RNA pull-down, and the results showed that *LncRIM* only directly bound to NF2 but not other molecules of the Hippo pathway (LAST1, MOB1, MST1, YAP). And this *LncRIM*-NF2 axis significantly inhibited NF2-LATS1 interaction (**New Fig. 2, S2**).
- Moreover, we also demonstrated that *LncRIM*-NF2 axis regulated cellular iron metabolism dependent on the Hippo-YAP pathway, and DMT1 and TFR1 are both the target genes of YAP (**New Fig. 3c-k, S3i, S3j**). More importantly, we also further demonstrated that *LncRIM* was also the downstream target of YAP, and *LncRIM*-Hippo feedback loop hyperactivated YAP and regulated cellular iron metabolism to promote cell proliferation (**New Fig. 4**).

Overall, we came to this conclusion step by step that iron-trigger lncRNA *LncRIM* regulated cellular iron metabolism by directly binding to NF2 and dependent on the Hippo-YAP pathway effectively. We have made it clear in the revised manuscript. Thanks again for pointing out this issue!

There seem to be some undesirable modifications done on several of the immunoblots presented in the ms [some examples include Figs 2e (WCL: IB: HA) & f (IP: IB: NF2) and Fig S2b (IP-flag: IB: HA), Figs. S3d, f].

Response: We apologize for this confusion! **Fig. 2e, 2f, S2b** is now **New Fig. 2g, 2h, S2f** respectively; **Fig. S3d, S3f** is now **New Fig. S3f, 3k** respectively. We have done more biological repeats of these experiments and provided the raw data of these

immunoblots in **Uncropped gel images**. Hopefully these revised data could alleviate your concern.

It is not clear why some of the experiments were done in HEK293T cells when the main focus of the paper is to understand the role of *lncRIM* in iron metabolism in breast cancer cells.

Response: Thanks for pointing out this issue! For the previous **Fig. 1b, c, 2g, 2h, 3l, 4h**, we have done more biological repeats in MCF7 cells (**New Fig.1b, 1c, 2i-k, S3j,4h**) and added these data in the revised manuscript. Previous studies have shown that molecules of the Hippo-YAP pathway play the same biological regulation and mechanism in HEK293T cells as in cancer cells (PMID:24012335, PMID:17974916, PMID:26045165). Meanwhile, in HEK293 cells, the Hippo-YAP pathway also responds to the various upstream metabolic signals and presents a similar change in cancer cells (PMID:29100056, PMID:30472188, PMID:22863277).

We demonstrated that *LncRIM* inhibited NF2-LATS1 association in both HEK293T and MCF7 cells (**New Fig. 2g, 2h, 2i, S2f**), and verified that DMT1 was the downstream target of YAP in these two cell lines (**New Fig. 3i, 3j, S3c-e, S3j**). Moreover, iron stimulation enhanced the role of *LncRIM*-NF2 axis in inhibiting LATS1-NF2 association and further promoted the interaction between *LncRIM* and NF2 (**New Fig.4e-h, S4f, S4g**), which presented the same finding in both HEK293T cells and MCF7 cells. Thus, we thought it was reasonable to do some auxiliary experiments in HEK293T cells to further support our findings.

Does KD of *lncRIM* enhance NF2-LATS1 interaction?

Response: Thanks for this experimental advice! As seen in the **Response Fig.14 (New Fig. S2h)**, knockdown of *LncRIM* significantly increased the interaction between NF2 and LATS1, which was consistent with the results that overexpression of *LncRIM* inhibited the association of LATS1 and NF2 (**New Fig.2g-k, S2f, S2g, S2j**).

Response Fig. 14

(a) An endogenous coimmunoprecipitation (Co-IP) by using NF2 antibodies was performed to test NF2 and LATS1 interaction in control and *LncRIM* knockdown MCF7 cells. The NF2 antibody was pulled down with Protein A/G beads. IgG was used as the negative control.

fig 1B shows changes in the expression of iron metabolism genes in *lncRIM*-depleted cells. If *lncRIM*-depleted cells show such a dramatic increase in YAP phosphorylation,

ultimately resulting in the inactivation of YAP, then one would expect reduced expression of a significant number of YAP target genes. Is that the case?

Response: Thanks for pointing out this issue! As shown in **Response Fig. 15 (New Fig. S1q)**, knockdown of *LncRIM* also significantly decreased the expression of YAP downstream targets. We have added this data to the revised manuscript.

Response Fig. 15

(a) RT-qPCR detection of the expression of YAP downstream targets in control and *LncRIM* knocked down MCF7 cells.

The model presented in fig 2r should be demonstrated by in vitro analyses. authors should test the binding kinetics of purified NF2 and LATS with varying concentrations of *lncRIM*.

Response: We appreciate the reviewer's constructive suggestions! The N-terminus of LATS1 (LATS1-NT) contributes to the binding to NF2 (PMID:26045165). By using purified GST-LATS1-NT and MBP-His-NF2 proteins, we performed GST-pull down to assess the binding kinetics of NF2 and LATS1 under different concentration of in vitro transcribed biotinylated *LncRIM* (**Response Fig. 16 (New Fig. S2j, k)**). And the result also showed that *LncRIM* significantly inhibited the interaction between LATS1 and NF2 in a dose-dependent. Besides, the result also showed that approximately ratio 1:1 of *LncRIM* (25nM) and NF2 (20nM) is sufficient to inhibit NF2-LATS1 association. We appreciated the reviewer's kind suggestion and we also provided this data in the revised manuscript as suggested.

Response Fig. 16

(a and b) GST-pull down was performed to test the interaction between LATS1 and NF2 under different concentration of in vitro-transcribed *LncRIM* by using 20nmol purified GST-LATS1-NT and 20nmol His-MBP-NF2 proteins.

Fig 1F panel wrote MCF7, whereas the text in the result section indicates HEK293T.

Response: We are sorry for this writing and thanks for pointing out this issue! Indeed, we screened out the potential lncRNAs responded to iron stimulation in MCF7 cells. We have corrected this in the revised manuscript.

YAP shRNAs do not seem to efficiently deplete YAP (Fig s3a). Also, there seems to be some issue with the YAP-IB.

Response: Thanks for pointing out this issue! **Fig. S3a** is now **New Fig. 3c**. We have re-constructed the YAP knockdown cell lines with two different shRNA in MCF7 cells and examined the knockdown efficiency (**New Fig. 3c**). Also, we provided the raw data in the **Uncropped gel images**.

Fig s3b: How do the control and YAP-depleted cells show similar levels of CTGF and CYR61 mRNAs?

Response: Thanks for pointing out this issue! We apologized for the quality of previous data. Herein, *LncRIM* was proved to regulate the expression of DMT1 and TFR1 depending on the Hippo-YAP pathway, and DMT1 and TFR1 both were downstream targets of YAP (**New Fig. 3c-e, 3i, 3j, S3i, S3j**). Besides, *LncRIM* also regulated YAP activation and downstream targets (**New Fig. S1q, 3a, 3b**). Moreover, re-expression of YAP significantly restored the expression of DMT1 and TFR1, and the cellular iron level to a similar level to that of *LncRIM* over-expressed so as the cellular iron level (**New Fig. 3g, h**). Consistent with this, the level of *CTGF* and *CYR61* also present a similar change. We have repeated this experiment by using effective YAP knockdown cell lines and revised the figure (**Response Fig. 17**) (**New Fig. 3g, h and S3a, b**).

Response Fig. 17

(a and b) Immunoblot and Calcein-AM assay were performed to examine the expression of DMT1 and TFR1, and cellular iron level in control and YAP-silenced MCF7 cell lines with over-expression of *LncRIM* or YAP. (mean \pm SD, n=3 independent experiment; n.s., not significant, ***P < 0.001, one-way ANOVA analysis).

(c and d) RT-qPCR detection of *CYR61*, *CTGF* in control and YAP-silenced MCF7 cell lines with over-expression of *LncRIM* or YAP. (mean \pm SD, n=3 independent experiment; n.s., not significant, ***P < 0.001, one-way ANOVA analysis).

We appreciated all of the reviewers characterized our revised manuscript detailed and thorough. We had initially the experiments that the reviewer had inquired about and rewritten any portions of the manuscript that may be confusing to the readers and conduct additional experiments that strengthen the biological relevance of our findings. Those changes in the revised manuscript are highlighted in red.

We believe that the findings in this paper strikingly advance current knowledge about lncRNAs, proposing their roles as mediators required for a specific metabolic-associated signaling event in cells. Importantly, a finding with potentially important clinical implications is the unexpected observations that depletion of *LncRIM* *in vivo* can effectively inhibit breast cancer progression, which has been an overarching challenge in the cancer field. We hope that you will find our manuscript suitable for publication in *Nature Communications*.

REVIEWERS' COMMENTS

Reviewer #2 (Remarks to the Author):

The authors have made further improvement of the manuscript. I think the manuscript is suitable for publication. I have the following minor comments.

I commented several times on the difficulty to appreciate the quality of immunohistology and iron staining data. The authors provide valuable magnification of microscopy images in their rebuttal. It's a pity that these inserts are not added to the manuscript.

I noticed in Fig. 1o that HEPH is drawn inside the cell. HEPH is facing the extracellular space. Please correct.

Fig. 5: there are still y-axis titles with space character missing (5e and 5k).

Rebuttal page 13,14- iron in microenvironment cells.

Discussion lanes 475-488: It is not clear how the data on iron metabolism in tumors of the leptomeningeal space (reference 34) relate to anemia and erythrocyte iron recycling.

Rebuttal page 14: quantification of data in Fig. 1m.

If I understand correctly, the authors modified the contrast/brightness settings of their images and now the data look different, from no difference in the initial version of the manuscript to a ~80% difference (with a p value below 0.001) in the revised version of the paper. This is possible, but the authors will understand that such situation can raise doubt.

Response Figure 3a:

Interpretation of the data presented in Fig. 1n. It would be good to include the numbers provided in Response Fig. 3 to the manuscript (this can be done in the text, not necessarily in a figure) and acknowledge the fact that the difference due to LncRIM is very minor in this assay (around 70% versus 50% is not much difference, the effect of DFO dominates).

Rebuttal page 15:

Given that the authors quantified the western blots and assessed statistical significance, I suggest to add data deviation + statistical significance to the ratio given below each western blot panel.

Rebuttal page 20:

In the legend to Figure S3i, be more precise and indicate that the red characters correspond to the YAP/TEAD4 motif that is DELETED in the mutant reporter construct.

Figure 3i is still not convincing, the fold change in reporter expression induced by YAP is the same with and without the DMT1 promoter. Why not replacing it by figure S3i+j, which is far more convincing?

Rebuttal page 21,22:

The authors must be more explicit about the limitation of their KD experiment. The western-blot data show blunted induction of DMT1 and TFR1 by YAP. IRPs are typically present in excess, if IRP2 would have been depleted more efficiently, the induction of TFR1/DMT1 by YAP may have been even more affected by the lack of IRP2. This would indicate that the effect of YAP may not be totally independent of IRP regulation. I suggest the authors acknowledge in the text this limitation as a note stating that the effect of YAP is (perhaps) to a large extent independent of IRP2, but not totally.

Rebuttal page 24,25:

The authors nicely show that iron does not affect NF2 pulldown efficiency (response Fig. 6) and hence the enrichment of LncRIM in Fig. 4g is legit. The authors could very simply mention that in the results section when describing the data displayed in Fig. 4g. This shall not be mentioned in the discussion (lanes 471-474), this is more a technical issue.

Rebuttal page 27,28 (response Fig. 25):

I am confused. Why writing that Fig. 5c is now new Fig. S5c? Do the authors mean Fig. S5c is new Fig. S5c? Also, there is no change in the figure, why writing it is a new figure?

Source data:

Fig. 2b: the membrane is cut just above the signal, but in the figure the signal appears more or less centred around the signal. This is misleading, there is nothing above the signal in the original data source.

Western blots in Fig. S2j are overall of very poor quality.

Reviewer #3 (Remarks to the Author):

The authors have addressed most of my earlier concerns.

Reviewer #2 (Remarks to the Author):

The authors have made further improvement of the manuscript. I think the manuscript is suitable for publication. I have the following minor comments.

Response: We appreciated your effort to review our revision and your approval of our work! We have carefully revised our manuscript accordingly based on the suggestions and included our point-by-point response letter. Those changes in the revised manuscript are highlighted with yellow background.

I commented several times on the difficulty to appreciate the quality of immunohistology and iron staining data. The authors provide valuable magnification of microscopy images in their rebuttal. It's a pity that these inserts are not added to the manuscript.

Response: Thanks for your suggestion in the last version to help better revise our manuscript! Considering your suggestion, we inserted magnification of microscopy images of partial key data in the manuscript (**New Fig. 5d, S5g and 6a**). And we are sorry that we can't enlarge all the figures due to the space.

I noticed in Fig. 1o that HEPH is drawn inside the cell. HEPH is facing the extracellular space. Please correct.

Response: Thanks for pointing out this issue! We have corrected this in the revised manuscript.

Fig. 5: there are still y-axis titles with space character missing (5e and 5k).

Response: Thanks for pointing out this issue! We have corrected this accordingly and carefully in the revised manuscript.

Rebuttal page 13,14- iron in microenvironmen cells.

Discussion lanes 475-488: It is not clear how the data on iron metabolism in tumors of the leptomengial space (reference 34) relate to anemia and erythrocyte iron recycling.

Response: Thanks for pointing out this issue! We apologized for this confusing description and we have re-written this part in the revised manuscript (Lanes:516-526).

Rebuttal page 14: quantification of data in Fig. 1m.

If I understand correctly, the authors modified the contrast/brightness settings of their images and now the data look different, from no difference in the initial version of the manuscript to a ~80% difference (with a p value below 0.001) in the revised version of the paper. This is possible, but the authors will understand that such situation can raise doubt.

Response: Thank you very much for pointing out this issue in the past revision to help better revise our manuscript! As you mentioned, we have recalculated the images in the later revision and we apologized again for our negligence!

Response Figure 3a:

Interpretation of the data presented in Fig. 1n. It would be good to include the numbers provided in Response Fig. 3 to the manuscript (this can be done in the text, not necessarily in a figure) and acknowledge the fact that the difference due to *LncRIM* is very minor in this assay (around 70% versus 50% is not much difference, the effect of DFO dominates).

Response: Thanks for this suggestion! And we have included the numbers in the revised manuscript, and acknowledged the limitation of *LncRIM* in this part (Lanes:194-197).

Rebuttal page 15:

Given that the authors quantified the western blots and assessed statistical significance, I suggest to add data deviation + statistical significance to the ratio given below each western blot panel.

Response: Thanks for this suggestion! As shown in the last response letter, we have done three independent biological repeats for each western blot in this study, with similar ratio under the same statistical analysis. Thus, we only chose one group western blot result in our manuscript and added the ration below. Thanks again for your suggestion!

Rebuttal page 20:

In the legend to Figure S3i, be more precise and indicate that the red characters correspond to the YAP/TEAD4 motif that is DELETED in the mutant reporter construct. Figure 3i is still not convincing, the fold change in reporter expression induced by YAP is the same with and without the DMT1 promoter. Why not replacing it by figure S3i+j, which is far more convincing?

Response: Thanks for this suggestion! **Figure S3i, j is now New Figure 3i, k, Figure 3i is now Figure S3i.** We have replaced the Figure as you suggested in the revised manuscript. And we also more precisely pointed out the deleted motif in the mutant reporter construct (Lanes:307-314).

Rebuttal page 21,22:

The authors must be more explicit about the limitation of their KD experiment. The western-blot data show blunted induction of DMT1 and TFR1 by YAP. IRPs are typically present in excess, if IRP2 would have been depleted more efficiently, the induction of TFR1/DMT1 by YAP may have been even more affected by the lack of IRP2. This would indicate that the effect of YAP may not be totally independent of IRP

regulation. I suggest the authors acknowledge in the text this limitation as a note stating that the effect of YAP is (perhaps) to a large extent independent of IRP2, but not totally.

Response: Thanks again for your suggestion to better help us revise our manuscript! We have acknowledged this limitation of *LncRIM*-YAP axis in the text in the revised manuscript (Lanes:47-48, 124-128, 329-335, 339-342, 549-558).

Rebuttal page 24,25:

The authors nicely show that iron does not affect NF2 pulldown efficiency (response Fig. 6) and hence the enrichment of *LncRIM* in Fig. 4g is legit. The authors could very simply mention that in the results section when describing the data displayed in Fig. 4g. This shall not be mentioned in the discussion (lanes 471-474), this is more a technical issue.

Response: Thanks for this suggestion! We have added this description in the results section after **Figure.4g** (Lanes:383-384) and deleted this in the discussion section.

Rebuttal page 27,28 (response Fig. 25):

I am confused. Why writing that Fig. 5c is now new Fig. S5c? Do the authors mean Fig. S5c is new Fig. S5c? Also, there is no change in the figure, why writing it is a new figure?

Response: Thanks for your reminder! We are really sorry for our careless mistakes.

Source data:

Fig. 2b: the membrane is cut just above the signal, but in the figure the signal appears more or less centered around the signal. This is misleading, there is nothing above the signal in the original data source.

Response: Thanks for pointing out this issue! In the last revision, we cut the membrane of **Figure 2b** along the 75kd. This misleading is caused due to the over-strong exposure of merged stripe. Thus, we repeated this data again (**Response Fig.1**) (**New Fig.2b**), and the result also showed the direct association between NF2 and sense-*LncRIM*. Besides, we also provided saw data in the Source Data File.

Response Fig.1

(a) *In vitro*-transcribed biotinylated *LncRIM* transcripts were incubated with GST-NF2 recombinant proteins for RNA pull-down assay.

Western blots in Fig. S2j are overall of very poor quality.

Response: Thanks for pointing out this issue! We have repeated this experiment with purified GST-LATS1-NT and His-NF2 protein (**Response Fig.2**) (**New Fig. S2j, k**). And the result showed that *LncRIM* significantly inhibited the interaction between LATS1 and NF2 in a dose-dependent manner. Besides, approximately ratio of *LncRIM* (25nM) and NF2 (20nM) is sufficient to inhibit NF2-LATS1 association.

Response Fig. 2

(a and b) GST-pull down was performed to test the interaction between LATS1 and NF2 under different concentration of *in vitro*-transcribed *LncRIM* by using 20nmol purified GST-LATS1-NT and 20nmol His-MBP-NF2 proteins.

Reviewer #3 (Remarks to the Author):

The authors have addressed most of my earlier concerns.

Response: We sincerely appreciated reviewer's effort to review our revised manuscript! And thanks for your suggestions and approval of our replies!